# Policy Learning Using Weak Supervision

**Jingkang Wang**[*1,2]    **Hongyi Guo**[*3]    **Zhaowei Zhu**[*4]    **Yang Liu**[4]

University of Toronto[1], Vector Institute[2], Northwestern University[3], UC Santa Cruz[4]

wangjk@cs.toronto.edu   hongyiguo2025@u.northwestern.edu
zwzhu@ucsc.edu   yangliu@ucsc.edu

## Abstract

Most existing policy learning solutions require the learning agents to receive high-quality supervision signals such as well-designed rewards in reinforcement learning (RL) or high-quality expert demonstrations in behavioral cloning (BC). These quality supervisions are usually infeasible or prohibitively expensive to obtain in practice. We aim for a unified framework that leverages the available cheap weak supervisions to perform policy learning efficiently. To handle this problem, we treat the "weak supervision" as imperfect information coming from a *peer agent*, and evaluate the learning agent's policy based on a "correlated agreement" with the peer agent's policy (instead of simple agreements). Our approach explicitly punishes a policy for overfitting to the weak supervision. In addition to theoretical guarantees, extensive evaluations on tasks including RL with noisy rewards, BC with weak demonstrations, and standard policy co-training show that our method leads to substantial performance improvements, especially when the complexity or the noise of the learning environments is high.

## 1  Introduction

Recent breakthroughs in policy learning (PL) open up the possibility to apply reinforcement learning (RL) or behavioral cloning (BC) in real-world applications such as robotics [1, 2] and self-driving [3, 4]. Most existing works require agents to receive high-quality supervision signals, e.g., reward or expert demonstrations, which are either infeasible or expensive to obtain in practice [5, 6].

The outputs of reward functions in RL are subject to multiple kinds of randomness. For example, the reward collected from sensors on a robot may be biased and have inherent noise due to physical conditions such as temperature and lighting [7, 8, 9]. For the human-defined reward, different human instructors might provide drastically different feedback that leads to biased rewards [10]. Besides, the demonstrations by an expert in behavioral cloning (BC) are often imperfect due to limited resources and environment noise [11, 12, 13]. Therefore, learning from weak supervision signals such as noisy rewards [7] or low-quality demonstrations produced by untrustworthy expert [12, 14] is one of the outstanding challenges that prevents a wider application of PL.

Although some works have explored these topics separately in their specific domains [7, 15, 14, 16], there lacks a unified solution for robust policy learning in imperfect situations. Moreover, the noise model as well as the corruption level in supervision signals is often required. To handle these challenges, we first formulate a meta-framework to study RL/BC with weak supervision and call it *weakly supervised policy learning*. Then we propose a theoretically principled solution, PeerPL, to perform efficient policy learning using the available weak supervision without requiring noise rates.

Our solution is inspired by peer loss [17], a recently proposed loss function for learning with noisy labels but does not require the specification of noise rates. In peer loss, the noisy labels are treated as a *peer agent*'s supervision. This loss function explicitly punishes the classifier from simply agreeing with the noisy labels, but would instead reward it for a "correlated agreement" (CA). We adopt a

35th Conference on Neural Information Processing Systems (NeurIPS 2021).

similar idea and treat the "weak supervision" as the noisy information coming from an imperfect *peer agent*, and evaluate the learning agent's policy based on a "correlated agreement" (CA) with the weak supervision signals. Compared to standard reward and evaluation functions that encourage simple agreements with the supervision, our approach punishes "over-agreement" to avoid overfitting to the weak supervision, which offers us a family of solutions that do not require prior knowledge of the corruption level in supervision signals.

To summarize, the contributions in the paper are: (1) We provide a unified formulation of the *weakly supervised policy learning* problems; (2) We propose PeerPL, a new way to perform policy evaluation for RL/BC tasks, and demonstrate how it adapts in challenging tasks including RL with noisy rewards and BC from weak demonstrations; (3) PeerPL is theoretically guaranteed to recover the optimal policy, as if the supervision are of high-quality and clean. (4) Experiment results show strong evidence that PeerPL brings significant improvements over state-of-the-art solutions. Code is online available at: https://github.com/wangjksjtu/PeerPL.

## 1.1 Related Work

**Learning with Noisy Supervision** Learning from noisy supervision is a widely explored topic. The seminal work [18] first proposed an unbiased surrogate loss function to recover the true loss from the noisy label distribution, given the knowledge of the noise rates of labels. Follow-up works offered ways to estimate the noise level from model predictions [19, 20, 21, 22, 23, 24, 25, 26, 27] or label consensuses of nearby representations [28]. Recent works also studied this problem in sequential settings including federated bandit [29] and RL [7]. The former work assumes the noise can be offset by averaging rewards from multiple agents. [7] designs a statistics-based estimation algorithm for noise rates in observed rewards, which can be inefficient especially when the state-action space is huge. Moreover, the error in the estimation can accumulate and amplify in sequential problems. Inspired by recent advances of *peer loss* [17, 30, 31], our solution is able to recover true supervision signals without requiring a priori specification of the noise rates.

**Behavioral Cloning (BC)** Standard BC [32, 33] tackles the sequential decision-making problem by imitating the expert actions using supervised learning. Specifically, it aims to minimize the one-step deviation error over the expert trajectory without reasoning about the sequential consequences of actions. Therefore, the agent suffers from compounding errors when there is a mismatch between demonstrations and real states encountered [33, 34, 35]. Recent works introduce data augmentations [36] and value-based regularization [37] or inverse dynamics models [38, 39] to encourage learning long-horizon behaviors. While being simple and straightforward, BC has been widely investigated in a range of application domains [40, 41] and often yields competitive performance [42, 37]. Our framework is complementary to the current BC literature by introducing a learning strategy from weak demonstrations (e.g., noisy or from a poorly-trained agent) and provides theoretical guarantees on how to retrieve clean policy under mild assumptions [43].

**Correlated Agreement** In [44, 45], a correlated agreement (CA) type of mechanism is proposed to evaluate the correlations between agents' reports. In addition to encouraging a certain agreement between agents' reports, CA also punishes over-agreement when two agents always report identically. Recently, [17, 30, 25] adapt a similar idea to noisy label learning thus offloading the burdens of estimating noise rates. We consider a more challenging sequential decision-making problem and study the convergence rates under noisy supervision signals.

## 2 Policy Learning from Weak Supervision

We begin by reviewing conventional reinforcement learning and behavioral cloning with clean supervision signals. Then we introduce the weak supervision problem in policy learning and define two concrete instantiations: (1) RL with noisy reward and (2) BC using weak expert demonstrations.

### 2.1 Overview of Policy Learning

The goal of policy learning (PL) is to learn a policy $\pi$ that the agent could follow to perform a series of actions in a stateful environment. For *reinforcement learning*, the interactive environment is characterized as an MDP $\mathcal{M} = \langle \mathcal{S}, \mathcal{A}, \mathcal{R}, \mathcal{P}, \gamma \rangle$. At each time $t$, the agent in state $s_t \in \mathcal{S}$ takes an action $a_t \in \mathcal{A}$ by following the policy $\pi : \mathcal{S} \times \mathcal{A} \to \mathbb{R}$, and *potentially* receives a reward $r(s_t, a_t) \in \mathcal{R}$. Then the agent transfers to the next state $s_{t+1}$ according to a transition probability

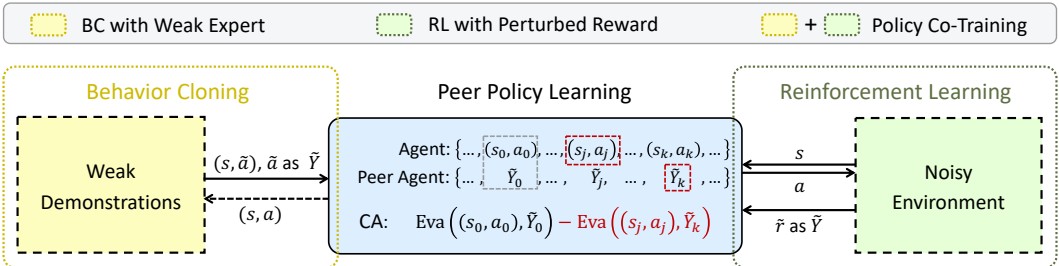

Figure 1: Illustration of *weakly supervised policy learning* and our PeerPL solution with correlated agreement (CA). We use $\widetilde{Y}$ to denote a weak supervision, be it a noisy reward, or a noisy demonstration. Eva stands for an evaluation function. "Peer Agent" corresponds to weak supervision.

function $\mathcal{P}$. We denote the generated trajectory $\tau = \{(s_t, a_t, r_t)\}_{t=0}^{T}$, where $T$ is a finite or infinite horizon. RL algorithms aim to maximize the expected reward over the trajectory $\tau$ induced by the policy: $J^{\text{clean}}(\pi) = \mathbb{E}_{(s_t, a_t, r_t) \sim \tau}[\sum_{t=0}^{T} \gamma^t r_t]$, where $\gamma \in (0, 1]$ is the discount factor.

Another popular policy learning method is *behavioral cloning*. Let $\pi(\cdot|s)$ denotes the distribution over actions formed by $\pi$, and $\pi(a|s)$ be the probability of choosing action $a$ given state $s$ and policy $\pi$. The goal of BC is to mimic the expert policy $\pi_E$ through a set of demonstrations $D_E = \{(s_i, a_i)\}_{i=1}^{N}$ drawn from a distribution $\mathcal{D}_E$, where $(s_i, a_i)$ is the sampled state-action pair from the expert trajectory and $a_i \sim \pi_E(\cdot|s_i)$ Then training a policy with standard BC corresponds to maximizing the following log-likelihood: $J^{\text{clean}}(\pi) = \mathbb{E}_{(s,a) \sim \mathcal{D}_E}[\log \pi(a|s)]$.

In both RL and BC, the learning agent receives supervision through either the *(clean) reward* $r$ by interacting with environments or the *expert policy* $\pi_E$ as observable demonstrations. Consider a particular policy class $\Pi$, the *optimal policy* is then defined as $\pi^* = \arg\max_{\pi \in \Pi} J^{\text{clean}}(\pi)$: $\pi^*$ obtains the maximum expected reward over the horizon $T$ in RL and $\pi^*$ corresponds to the clean expert policy $\pi_E$ in BC. In practice, one can also combine both RL and BC approaches to take advantage of both learning paradigm [46, 47, 15, 43]. Specifically, a recent hybrid framework called policy co-training [43] will be considered in this paper.

## 2.2 Weak Supervision in Policy Learning

The *weak supervision signal* $\widetilde{Y}$ could be noisy reward $\tilde{r}$ for RL or noisy action $\tilde{a}$ from an imperfect expert policy $\tilde{\pi}_E$ for BC, which are noisy versions of the corresponding high-quality supervision signals. See more details below.

**RL with Noisy Reward** Consider a finite MDP $\widetilde{\mathcal{M}} = \langle \mathcal{S}, \mathcal{A}, \mathcal{R}, F, \mathcal{P}, \gamma \rangle$ with noisy reward channels [7], where $\mathcal{R} : \mathcal{S} \times \mathcal{A} \to \mathbb{R}$, and the noisy reward $\tilde{r}$ is generated following a certain function $F : \mathcal{R} \to \widetilde{\mathcal{R}}$. Denote the trajectory a policy $\pi_\theta$ generates via interacting with $\widetilde{\mathcal{M}}$ as $\tilde{\tau}_\theta$. Assume the reward is discrete and has $|\mathcal{R}|$ levels. The noisy reward can be characterized via a unknown matrix $\mathbf{C}_{|\mathcal{R}| \times |\mathcal{R}|}^{\text{RL}}$, where each entry $c_{j,k}$ indicates the flipping probability for generating a possibly different outcome: $c_{j,k}^{\text{RL}} = \mathbb{P}(\tilde{r}_t = R_k | r_t = R_j)$. We call $r$ and $\tilde{r}$ the *true reward* and *noisy reward*.

**BC with Weak Demonstration** Instead of observing the true expert demonstration generated according to $\pi_E$, denote the available weak demonstrations by $\{(s_i, \tilde{a}_i)\}_{i=1}^{N}$, where $\tilde{a}_i$ is is the noisy expert action drawn according to a random variable $\tilde{a}_i = \tilde{\pi}_E(s_i) \sim \tilde{\pi}_E(\cdot|s_i)$, each state-action pair $(s_i, \tilde{a}_i)$ is sampled from distribution $\widetilde{\mathcal{D}}_E$. Note there may exist two randomness factors in getting $\tilde{a}_i$: uncertainty in true policy $\pi_E$ and noise from imperfect policy $\tilde{\pi}_E$. In particular, we do not consider the former randomness in theoretical analyses: given the output distribution $\pi_E(\cdot|s_i)$, only one deterministic action $\pi_E(s_i)$ is taken by expert. This is because with uncertainty in true expert actions, it is hard to distinguish a clean case with true expert actions from the weak supervision case without addition knowledge. Similar assumptions are also adopted in [23, 28]. The noisy action is modeled by a unknown confusion matrix $\mathbf{C}_{|\mathcal{A}| \times |\mathcal{A}|}^{\text{BC}}$, where each entry $c_{j,k}$ indicates the flipping probability for taking a sub-optimal action that differs from $\pi_E(s)$: $c_{j,k}^{\text{BC}} = \mathbb{P}(\tilde{\pi}_E(s) = A_k | \pi_E(s) = A_j)$, $A_k$ and $A_j$ denote the $k$-th and the $j$-th action from the action space $\mathcal{A}$. In the above definition, we assume the noisy action $\tilde{a}_i$ is independent of the state $s$ given the deterministic expert action $\pi_E(s)$,

i.e., $\mathbb{P}(\tilde{a}_i|\pi_E(s_i)) = \mathbb{P}(\tilde{a}_i|s_i, \pi_E(s_i))$. We aim to recover $\pi^*$ as if we were able to access the quality expert demonstration $\pi_E$ instead of $\tilde{\pi}_E$.

**Knowledge of C** Recall $\mathbf{C}$: $\mathbf{C}_{|\mathcal{R}|\times|\mathcal{R}|}^{\mathrm{RL}}$ or $\mathbf{C}_{|\mathcal{A}|\times|\mathcal{A}|}^{\mathrm{BC}}$ is unknown in practice. While recent works estimate this matrix [26, 23, 28] in supervised classification problems, it is still challenging to generalize them to a sequential setting [7]. When $\mathbf{C}$ is not perfectly estimated, the estimation error of $\mathbf{C}$ may lead to unexpected state-action pairs then the error of reward estimates will be accumulated in sequential learning. Besides, estimating $\mathbf{C}$ involves extra computation burden. In contrast, our method gets rid of the above issues since it is free of any knowledge of $\mathbf{C}$ and leads to more robust policy learning algorithms.

**Learning Goal** With full supervision, both RL and BC can converge to the optimal policy $\pi^*$. However, when only weak supervision is available, with an over-parameterized model such as a deep neural network, the learning agent will easily memorize the weak supervision and learn a biased policy [48]. In our meta framework, instead of converging to any biased policy, we focus on learning the optimal policy $\pi^*$ with only a weak supervision sequence denoted as $\{(s_t, a_t), \widetilde{Y}_t\}_{t=1}^T$ (RL) or $\{(s_i, a_i), \widetilde{Y}_i\}_{i=1}^N$ (BC).

# 3   PeerPL: Weakly Supervised PL via Correlated Agreement

To deal with weak supervision in PL, we propose a unified and theoretically principled framework PeerPL. We treat the weak supervision as information coming from a "peer agent", and then evaluate the policy using a certain type of "correlated agreement" function between the learning policy and the peer agent's information.

## 3.1   A Unified Evaluation Function

We use an evaluation function $\mathsf{Eva}_\pi((s_i, a_i), \widetilde{Y}_i)$ to evaluate a taken policy $\pi$ at agent state $(s_i, a_i)$ using the weak supervision $\widetilde{Y}_i$. For RL, $\mathsf{Eva}_\pi$ is the instance-wise measure (negative loss) for different RL algorithms, which is a function of the noisy reward $\tilde{r}$ received at $(s_i, a_i)$. In the BC setting, $\mathsf{Eva}_\pi$ is the loss to evaluate the action $a_i$ taken by the agent given the expert's demonstration $\tilde{a}_i$. Note that the larger the $\mathsf{Eva}_\pi$ is at state $(s_i, a_i)$, the better it follows the supervision $\widetilde{Y}_i$. Specifically, we have

$$\mathsf{Eva}_\pi^{\mathrm{RL}}((s,a), \tilde{r}) = -\ell(\pi, (s,a,\tilde{r})) \text{ (RL)} \quad \text{and} \quad \mathsf{Eva}_\pi^{\mathrm{BC}}((s,a), \tilde{a}) = \log \pi(\tilde{a}|s) \text{ (BC)},$$

where the RL loss function $\ell$ can be temporal difference error [49, 50] or the policy gradient loss [51]. Furthermore, we let $J(\pi)$ denote the function that evaluates policy $\pi$ under a set of state action pairs with weak supervision sequence $\{(s_i, a_i), \widetilde{Y}_i\}_{i=1}^N$, i.e.,

$$J(\pi) = \mathbb{E}_{(s,a)\sim\tau}[\mathsf{Eva}_\pi((s,a), \widetilde{Y})].$$

Then the goal of weakly supervised policy learning is to recover the optimal policy $\pi^*$ as if we receive clean supervision $Y$. Note that directly maximizing $J(\pi)$ might result in sub-optimal performance due to the weak supervisions. The above unified notations are only for better delivery of our framework and we still treat PL as a sequential decision problem.

## 3.2   Overview of the Idea: Correlated Agreement with Weak supervision

We first present the general idea of our PeerPL framework using a concept named correlated agreement (CA). For each weakly supervised sample $((s_i, a_i), \widetilde{Y}_i)$, we randomly sample (with replacement) two other peer samples indexed by $j$ and $k$. Then we take the state-action pair $(s_j, a_j)$ of sample $j$ and the supervision signal $\widetilde{Y}_k$ of sample $k$, and evaluate $((s_i, a_i), \widetilde{Y}_i)$ as follows:

$$\text{CA with Weak Supervision:} \quad \mathsf{Eva}_\pi((s_i, a_i), \widetilde{Y}_i) - \mathsf{Eva}_\pi((s_j, a_j), \widetilde{Y}_k).$$

This operation is illustrated in Figure 1. We further show intuitions and a toy example below.

**Intuition** The first term above encourages an "agreement" with the weak supervision (that a policy agrees with the corresponding supervision), while the second term punishes a "blind" and "over" agreement that happens when the agent's policy always matches with the weak supervision even

on randomly paired traces (noise). The randomly paired instances $j, k$ help us achieve this check. Note our mechanism does not require the knowledge of $\mathbf{C}^{\text{RL}}_{|\mathcal{R}| \times |\mathcal{R}|}$ nor $\mathbf{C}^{\text{BC}}_{|\mathcal{A}| \times |\mathcal{A}|}$, and offers a **prior-knowledge free** way to learn effectively with weak supervision.

**Toy Example** Consider a toy BC setting where the policy fully memorizes the weak supervision and outputs the same sequence of actions given the same sequence of states, i.e.,

$$\text{Weak-supervision: } \tilde{a}_1 = \tilde{a}_2 = \tilde{a}_3 = 1, \tilde{a}_4 = 0; \quad \text{Outputs: } a_1 = a_2 = a_3 = 1, a_4 = 0.$$

Let $\text{Eva}_\pi((s_i, a_i), \tilde{a}_i) = 1$ if the policy output agrees with the weak demonstration ($a_i = \tilde{a}_i$), and 0 otherwise. When the policy fully memorizes weak supervisions, we have:

$$\underline{\textit{Without CA:}} \quad \mathbb{E}[\text{Eva}_\pi((s_i, a_i), \tilde{a}_i)] = 1,$$
$$\underline{\textit{With CA:}} \quad \mathbb{E}[\text{Eva}_\pi((s_i, a_i), \tilde{a}_i) - \text{Eva}_\pi((s_j, a_j), \tilde{a}_k)] = 0.375,$$

where $0.375 = 1 - (0.75^2 + 0.25^2)$ is obtained by considering the probability of randomly paired $a_j$ and $\tilde{a}_k$ matching each other. The above example shows that a full agreement with the weak supervision will instead be punished.

In what follows, we showcase two concrete implementations: PeerRL (peer reinforcement learning) and PeerBC (peer behavioral cloning). We provide algorithms and theoretical guarantees under weak supervisions.

# 4 PeerRL: Peer Reinforcement Learning

We propose the following objective function to punish the over-agreement of parametric policy $\pi_\theta$ based on CA:

$$J^{\text{RL}}(\pi_\theta) = \mathbb{E}\left[\text{Eva}^{\text{RL}}_\pi\big((s_i, a_i), \tilde{r}_i\big)\right] - \xi \cdot \mathbb{E}\left[\text{Eva}^{\text{RL}}_\pi\big((s_j, a_j), \tilde{r}_k\big)\right], \qquad (1)$$

$$\text{where} \quad \text{Eva}^{\text{RL}}_\pi\big((s, a), \tilde{r}\big) = -\ell\big(\pi_\theta, (s, a, \tilde{r})\big). \qquad (2)$$

In (1), the first expectation is taken over $(s_i, a_i, \tilde{r}_i) \sim \tilde{\tau}$ and second one is taken over $(s_j, a_j, \tilde{r}_j) \sim \tilde{\tau}, (s_k, a_k, \tilde{r}_k) \sim \tilde{\tau}$, where $\tilde{\tau}$ is the trajectory specified by the noisy reward function $\tilde{r}$. Recall $j, k$ denote two randomly and independently sampled instances. Loss function $\ell$ depends on the employed RL algorithms, e.g., temporal difference error [49, 50] or the policy gradient loss [51]. The learning sequence is encoded in $\pi$. The objective $J^{\text{RL}}(\pi)$ represents the accumulated peer RL reward. Parameter $\xi \geq 0$ balances the penalty for blind agreements induced by CA.

## 4.1 Peer Reward

In what follows, we consider the $Q$-Learning [52] as the underlying learning algorithm where $\ell(\pi_\theta, (s, a, \tilde{r})) = -\tilde{r}(s, a)$ and demonstrate that the CA mechanism provides strong guarantees for $Q$-Learning with only observing the noisy reward. For clarity, we define *peer RL reward*:

$$\text{Peer Reward:} \quad \tilde{r}_{\text{peer}}(s, a) = \tilde{r}(s, a) - \xi \cdot \tilde{r}',$$

where $\tilde{r}' \overset{\pi_{\text{sample}}}{\sim} \{\tilde{r}(s, a) | s \in \mathcal{S}, a \in \mathcal{A}\}$ is a reward sampled over all state-action pairs according to a fixed policy $\pi_{\text{sample}}$. Note the sampling policy $\pi_{\text{sample}}$ is independent of $\pi$ and the choice of $\pi_{\text{sample}}$ does not affect our theoretical results. We adopt a random sampling strategy in practice. Parameter $\xi \geq 0$ balances the noisy reward and the punishment for blind agreement (with $\tilde{r}'$). We set $\xi = 1$ (for binary case) in the following analysis and treat each $(s, a)$ equally when sampling $\tilde{r}'$. In experiments, we find $\tilde{r}_{\text{peer}}$ is not sensitive to the choice of $\xi$ and keep $\xi$ constant for each run.

**Robustness to Noisy Rewards** Now we show peer reward $\tilde{r}_{\text{peer}}$ offers us an affine transformation of the true reward in expectation, which guarantees that our PeerRL algorithm converges to $\pi^*$. Consider the binary reward setting ($r_+$ and $r_-$) and denote the error in $\tilde{r}$ as $e_+ = \mathbb{P}(\tilde{r} = r_- | r = r_+), e_- = \mathbb{P}(\tilde{r} = r_+ | r = r_-)$ (a simplification of $\mathbf{C}^{\text{RL}}_{|\mathcal{R}| \times |\mathcal{R}|}$ in the binary setting).

**Lemma 1.** *Let $r \in [0, R_{\max}]$ be a bounded reward, $\xi = 1$. Assume $1 - e_- - e_+ > 0$. We have:*

$$\mathbb{E}[\tilde{r}_{\text{peer}}(s, a)] = (1 - e_- - e_+) \cdot \mathbb{E}[r_{\text{peer}}(s, a)] = (1 - e_- - e_+) \cdot \mathbb{E}[r(s, a)] + const,$$

*where $r_{peer}(s, a) = r(s, a) - r'$ is the peer RL reward when observing the true reward $r$, and $r'$ is the true reward corresponding to $\tilde{r}'$.*

Lemma 1 shows that by subtracting the peer penalty term $\tilde{r}'$ from noisy reward $\tilde{r}(s,a)$, $\tilde{r}_{\text{peer}}(s,a)$ recovers the clean and true reward $r(s,a)$ in expectation. Based on Lemma 1, we prove in Theorem A1 that the $Q$-learning agent will converge to the optimal policy *w.p.1* with peer rewards without requiring any knowledge of the corruption in rewards ($\mathbf{C}^{\text{RL}}_{|\mathcal{R}|\times|\mathcal{R}|}$, as opposed to previous work [7] that requires such knowledge). Moreover, we prove in Theorem A2 that to guarantee the convergence to $\pi^*$, the number of samples needed for our approach is no more than $\mathcal{O}(1/(1-e_- -e_+)^2)$ times of the one needed when the RL agent observes true rewards perfectly (see Appendix A).

**Extension**  Even though we only present an analysis for the binary case for $Q$-Learning, our approach is rather generic and is ready to be plugged into modern DRL algorithms. We provide *multi-reward extensions*, implementations with DQN [49] and policy gradient [51] in Appendix A.

### 4.2  Why does Peer Reward Work?

Compared with noisy reward, proposed peer variant is a less biased estimation of true reward *(Benefit-1)*. On the other hand, PeerRL helps break the unstable "tie" states, which might encourage the agent to explore in the early stage [53] *(Benefit-2)*.

**Benefit-1: PeerRL reduces the bias**  We highlight that the biased noise model considered is rather generic, departing from the previous noise assumption such as zero-mean Gaussian noise [8, 9]. In zero-mean noise models, the major focus is on variance reduction so adding the random term $\tilde{r}'$ increases the variance thus resulting in worse estimation. However, in the discrete biased noise model [18], bias correction also plays an important role especially the noise rate is high [7].

Similar to peer reward (Lemma 1), the expectation of the noisy reward writes as: $\mathbb{E}[\tilde{r}(s,a)] = (1-e_- -e_+)\mathbb{E}[r(s,a)] + e_- r_+ + e_+ r_- = (1-e_- -e_+)\mathbb{E}[r(s,a)] + const$. But the constant in peer reward has less effect on the true reward $r$, especially when the noise rate is high. To see this:

$$\text{noisy reward:} \quad \mathbb{E}[\tilde{r}(s,a)] = \eta \cdot \left( \mathbb{E}[r(s,a)] + \frac{e_+}{1-e_- -e_+}r_- + \frac{e_-}{1-e_- -e_+}r_+ \right),$$

$$\text{peer reward:} \quad \mathbb{E}[\tilde{r}_{\text{peer}}(s,a)] = \eta \cdot (\mathbb{E}[r(s,a)] - (1-p_{\text{peer}})r_- - p_{\text{peer}}r_+),$$

where $\eta = 1-e_- -e_+ > 0$, $p_{\text{peer}} \in [0,1]$ denotes the probability that a sample policy sees a reward $r_+$ overall. Since the magnitude of noise terms $\frac{e_-}{1-e_- -e_+}$ and $\frac{e_+}{1-e_- -e_+}$ can potentially become much larger than $1-p_{\text{peer}}$ and $p_{\text{peer}}$ in a high-noise regime, $\frac{e_-}{1-e_- -e_+}r_+ + \frac{e_+}{1-e_- -e_+}r_-$ will dilute the informativeness of $\mathbb{E}[r(s,a)]$. On the contrary, $\mathbb{E}[\tilde{r}_{\text{peer}}(s,a)]$ contains a moderate constant noise thus maintaining more useful training signals of the true reward in practice. In summary, although peer reward (similar to the surrogate reward in previous literature [7]) increases the variance (no free-lunch), it will lead to a better estimation of the true reward due to lower bias.

**Benefit-2: PeerRL helps break ties**  For RL, "tie" states indicate that the rewards for different states are the same, which are less informative as they neither serve as positive nor negative examples. Due to the discrete nature of the noise model, adding a randomly sampled penalty term helps break the tie states and treats them as either positive examples or negative examples such that it can encourage exploration in the early stage, which has similar intuitions to some RL exploration works [53]. It has also been demonstrated that reducing the uncertainty, a.k.a. pushing confident predictions, makes the learning robust to weak-supervisions in supervised learning [17, 54]. On the other

2-state Markov process (no actions)

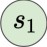 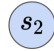

$r_1 \sim \text{clamp}[\mathcal{N}(0.6,1), \min=0, \max=1]$

$r_2 \sim \text{clamp}[\mathcal{N}(0.4,1), \min=0, \max=1]$

| | Correct | Tie | Incorrect |
|---|---|---|---|
| baseline | 54.6% | 5.6% | 39.8% |
| PeerRL | **58.0%** | **0.3%** | **41.7%** |

hand, it is known that positive examples are sparse yet important in RL. To leverage these useful experiences sufficiently, experience replay [55, 56] is invented to store and up-sample the positive examples for faster convergence. Tie breaking potentially provides an alternative way to access more positive examples. To illustrate *tie-breaking* phenomenon when using peer reward, we consider a two-state Markov process (no actions) with bounded Gaussian noise and see how well we could infer which state was better by correcting the reward signals. We collect two observations for each state and conduct $10^4$ trials to calculate the success rate of inferring which state has larger returns ("correct" in the Table). As we can see, PeerRL exploits the "discreteness" of the reward thus breaking ties to obtain more examples with good-quality supervision. More examples on varied noise models (bounded continuous noise, discrete noise) are deferred to Appendix B.

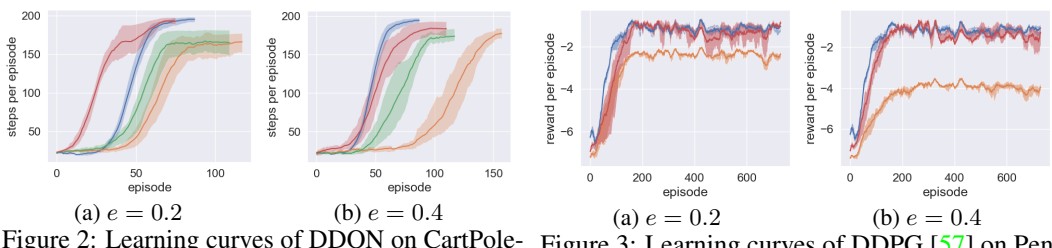

Figure 2: Learning curves of DDQN on CartPole-v0 with true reward ($r$) ■, noisy reward ($\tilde{r}$) ■, surrogate reward [7] ($\hat{r}$) ■, and peer reward ($\tilde{r}_{\text{peer}}$, $\xi = 0.2$) ■.

Figure 3: Learning curves of DDPG [57] on Pendulum with true reward ($r$) ■, noisy reward ($\tilde{r}$) ■, and peer reward ($\tilde{r}_{\text{peer}}$, $\xi = 0.2$) ■.

## 5 PeerBC: Peer Behavioral Cloning

Similarly, we present our CA solution in the setting of behavioral cloning (PeerBC). In BC, the supervision is given by the weak expert's noisy trajectory. At each iteration, the agent learns under weak supervision $\tilde{a}$, and the training samples are generated from the distribution $\widetilde{\mathcal{D}}_E$ determined by the weak expert. The $\mathsf{Eva}_\pi^{\text{BC}}$ function in BC evaluates the agent policy $\pi_\theta$, parametrized by $\theta$, and the weak trajectory $\{(s_i, \tilde{a}_i)\}_{i=1}^N$ using $\ell(\pi_\theta, (s_i, \tilde{a}_i))$, where $\ell$ is an arbitrary classification loss. Taking the cross-entropy for instance, the objective of PeerBC is:

$$J^{\text{BC}}(\pi_\theta) = \mathbb{E}\Big[\mathsf{Eva}_\pi^{\text{BC}}\big((s_i, a_i), \tilde{a}_i\big)\Big] - \xi \cdot \mathbb{E}\Big[\mathsf{Eva}_\pi^{\text{BC}}\big((s_j, a_j), \tilde{a}_k\big)\Big], \tag{3}$$

$$\text{where} \quad \mathsf{Eva}_\pi^{\text{BC}}\big((s, a), \tilde{a}\big) = -\ell\big(\pi_\theta, (s, \tilde{a})\big) = \log \pi_\theta(\tilde{a}|s). \tag{4}$$

In (3), the first expectation is taken over $(s_i, \tilde{a}_i) \sim \widetilde{\mathcal{D}}_E, a_i \sim \pi(\cdot|s_i)$ and the second is taken over $(s_j, \tilde{a}_j) \sim \widetilde{\mathcal{D}}_E, a_j \sim \pi(\cdot|s_j), (s_k, \tilde{a}_k) \sim \widetilde{\mathcal{D}}_E, a_k \sim \pi(\cdot|s_k)$. Again, the second $\mathsf{Eva}_\pi^{\text{BC}}$ term in $J^{\text{BC}}$ serves the purpose of punishing over-agreement with the weak demonstration. Similarly, $\xi \geq 0$ is a parameter to balance the penalty for blind agreements.

**Robustness to Noisy Demonstrations** We prove that the policy learned by PeerBC converges to the expert policy when observing a sufficient amount of weak demonstrations. We focus on the binary action setting for theoretical analyses, where the action space is given by $\mathcal{A} = \{A_+, A_-\}$ and the weakness or noise in the weak expert $\tilde{\pi}_E$ is quantified by $e_+ = \mathbb{P}(\tilde{\pi}_E(s) = A_-|\pi_E(s) = A_+)$ and $e_- = \mathbb{P}(\tilde{\pi}_E(s) = A_+|\pi_E(s) = A_-)$. Let $\pi_{\widetilde{D}_E}$ be the optimal policy for maximizing the objective in (3) with imperfect demonstrations $\widetilde{D}_E$ (a particular set of with $N$ i.i.d. imperfect demonstrations). Note $\ell(\cdot)$ is specified as the 0-1 loss: $\mathbb{1}(\pi(s), a) = 1$ when $\pi(s) \neq a$, otherwise $\mathbb{1}(\pi(s), a) = 0$. We have the following upper bound on the error rate.

**Theorem 1.** *Denote by $R_{\widetilde{D}_E} := \mathbb{P}_{(s,a)\sim\mathcal{D}_E}(\pi_{\widetilde{D}_E}(s) \neq a)$ the error rate for PeerBC. When $e_+ + e_- < 1$, with probability at least $1 - \delta$, it is upper-bounded as: $R_{\widetilde{D}_E} \leq \frac{1+\xi}{1-e_--e_+}\sqrt{\frac{2\log 2/\delta}{N}}$.*

Theorem 1 states that as long as weak demonstrations are observed sufficiently, i.e., $N$ is sufficiently large, the policy learned by PeerBC is able to converge to the clean expert policy $\pi_E(s)$ with a convergence rate of $\mathcal{O}(1/\sqrt{N})$.

**Peer Policy Co-Training** Our discussion of BC allows us to study a more challenging co-training task [43]. Given a finite MDP $\mathcal{M}$, there are two agents that receive partial observations and we let $\pi_A$ and $\pi_B$ denote the policies for agent $A$ and $B$. Moreover, two agents are trained jointly to learn with rewards and noisy demonstrations from each other (e.g., at the preliminary training phase). Symmetrically, we consider the case where agent $A$ learns with the demonstrations from $B$ on sampled trajectories, and $\pi_B$ effectively serves as a noisy version of expert policy.

Following [43], we assume a mapping function $f_{A\to B}$ exists that transforms states under view $A$ into $B$. Denote by $\tau^A = \{(s_i^A, a_i^A, r_i^A)\}_{i=1}^N$ the trajectory that $\pi_A$ generates via interacting with the partial world $\mathcal{M}^A$. Then $\pi_B$ replaces each action $a_i^A$ with its selection $\tilde{a}_i^B = \pi_B(f_{A\to B}(s_i^A))$ as the weak supervision. To recover the clean expert policy, we adapt the BC peer evaluation term to the

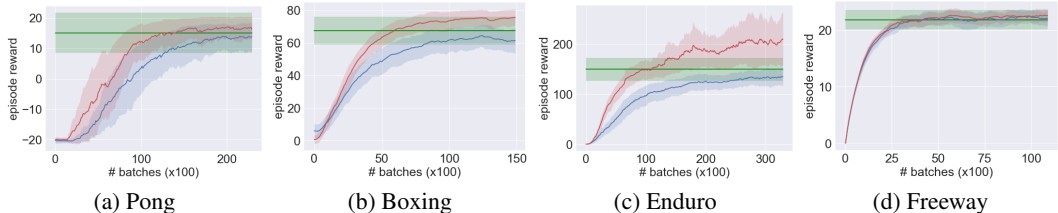

Figure 4: Learning curves of BC on Atari. Standard BC ■, PeerBC (ours) ■, expert ■.

co-learning objective function:

$$J^{\mathrm{CT}}(\pi_\theta) = \mathbb{E}\Big[\mathsf{Eva}_\pi^{\mathrm{RL}}\big((s_i^A, a_i^A), r_i^A\big) + \mathsf{Eva}_\pi^{\mathrm{BC}}\big((s_i^A, a_i^A), \tilde{a}_i^B\big)\Big] - \xi \cdot \mathbb{E}\Big[\mathsf{Eva}_\pi^{\mathrm{BC}}\big((s_j^A, a_j^A), \tilde{a}_k^B\big)\Big], \quad (5)$$

where the first expectation is taken over $(s_i^A, a_i^A, r_i^A) \sim \tau^A$, and $\tilde{a}_i^B = \pi_B(f_{A\to B}(s_i^A))$, and the second is taken over $(s_j^A, a_j^A, r_j^A) \sim \tau^A$, $(s_k^A, a_k^A, r_k^A) \sim \tau^A$, and $\tilde{a}_k^B = \pi_B(f_{A\to B}(s_k^A))$, $\ell$ is the loss function defined in Eqn. (4) to measure the policy difference, and $\mathsf{Eva}_\pi^{\mathrm{RL}}, \mathsf{Eva}_\pi^{\mathrm{BC}}$ are defined in Eqn. (2) and (4) respectively. The full algorithm PeerCT is provided in Algorithm 1. We omit detailed discussions on the convergence of PeerCT - it can be viewed as a straight-forward extension of Theorem 1 in the context of co-training.

## 6 Experiments

We evaluate our solution in three challenging weakly supervised PL problems. Experiments on control games and Atari show that, without any prior knowledge of the noise, our approach is able to leverage weak supervision more effectively.

**Experiment Setup & Baselines** We evaluate PeerPL on a wide variety of control and Atari games. For RL with noisy reward, we add synthetic noise to reward signals and compare with previous work [7], where an unbiased estimator of true reward is constructed

---

**Algorithm 1** Peer policy co-training (PeerCT)

**Require:** Views $A$, $B$, MDPs $\mathcal{M}^A$, $\mathcal{M}^B$, policies $\pi_A, \pi_B$, mapping functions $f_{A\to B}, f_{B\to A}$ that maps states from one view to the other view, CA coefficient $\xi$, step size $\beta$ for policy update.
1: **repeat**
2:     Run $\pi^A$ to generate trajectories $\tau^A = \{(s_i^A, a_i^A, r_i^A)\}_{i=1}^N$.
3:     Run $\pi^B$ to generate trajectories $\tau^B = \{(s_j^B, a_j^B, r_j^B)\}_{j=1}^M$.
4:     Agents label the trajectories for each other
$$\tilde{\tau}^A \leftarrow \big\{(s_i^A, \pi_B(f_{B\leftarrow A}(s_i^A)))\big\}_{i=1}^N,$$
$$\tilde{\tau}^B \leftarrow \big\{(s_j^B, \pi_A(f_{A\leftarrow B}(s_j^B)))\big\}_{j=1}^M.$$
5:     Update policies: $\pi_{\{A,B\}} \leftarrow \pi_{\{A,B\}} + \beta \cdot \nabla J^{\mathrm{CT}}(\pi_{\{A,B\}})$
6: **until** convergence

---

by approximating the confusion matrix. For BC from weak demonstrations, we adopt not fully converged PPO agents as the weak experts and unroll the trajectories. We also consider a standard policy co-training setting [43] without any synthetic noise added and compare PeerCT with single-view training paradigm and CoPiEr [43].

### 6.1 PeerRL with Noisy Reward

***CartPole-v0*:** We first evaluate our method in RL with noisy reward setting. Following [7], we consider the binary reward $\{-1, 1\}$ for Cartpole where the symmetric noise is synthesized with different error rates $e = e_- = e_+$. We choose DQN [49] and DDQN [50] algorithms and train the models for 10,000 steps. We repeat each experiment 10 times with different random seeds and leave extra results in Appendix D. Figure 2 shows the learning curves for DDQN with different approaches in noisy environments ($\xi = 0.2$) [1]. Since the number of training steps is fixed, the faster the algorithm converges, the fewer total episodes the agent will involve thus the learning curve is on the left side. As a consequence, the proposed peer reward outperforms other baselines significantly even in a high-noise regime (e.g., $e = 0.4$). Table 1 provides quantitative results on the average reward $\mathcal{R}_{avg}$ and total episodes $N_{epi}$. We find the agents with peer reward lead to a larger $\mathcal{R}_{avg}$ (less generalization error) and a smaller $N_{epi}$ (faster convergence) consistently.

---

[1]We analysed the sensitivity of $\xi$ and found the algorithm performs reasonable when $\xi \in (0.1, 0.4)$. More insights and experiments with varied $\xi$ is deferred to Appendix D.

Table 1: Numerical performance of DDQN on CartPole with true reward ($r$), noisy reward ($\tilde{r}$), surrogate reward $\hat{r}$ [7], and peer reward $\tilde{r}_{\text{peer}}(\xi = 0.2)$. $\mathcal{R}_{avg}$ denotes average reward per episode after convergence, the higher ($\uparrow$) the better; $N_{epi}$ denotes total episodes involved in 10,000 steps, the lower ($\downarrow$) the better. Note $0 \leq e < 0.5$.

| | | $e = 0.1$ | | $e = 0.2$ | | $e = 0.3$ | | $e = 0.4$ | |
|---|---|---|---|---|---|---|---|---|---|
| | | $\mathcal{R}_{avg} \uparrow$ | $N_{epi} \downarrow$ | $\mathcal{R}_{avg} \uparrow$ | $N_{epi} \downarrow$ | $\mathcal{R}_{avg} \uparrow$ | $N_{epi} \downarrow$ | $\mathcal{R}_{avg} \uparrow$ | $N_{epi} \downarrow$ |
| DDQN | $r$ | $195.6 \pm 3.1$ | $101.2 \pm 3.2$ | $195.6 \pm 3.1$ | $101.2 \pm 3.2$ | $195.6 \pm 3.1$ | $101.2 \pm 3.2$ | $195.2 \pm 3.0$ | $101.2 \pm 3.3$ |
| | $\tilde{r}$ | $185.2 \pm 15.6$ | $114.6 \pm 6.0$ | $168.8 \pm 13.6$ | $123.9 \pm 9.6$ | $177.1 \pm 11.2$ | $133.2 \pm 9.1$ | $185.5 \pm 10.9$ | $163.1 \pm 11.0$ |
| | $\hat{r}$ | $183.9 \pm 10.4$ | $110.6 \pm 6.7$ | $165.1 \pm 18.2$ | $113.9 \pm 9.6$ | $\mathbf{192.2 \pm 10.9}$ | $115.5 \pm 4.3$ | $179.2 \pm 6.6$ | $125.8 \pm 9.6$ |
| | $\tilde{r}_{\text{peer}}$ | $\mathbf{198.5 \pm 2.3}$ | $\mathbf{86.2 \pm 5.0}$ | $\mathbf{195.5 \pm 9.1}$ | $\mathbf{85.3 \pm 5.4}$ | $174.1 \pm 32.5$ | $\mathbf{88.8 \pm 6.3}$ | $\mathbf{191.8 \pm 8.5}$ | $\mathbf{106.9 \pm 9.2}$ |

(a) Acrobot      (b) CartPole      (c) Pong      (d) Breakout

Figure 5: Policy co-training on control/Atari. Single view ■, [43] ■, PeerCT (ours) ■.

**Pendulum**: We further conduct experiments on a continuous control task *Pendulum*, where the goal is to keep a frictionless pendulum standing up. Since the rewards in pendulum are continuous: $r \in (-16.3, 0.0]$, we discretized it into 17 intervals: $(-17, -16], (-16, -15], \cdots, (-1, 0]$, with its value approximated using its maximum point. We test DDPG [57] with uniform noise in this environment following [7]. In Figure 3, the RL agents with the proposed CA objective successfully converge to the optimal policy under different amounts of noise. On the contrary, the agents with noisy rewards suffer from biased noise, especially in a high-noise regime.

**Analysis of the benefits in PeerRL** More surprisingly, we observed that the agents on CartPole with peer reward even lead to faster convergence than the ones observing true reward perfectly when the noise rate $e$ is small. This indicates the possibility of other benefits to further promote peer reward, other than the noise reduction one we primarily focused on. We hypothesize this is because (1) the peer penalty term breaks the tie states (Benefit-2 in Section 4.1) and encourages explorations in RL; (2) PeerRL scales the reward signals appropriately for easier learning; (3) the human-specific "true reward" might be also imperfect which leads to a weak supervision scenario. We emphasize that the advantage of recovering from noisy reward signal is non-negligible, especially in a high-noise regime (e.g., $e = 0.4$ in Figure 2 and 3).

## 6.2 PeerBC from Weak Demonstrations

**Atari**: In BC setting, we evaluate our approach on four vision-based Atari games. For each environment, we train an imperfect RL model with PPO [58] algorithm. Here, "imperfect" means the training is terminated before convergence when the performance is about $70\% \sim 90\%$ as good as the fully converged model. We then collect the imperfect demonstrations using the expert model and generate 100 trajectories for each environment. The results are reported under three random seeds.

Figure 4 shows that our approach outperforms standard BC and even the expert it learns from. Note that during the whole training process, the agent never learns by interacting directly with the environment but only have access to the expert trajectories. Therefore, we owe this performance gain to PeerBC's strong ability for learning from weak supervision. The peer term we add not only provably eliminates the effects of noise but also extracts useful strategy from the demonstrations. As shown in Table 2, our approach consistently outperforms the expert and standard BC. We provide the sensitivity analysis of $\xi$ in Appendix D.

**Comparison with imitation learning baselines** We further extend the empirical study to imitation learning (IL) algorithms on *CartPole-v1*. To collect weak demonstrations, we train a PPO agent for 50k iterations that are not fully converged. As shown in Figure 6, standard IL algorithms such as BC, AIRL [37], or GAIL [59] cannot handle noisy demonstrations well and lead to sub-optimal performance. Our PeerBC brings 18% improvement over standard BC by penalizing blind agreements with the weak demonstrations. We remark that performance of PeerBC is worse than DAgger due to notorious distribution shift issue. To further improve performance, we train PeerBC in the DAgger fashion (Peer-DAgger) by querying the imperfect expert to augment the training sets. Not surprisingly,

Table 2: BC from weak demonstrations. PeerBC successfully recovers better policies than expert.

| Environment | | Pong | Boxing | Enduro | Freeway | Lift ($\uparrow$) |
|---|---|---|---|---|---|---|
| Expert | | $15.1 \pm 6.6$ | $67.5 \pm 8.5$ | $150.1 \pm 23.0$ | $21.9 \pm 1.7$ | - |
| Standard BC | | $14.7 \pm 3.2$ | $56.2 \pm 7.7$ | $138.9 \pm 14.1$ | $22.0 \pm 1.3$ | $-6.6\%$ |
| PeerBC | $\xi = 0.2$ | $\mathbf{18.8 \pm 0.6}$ | $67.2 \pm 8.4$ | $177.9 \pm 29.3$ | $\mathbf{22.5 \pm 0.6}$ | $+11.3\%$ |
| | $\xi = 0.5$ | $16.6 \pm 4.0$ | $\mathbf{75.6 \pm 5.4}$ | $\mathbf{230.9 \pm 73.0}$ | $22.4 \pm 1.3$ | $+\mathbf{19.5\%}$ |
| | $\xi = 1.0$ | $16.7 \pm 4.3$ | $69.7 \pm 4.7$ | $230.4 \pm 61.6$ | $8.9 \pm 4.9$ | $+2.0\%$ |
| Fully converged PPO | | $20.9 \pm 0.3$ | $89.3 \pm 5.4$ | $389.6 \pm 216.9$ | $33.3 \pm 0.8$ | - |

Table 3: Comparison with single view training and CoPiEr [43] on standard policy co-training.

| Environment | | Acrobot | CartPole | Pong | Breakout |
|---|---|---|---|---|---|
| Single View | A | $-136.6 \pm 15.6$ | $172.8 \pm 5.5$ | $17.8 \pm 0.6$ | $148.0 \pm 16.5$ |
| | B | $-126.4 \pm 8.0$ | $186.7 \pm 8.1$ | $17.7 \pm 0.5$ | $137.8 \pm 12.5$ |
| CoPiEr | A | $-136.2 \pm 5.2$ | $174.1 \pm 5.1$ | $16.8 \pm 0.5$ | $107.5 \pm 5.8$ |
| | B | $-131.5 \pm 4.5$ | $174.3 \pm 5.4$ | $16.5 \pm 0.2$ | $82.7 \pm 6.9$ |
| PeerCT | A | $\mathbf{-87.0 \pm 3.9}$ | $\mathbf{188.8 \pm 2.7}$ | $\mathbf{20.5 \pm 0.4}$ | $263.6 \pm 36.0$ |
| | B | $-87.1 \pm 6.3$ | $184.7 \pm 3.9$ | $20.4 \pm 0.5$ | $\mathbf{268.6 \pm 33.6}$ |

Peer-DAgger surpasses DAgger by a large margin, which indicates that our framework has wide applicability and successfully recovers the true supervision signals. Adapting PeerPL idea to more IL algorithms such as GAIL [59] and DART [35] together with rigorous analysis is left as future works.

**Analysis of benefits in PeerBC** Similarly, the performance improvement of PeerBC might be also coupled with multiple possible factors. (1) The imperfect expert model might be a noisy version of the fully-converged agent since there are less visited states on which the selected actions of the model contains noise. (2) The improvements might be brought up by biasing against high-entropy policies thus PeerBC is useful when the true policy itself is deterministic. We provide more discussions about the second factor in Appendix D.5.

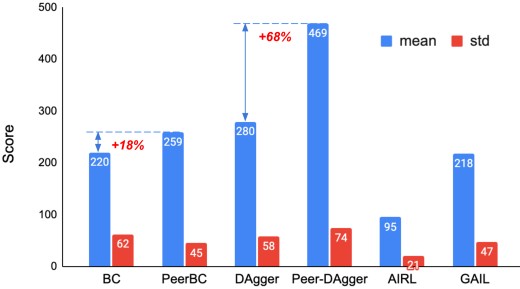

Figure 6: Comparison of imitation learning approaches on *CartPole-v1* with imperfect expert.

### 6.3 PeerCT for Standard Policy Co-training

***Continuous Control/Atari***: Finally, we verify the effectiveness of the PeerCT algorithm in policy co-training setting [43]. This setting is more challenging since the states are partially observable and each agent needs to imitate another agent's behavior that is highly biased and imperfect. Note that we adopt the exact same setting as [43] **without any synthetic noise** included. This implies the potential of our approach to deal with natural noise in real-world applications. Following [43], we mask the first two dimensions respectively in the state vector to create two views for co-training in classic control games (Acrobot and CartPole). Similarly, the agent either removes all even index coordinates (view-$A$) in the state vector or removing all odd index ones (view-$B$) on Atari games. As shown in Table 3 and Figure 5, PeerCT algorithm outperforms training from single view, and CoPiEr algorithm consistently on both control games ($\xi = 0.5$ in Figure 5a, 5b) and Atari games ($\xi = 0.2$ in Figure 5c, 5d). In most cases, our approach leads to a faster convergence and lower generalization error compared to CoPiEr, showing that our ways of leveraging information from peer agent enables recovery of useful knowledge from highly imperfect supervision.

## 7 Conclusion

We have proposed PeerPL, a weakly supervised policy learning framework to unify a series of RL/BC problems with low-quality supervision signals. In PeerPL, instead of blindly memorizing the weak supervision, we evaluate a learning policy's correlated agreements with the weak supervision. We demonstrate how our method adapts in RL/BC and the hybrid co-training tasks and provide analysis of the convergence rate and sample complexity. Current theorems focus on the specific discrete noise model. Future work may extend it to more general noise scenarios and evaluate our method on real RL/BC systems, such as robotics and self-driving.

## Acknowledgement

We sincerely thank the anonymous reviewers for their insightful suggestions. Our final version benefited substantially from the discussions with Reviewer 58fG. In particular, the tie-breaking analysis together with the code snippet is designed and contributed by Reviewer 58fG. This work is partially supported by the National Science Foundation (NSF) under grant IIS-2007951 and the Office of Naval Research under grant N00014-20-1-2240. Resources used in preparing this research were provided, in part, by the Province of Ontario, the Government of Canada through CIFAR, and companies sponsoring the Vector Institute.

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
