# Supplementary Material
# Policy Learning Using Weak Supervision

## Abstract

In this supplementary material, we first provide theoretical analysis of the convergence rate (Sec A.1) and sample complexity (Sec A.2) for Peer $Q$-Learning algorithm. Then we provide the extension to multi-outcome setting with theoretical proofs (Sec A.3). We also show the extensions to other modern DRL algorithms in Sec A.4, and further discussions on the effectiveness of PeerRL in Sec A.5. We then provide more "tie-breaking" examples on varied noise models together with the python-style code snippet in Sec B. In Sec C, we provide the technical proofs for proposed PeerBC approach under mild assumptions. Then, we report the experimental setup details (Sec D.1), the implementation details (Sec D.2), and additional experiments including complete results for Figure 2 and Table 1 (Sec D.3), sensitivity analysis of peer penalty coefficient $\xi$ (Sec D.4), and study of stochasticity for behavioral cloning policy (Sec D.5). The summary of contents in the supplementary is provided in the following.

## Contents

# A  Analysis of PeerRL

We start this section by providing the proof of the convergence of $Q$-Learning under peer reward $\tilde{r}_{\text{peer}}$ (Theorem A1). Moreover, we give the sample complexity of *phased* value iteration (Theorem A2). In the rest of this section, we show how to extend the proposed method to multi-outcome setting (Section A.3) and modern deep reinforcement learning (DRL) algorithms such as policy gradient [51] and DQN [49, 56] (Section A.4).

## A.1  Convergence

Recall that we consider the binary reward case $\{r_+, r_-\}$, where $r_+$ and $r_-$ are two reward levels. The flipping errors of the reward are defined as $e_+ = \mathbb{P}(\tilde{r}_t = r_- | r_t = r_+)$ and $e_- = \mathbb{P}(\tilde{r}_t = r_+ | r_t = r_-)$. The *peer reward* is defined as $r_{\text{peer}}(s, a) = r(s, a) - r'$, where $r'$ is randomly sampled reward over all state-action pair $(s, a)$. Note that we treat each $(s, a)$ equally when sampling the $r'$ due to lack of the knowledge of true transition probability $\mathcal{P}$. In practice, the agent could only noisy observation of peer reward $\tilde{r}_{\text{peer}}(s, a) = \tilde{r}(s, a) - \tilde{r}'$. We provide the $Q$-learning with peer reward in Algorithm A1.

---

**Algorithm A1** $Q$-Learning with Peer Reward

---

**Require:** $\widetilde{\mathcal{M}} = (\mathcal{S}, \mathcal{A}, \widetilde{\mathcal{R}}, \mathcal{P}, \gamma)$, learning rate $\alpha \in (0, 1)$, initial state distribution $\beta_0$.
1: Initialize $Q$: $\mathcal{S} \times \mathcal{A} \to \mathbb{R}$ arbitrarily
2: **while** $Q$ is not converged **do**
3:     Start in state $s \sim \beta_0$
4:     **while** $s$ is not terminal **do**
5:         Calculate $\pi$ according to $Q$ and exploration strategy
6:         $a \leftarrow \pi(s); s' \sim \mathcal{P}(\cdot | s, a)$
7:         Observe noisy reward $\tilde{r}(s, a)$ and randomly sample another $\tilde{r}'$ from all state-action pairs
8:         Calculate peer reward $\tilde{r}_{\text{peer}}(s, a) = \tilde{r}(s, a) - \tilde{r}'$
9:         $Q(s, a) \leftarrow (1 - \alpha) \cdot Q(s, a) + \alpha \cdot (\tilde{r}_{\text{peer}}(s, a) + \gamma \cdot \max_{a'} Q(s', a'))$
10:        $s \leftarrow s'$
11:    **end while**
12: **end while**
**Ensure:** $Q(s, a)$ and $\pi(s)$

---

We then show the proposed peer reward $\tilde{r}_{\text{peer}}$ offers us an affine transformation of true reward in expectation, which is the key to guaranteeing the convergence for RL algorithms.

**Lemma 1.** *Let $r \in [0, R_{\max}]$ be bounded reward and assume $1 - e_- - e_+ > 0$. Then, if we define the peer reward $\tilde{r}_{\text{peer}}(s, a) = \tilde{r}(s, a) - \tilde{r}'$, in which the penalty term $\tilde{r}'$ is randomly sampled noisy reward over all state-action pair $(s, a)$, we have*

$$\mathbb{E}[\tilde{r}_{\text{peer}}(s, a)] = (1 - e_- - e_+)\mathbb{E}[r_{\text{peer}}(s, a)] = (1 - e_- - e_+)\mathbb{E}[r(s, a)] + const,$$

*where $r_{peer}(s, a)$ is the clean version of peer reward when observing the true reward.*

*Proof.* With slight notation abuse, we let $\tilde{r}_{\text{peer}}, r, \tilde{r}$ represent the random variables $\tilde{r}_{\text{peer}}(s, a), r(s, a), \tilde{r}(s, a)$. Let $\pi(s, a)$ denotes the RL agent's policy. Consider the two terms on the RHS of noisy peer reward separately,

$$\mathbb{E}[\tilde{r}] = \mathbb{P}(r = r_+ | \pi) \cdot \mathbb{E}_{r=r_+} \left[ \mathbb{P}(\tilde{r} = r_- | r = r_+) \cdot r_- + \mathbb{P}(\tilde{r} = r_+ | r = r_+) \cdot r_+ \right] \tag{6}$$

$$+ \mathbb{P}(r = r_- | \pi) \cdot \mathbb{E}_{r=r_-} \left[ \mathbb{P}(\tilde{r} = r_- | r = r_-) \cdot r_- + \mathbb{P}(\tilde{r} = r_+ | r = r_-) \cdot r_+ \right] \tag{7}$$

$$= \mathbb{P}(r = r_+ | \pi) \cdot \mathbb{E}_{r=r_+} \left[ e_+ r_- + (1 - e_+) r_+ \right] \tag{8}$$

$$+ \mathbb{P}(r = r_- | \pi) \cdot \mathbb{E}_{r=r_-} \left[ (1 - e_-) r_- + e_- r_+ \right] \tag{9}$$

$$= \mathbb{P}(r = r_+ | \pi) \cdot \mathbb{E}_{r=r_+} \left[ (1 - e_+ - e_-) \cdot r_+ + e_+ r_- + e_- r_+ \right] \tag{10}$$

$$+ \mathbb{P}(r = r_- | \pi) \cdot \mathbb{E}_{r=r_-} \left[ (1 - e_- - e_+) \cdot r_- + e_- r_+ + e_+ r_- \right] \tag{11}$$

$$= (1 - e_+ - e_-) \mathbb{E}[r] + e_- r_+ + e_+ r_-. \tag{12}$$

Since we are treating the visitation probability of all state-action pair $(s, a)$ equally while sampling the peer penalty $r'$, then the probability of true reward $r$ under this sampling policy $\pi_{\text{sample}}$ is a

constant, denoting as $p_{\text{peer}}$, i.e., $p_{\text{peer}} = \mathbb{P}(r = r_- | \pi_{\text{sample}})$ is a constant. Then we have,

$$\mathbb{E}[\tilde{r}'] = \mathbb{P}(\tilde{r} = r_- | \pi_{\text{sample}}) \cdot r_- + \mathbb{P}(\tilde{r} = r_+ | \pi_{\text{sample}}) \cdot r_+ \tag{13}$$

$$= (e_+ p_{\text{peer}} + (1 - e_-)(1 - p_{\text{peer}})) \cdot r_- + ((1 - e_+)p_{\text{peer}} + e_-(1 - p_{\text{peer}})) \cdot r_+ \tag{14}$$

$$= (1 - e_- - e_+)[(1 - p_{\text{peer}}) \cdot r_- + p_{\text{peer}} \cdot r_+] + e_+ r_- + e_- r_+. \tag{15}$$

As a consequence, we obtain the expectation of peer reward satisfies

$$\mathbb{E}[\tilde{r}_{\text{peer}}] = \mathbb{E}[\tilde{r}] - \mathbb{E}[\tilde{r}'] \tag{16}$$

$$= (1 - e_+ - e_-)\mathbb{E}[r] - (1 - e_- - e_+)[(1 - p_{\text{peer}}) \cdot r_- + p_{\text{peer}} \cdot r_+] \tag{17}$$

$$= (1 - e_- - e_+)\mathbb{E}[r] + \text{const}. \tag{18}$$

Similarly, it is easy to obtain that $\mathbb{E}[r_{\text{peer}}] = \mathbb{E}[r] - [(1 - p_{\text{peer}}) \cdot r_- + p_{\text{peer}} \cdot r_+]$. Therefore, we have $\mathbb{E}[\tilde{r}_{\text{peer}}] = (1 - e_- - e_+)\mathbb{E}[r_{\text{peer}}] = (1 - e_- - e_+)\mathbb{E}[r] + \text{const}.$ □

Lemma 1 shows the proposed peer reward $\tilde{r}_{\text{peer}}$ offers us a "noise-free" positive ($1 - e_- - e_+ > 0$) linear transformation of true reward $r$ in expectation, which is shown the key to govern the convergence. It is widely known in utility theory and reward shaping literature [60, 61, 62] that any positive linear transformations leave the optimal policy unchanged. As a consequence, we consider a "transformed MDP" $\hat{\mathcal{M}}$ with reward $\hat{r} = (1 - e_- - e_+)r + \text{const}$, where the const is the same as the constant in Eqn. (18).

In what follows, we provide the formulation of the concept of "transformed MDP" with the policy invariance guarantee.

**Lemma A1.** *Given a finite MDP $\mathcal{M} = \langle \mathcal{S}, \mathcal{A}, \mathcal{R}, \mathcal{P}, \gamma \rangle$, a transformed MDP $\hat{\mathcal{M}} = \langle \mathcal{S}, \mathcal{A}, \hat{\mathcal{R}}, \mathcal{P}, \gamma \rangle$ with positive linear transformation in reward $\hat{r} := a \cdot r + b$, where $a, b$ are constants and $a > 0$, is guaranteed consistency in optimal policy.*

*Proof.* The $Q$ function for transformed MDP $\hat{\mathcal{M}}$ (denoting as $\hat{Q}$) is given as follows:

$$\hat{Q}(s, a) = \sum_{t=0}^{\infty} \gamma^t \hat{r}_t = \sum_{t=0}^{\infty} \gamma^t (a \cdot r_t + b)$$

$$= a \sum_{t=0}^{\infty} \gamma^t r_t + \sum_{t=0}^{\infty} \gamma^t b$$

$$= a \cdot Q(s, a) + B,$$

where $B = \sum_{t=0}^{\infty} \gamma^t b$ is a constant. Therefore, there is only a postive linear shift ($a > 0$) in $\hat{Q}(s, a)$ thus resulting in invariance in optimal policy for transformed MDP:

$$\hat{\pi}^*(s) = \arg\max_{a \in \mathcal{A}} \hat{Q}^*(s, a) = \arg\max_{a \in \mathcal{A}} [a \cdot Q(s, a) + B]$$

$$= \arg\max_{a \in \mathcal{A}} Q(s, a) = \pi^*(s).$$

□

Lemma A1 states that we only need to analysis the convergence of learned policy $\pi(s)$ to the optimal policy $\hat{\pi}^*(s)$ for transformed MDP $\hat{\mathcal{M}}$, which is equivalent to the optimal policy $\pi(s)^*$ for original MDP. This result is relevant to potential-based reward shaping [60, 61] where a specific class of state-dependent transformation is adopted to speed up the convergence speed of $Q$-Learning meanwhile maintaining the optimal policy invariance. Moreover, a degenerate case for single-step decisions is studied in utility theory [62] which also implies our result.

Finally, we need an auxiliary result (Lemma A2) from stochastic process approximation to analyse the convergence for $Q$-Learning.

**Lemma A2.** *The random process $\{\Delta_t\}$ taking values in $\mathbb{R}^n$ and defined as*

$$\Delta_{t+1}(x) = (1 - \alpha_t(x))\Delta_t(x) + \alpha_t(x)F_t(x)$$

*converges to zero w.p.1 under the following assumptions:*

- $0 \le \alpha_t \le 1$, $\sum_t \alpha_t(x) = \infty$ *and* $\sum_t \alpha_t(x)^2 < \infty$;

- $||\mathbb{E}\left[F_t(x)|\mathcal{F}_t\right]||_W \le \gamma ||\Delta_t||$, *with* $\gamma < 1$;

- *Var* $[F_t(x)|\mathcal{F}_t] \le C(1 + ||\Delta_t||_W^2)$, *for* $C > 0$.

*Here $\mathcal{F}_t = \{\Delta_t, \Delta_{t-1}, \cdots, F_{t-1} \cdots, \alpha_t, \cdots\}$ stands for the past at step $t$, $\alpha_t(x)$ is allowed to depend on the past insofar as the above conditions remain valid. The notation $|| \cdot ||_W$ refers to some weighted maximum norm.*

*Proof of Lemma A2.* See previous literature [63, 64]. □

**Theorem A1.** *(Convergence) Given a finite MDP with noisy reward, denoting as $\widetilde{\mathcal{M}} = \langle \mathcal{S}, \mathcal{A}, \widetilde{\mathcal{R}}, F, \mathcal{P}, \gamma \rangle$, the Q-learning algorithm with peer rewards, given by the update rule,*

$$Q_{t+1}(s_t, a_t) = (1 - \alpha_t)Q_t(s_t, a_t) + \alpha_t \left[ \tilde{r}_{\text{peer}}(s_t, a_t) + \gamma \max_{b \in \mathcal{A}} Q_t(s_{t+1}, b) \right], \quad (19)$$

$$\pi_t(s) = \arg\max_{a \in \mathcal{A}} Q_t(s, a) \quad (20)$$

*converges w.p.1 to the optimal policy $\pi^*(s)$ as long as $\sum_t \alpha_t = \infty$ and $\sum_t \alpha_t^2 < \infty$.*

*Proof.* Firstly, we construct a surrogate MDP $\hat{\mathcal{M}}$ with the positive-linearly transformed reward $\hat{r} = (1 - e_- - e_+) \cdot r + \text{const}$, where const $= -(1 - e_- - e_+)((1 - p) \cdot r_- + p \cdot r_+)$ is a constant. From Lemma A1, we know the optimal policy for $\hat{\mathcal{M}}$ is precisely the optimal policy for $\mathcal{M}$: $\hat{\pi}^*(s) = \pi^*(s)$.

Let $\hat{Q}^*$ denotes the optimal state-action function for this transformed MDP $\hat{\mathcal{M}}$. For notation brevity, we abbreviate $s_t$, $s_{t+1}$, $\tilde{r}_{\text{peer}}(s_t, s_{t+1})$, $Q_t$, $Q_{t+1}$, and $\alpha_t$ as $s$, $s'$, $Q$, $Q'$, $\tilde{r}_{\text{peer}}$ and $\alpha$, respectively.

Subtracting from both sides the quantity $\hat{Q}^*(s, a)$ in Eqn. (20):

$$Q'(s, a) - \hat{Q}^*(s, a) = (1 - \alpha)\left( Q(s, a) - \hat{Q}^*(s, a) \right) + \alpha \left[ \tilde{r}_{\text{peer}} + \gamma \max_{b \in \mathcal{A}} Q(s', b) - \hat{Q}^*(s, a) \right].$$

Let $\Delta_t(s, a) = Q(s, a) - \hat{Q}^*(s, a)$ and $F_t(s, a) = \tilde{r}_{\text{peer}} + \gamma \max_{b \in \mathcal{A}} Q(s', b) - \hat{Q}^*(s, a)$.

$$\Delta_{t+1}(s', a) = (1 - \alpha)\Delta_t(s, a) + \alpha F_t(s, a).$$

In consequence,

$$\mathbb{E}\left[F_t(s, a)|\mathcal{F}_t\right] = \mathbb{E}\left[\tilde{r}_{\text{peer}} + \gamma \max_{b \in \mathcal{A}} Q(s', b)\right] - \hat{Q}^*(s, a)$$

$$= \mathbb{E}\left[\tilde{r}_{\text{peer}} + \gamma \max_{b \in \mathcal{A}} Q(s', b) - \hat{r} - \gamma \max_{b \in \mathcal{A}} \hat{Q}^*(s', b)\right]$$

$$= \mathbb{E}\left[\tilde{r}_{\text{peer}}\right] - \mathbb{E}\left[\hat{r}\right] + \gamma \mathbb{E}\left[\max_{b \in \mathcal{A}} Q(s', b) - \max_{b \in \mathcal{A}} \hat{Q}^*(s', b)\right]$$

$$= \gamma \mathbb{E}\left[\max_{b \in \mathcal{A}} Q(s', b) - \max_{b \in \mathcal{A}} \hat{Q}^*(s', b)\right]$$

$$\le \gamma \mathbb{E}\left[\max_{b \in \mathcal{A}, s' \in \mathcal{S}} \left| Q(s', b) - \hat{Q}^*(s', b) \right|\right]$$

$$= \gamma \mathbb{E}\left[||Q - \hat{Q}^*||_\infty\right] = \gamma ||Q - \hat{Q}^*||_\infty = \gamma ||\Delta_t||_\infty.$$

In above derivations, we utilize the unbiasedness property for peer reward (Lemma 1) and the inequality $\max_{b\in\mathcal{A}} Q(s',b) - \max_{b\in\mathcal{A}} \hat{Q}^*(s',b) \leq \max_{b\in\mathcal{A},s'\in\mathcal{S}} \left| Q(s',b) - \hat{Q}^*(s',b) \right|$.

$$\mathbf{Var}\left[F_t(s,a)|\mathcal{F}_t\right] = \mathbb{E}\left[\left(\tilde{r}_{\text{peer}} + \gamma\max_{b\in\mathcal{A}} Q(s',b) - \hat{Q}^*(s,a) - \mathbb{E}\left[\tilde{r}_{\text{peer}} + \gamma\max_{b\in\mathcal{A}} Q(s',b) - \hat{Q}^*(s,a)\right]\right)^2\right]$$

$$= \mathbb{E}\left[\left(\tilde{r}_{\text{peer}} + \gamma\max_{b\in\mathcal{A}} Q(s',b) - \mathbb{E}\left[\tilde{r}_{\text{peer}} + \gamma\max_{b\in\mathcal{A}} Q(s',b)\right]\right)^2\right]$$

$$= \mathbf{Var}\left[\tilde{r}_{\text{peer}} + \gamma\max_{b\in\mathcal{A}} Q(s',b)\right].$$

Since $\tilde{r}_{\text{peer}}$ is bounded, it can be clearly verified that

$$\mathbf{Var}\left[F_t(s,a)|\mathcal{F}_t\right] \leq C''(1 + ||\Delta_t(s,a)||_\infty^2)$$

for some constant $C'' > 0$. Then, $\Delta_t$ converges to zero w.p.1 from Lemma A2, *i.e.*, $Q(s,a)$ converges to $\hat{Q}^*(s,a)$. As a consequence, we know the policy $\pi_t(s)$ converges to the optimal policy $\hat{\pi}^*(s) = \pi^*(s)$. □

## A.2 Sample Complexity

In this section, we establish the sample complexity for $Q$-Learning with peer reward as discussed in Sec 4. Since the transition probability $\mathcal{P}$ in MDP remains unknown in practice, we firstly introduce a practical sampling model $G(\mathcal{M})$ following previous literature [65, 66, 67]. in which the transition can be observed by calling the generative model. Then the sample complexity is analogous to the number of calls for $G(\mathcal{M})$ to obtain a near optimal policy.

**Definition A1.** *A generative model $G(\mathcal{M})$ for an MDP $\mathcal{M}$ is a sampling model which takes a state-action pair $(s_t, a_t)$ as input, and outputs the corresponding reward $r(s_t, a_t)$ and the next state $s_{t+1}$ randomly with the probability of $\mathbb{P}_a(s_t, s_{t+1})$, i.e., $s_{t+1} \sim \mathbb{P}(\cdot|s,a)$.*

It is known that exact value iteration is not feasible when the agent interacts with generative model $G(\mathcal{M})$ [7, 68]. For the convenience of analysing sample complexity, we introduce a *phased value iteration* following [7, 65, 68].

---

**Algorithm A2** Phased Value Iteration

---

**Require:** $G(\mathcal{M})$: generative model of $\mathcal{M} = (\mathcal{S}, \mathcal{A}, \mathcal{R}, \mathcal{P}, \gamma)$, $T$: number of iterations.
 1: Set $V_T(s) = 0$
 2: **for** $t = T - 1, \cdots, 0$ **do**
 3:    Calling $G(\mathcal{M})$ $m$ times for each state-action pair.

$$\bar{\mathbb{P}}_a(s_t, s_{t+1}) = \frac{\#[(s_t, a_t) \to s_{t+1}]}{m}$$

 4:    Set

$$V(s_t) = \max_{a\in\mathcal{A}} \sum_{s_{t+1}\in\mathcal{S}} \bar{\mathbb{P}}_a(s_t, s_{t+1})\left[r_t + \gamma V(s_{t+1})\right]$$

$$\pi(s) = \arg\max_{a\in\mathcal{A}} V(s_t)$$

 5: **end for**
 6: **return** $V(s)$ and $\pi(s)$

---

Note that $\bar{P}_a(s_t, s_{t+1})$ is the estimation of transition probability $P_a(s_t, s_{t+1})$ by calling $G(\mathcal{M})$ m times. For the simplicity of notations, the iteration index $t$ decreases from $T - 1$ to 0.

We could also adopt peer reward in phased value iteration by replacing Line 4 in Algorithm A2 by

$$V(s_t) = \max_{a\in\mathcal{A}} \sum_{s_{t+1}\in\mathcal{S}} \bar{\mathbb{P}}_a(s_t, s_{t+1})\left[\tilde{r}_{\text{peer}}(s_t, a) + \gamma V(s_{t+1})\right].$$

Then the sample complexity of one variant (phased value iteration) of $Q$-Learning is given as follows:

**Theorem A2.** *(Sample Complexity) Let $r \in [0, R_{\max}]$ be bounded reward, for an appropriate choice of $m$, the phased value iteration algorithm with peer reward $\tilde{r}_{\text{peer}}$ calls the generative model $G(\widetilde{\mathcal{M}})$*

$O\left( \frac{|\mathcal{S}||\mathcal{A}|T}{\epsilon^2(1-e_--e_+)^2} \log \frac{|\mathcal{S}||\mathcal{A}|T}{\delta} \right)$ *times in $T$ epochs, and returns a policy such that for all state $s \in \mathcal{S}$,*

$\left| \frac{1}{\eta} V^\pi(s) - V^*(s) \right| \leq \epsilon$, *w.p.* $\geq 1 - \delta$, $0 < \delta < 1$, *where $\eta = 1 - e_- - e_+ > 0$ is a constant.*

*Proof.* Similar to Theorem A1, we firstly construct a transformed MDP $\hat{\mathcal{M}}$ and the optimal policies for these two MDP are equivalent (Lemma A1). As a result, we could analyse the sample complexity of phased value iteration under $\hat{\mathcal{M}}$.

It is easy to obtain that $\tilde{r}_{\text{peer}} \in [0, R_{\max}]$ and $V^\pi(s) \in \left[ 0, \frac{R_{\max}}{1-\gamma} \right]$ are also bounded. Using Hoeffding's inequality, we have

$$\Pr\left( \left| \mathbb{E}\left[ \hat{V}_{t+1}^*(s_{t+1}) \right] - \sum_{s_{t+1} \in \mathcal{S}} \bar{\mathbb{P}}_a(s_t, s_{t+1}) \hat{V}_{t+1}^*(s_{t+1}) \right| \geq \epsilon \right) \leq 2 \exp\left( \frac{-2m\epsilon^2(1-\gamma)^2}{R_{\max}^2} \right),$$

$$\Pr\left( \left| \mathbb{E}\left[ \tilde{r}_{\text{peer}}(s_t, a) \right] - \sum_{s_{t+1} \in \mathcal{S}} \hat{\mathbb{P}}_a(s_t, s_{t+1}) \tilde{r}_{\text{peer}}(s_t, a) \right| \geq \epsilon \right) \leq 2 \exp\left( \frac{-2m\epsilon^2}{R_{\max}^2} \right).$$

Then the difference between learned value function $V^\pi(s)_t$ and optimal value function $\hat{V}^*(s)_t$ under transformed MDP at iteration $t$ is given:

$$\left| \hat{V}_t^*(s) - V_t(s) \right| = \max_{a \in \mathcal{A}} \mathbb{E}\left[ r_t + \gamma V_{t+1}^*(s_{t+1}) \right] - \max_{a \in \mathcal{A}} \sum_{s_{t+1} \in \mathcal{S}} \bar{\mathbb{P}}_a(s_t, s_{t+1}) \left[ \tilde{r}_{\text{peer}}(s_t, a) + \gamma V_{t+1}(s_{t+1}) \right]$$

$$\leq \max_{a \in \mathcal{A}} \left| \mathbb{E}[r_t] - \sum_{s_{t+1} \in \mathcal{S}} \bar{\mathbb{P}}_a(s_t, s_{t+1}) \tilde{r}_{\text{peer}}(s_t, a) \right|$$

$$+ \gamma \max_{a \in \mathcal{A}} \left| \mathbb{E}\left[ \hat{V}_{t+1}^*(s_{t+1}) \right] - \sum_{s_{t+1} \in \mathcal{S}} \bar{\mathbb{P}}_a(s_t, s_{t+1}) V_{t+1}(s_{t+1}) \right|$$

$$\leq \epsilon_1 + \max_{a \in \mathcal{A}} |\mathbb{E}[r_t] - \mathbb{E}[\tilde{r}_{\text{peer}}]| + \gamma \epsilon_2 + \left| \mathbb{E}\left[ \hat{V}_{t+1}^*(s_{t+1}) \right] - \mathbb{E}[V_{t+1}(s_{t+1})] \right|$$

$$\leq \gamma \max_{s \in \mathcal{S}} \left| \hat{V}_{t+1}^*(s) - V_{t+1}(s) \right| + \epsilon_1 + \gamma \epsilon_2$$

Recursing above equation, we get

$$\max_{s \in \mathcal{S}} \left| \hat{V}^*(s) - V(s) \right| \leq (\epsilon_1 + \gamma \epsilon_2) + \gamma(\epsilon_1 + \gamma \epsilon_2) + \cdots + \gamma^{T-1}(\epsilon_1 + \gamma \epsilon_2)$$

$$= \frac{(\epsilon_1 + \gamma \epsilon_2)(1 - \gamma^T)}{1 - \gamma}$$

Let $\epsilon_1 = \epsilon_2 = \frac{(1-\gamma)\epsilon}{(1+\gamma)}$, then $\max_{s \in \mathcal{S}} \left| \hat{V}^*(s) - V(s) \right| \leq \epsilon$. In other words, for arbitrarily small $\epsilon$, by choosing $m$ appropriately, there always exists $\epsilon_1$ and $\epsilon_2$ such that the value function error is bounded within $\epsilon$. As a consequence the *phased value iteration* algorithm can converge to the near optimal policy within finite steps using peer reward.

Note that there are in total $|\mathcal{S}||\mathcal{A}|T$ transitions under which these conditions must hold, where $|\cdot|$ represent the number of elements in a specific set. Using a union bound, the probability of failure in any condition is smaller than

$$2|\mathcal{S}||\mathcal{A}|T \cdot \exp\left( -m \frac{\epsilon^2(1-\gamma)^2}{(1+\gamma)^2} \cdot \frac{(1-\gamma)^2}{R_{\max}^2} \right).$$

We set above failure probability less than $\delta$, and $m$ should satisfy that

$$m = O\left(\frac{1}{\epsilon^2} \log \frac{|\mathcal{S}||\mathcal{A}|T}{\delta}\right).$$

In consequence, after $m|\mathcal{S}||\mathcal{A}|T$ calls, which is, $O\left(\frac{|\mathcal{S}||\mathcal{A}|T}{\epsilon^2} \log \frac{|\mathcal{S}||\mathcal{A}|T}{\delta}\right)$, the value function converges to the optimal value function $\hat{V}^*(s)$ for every $s$ in transformed MDP $\widetilde{M}$, with probability greater than $1 - \delta$.

From Lemma A1, we know $\hat{V}^*(s) = (1 - e_- - e_+) \cdot V^*(s) + C$, where $C$ is a constant. Let $\epsilon = (1 - e_- - e_+) \cdot \epsilon'$ and $V(s) = (1 - e_- - e_+) \cdot V'(s) + C$, we have

$$|V^*(s) - V'(s)| = \left| \frac{\hat{V}^*(s) - C}{(1 - e_- - e_+)} - \frac{V(s) - C}{(1 - e_- - e_+)} \right| \tag{21}$$

$$= \frac{1}{(1 - e_- - e_+)} \left| \hat{V}^*(s) - V(s) \right| \leq \epsilon' \tag{22}$$

This indicates that when the algorithm converges to the optimal value function for transformed MDP $\hat{\mathcal{M}}$, it also finds a underlying value function $V'(s) = \frac{1}{\eta} V(s)$ that converges the optimal value function $V^*(s)$ for original MDP $\mathcal{M}$.

As a consequence, we know it needs to call $\mathcal{O}\left(\frac{|\mathcal{S}||\mathcal{A}|T}{\epsilon'^2(1-e_--e_+)^2} \log \frac{|\mathcal{S}||\mathcal{A}|T}{\delta}\right)$ to achieve an $\epsilon'$ error in value function for original MDP $\mathcal{M}$, which is no more than $\mathcal{O}\left(\frac{1}{(1-e_--e_+)^2}\right)$ times of the one needed when the RL agent observes true rewards perfectly. When the noise is in high-regime, the algorithm suffers from a large $\frac{1}{(1-e_--e_+)^2}$ thus less efficient. Moreover, the sample complexity of phased value iteration with peer reward is equivalent to the one with surrogate reward in [7] though sampling peer reward is less expensive and does not rely on any knowledge of noise rates. $\qquad \square$

### A.3 Multi-outcome Extension

In this section, we show our peer reward is generalizable to multi-class setting. Recall that in Section 2.2 we suppose the reward is discrete and has $|\mathcal{R}|$ levels, and the noise rates are characterized as $\mathbf{C}_{|\mathcal{R}|\times|\mathcal{R}|}^{\mathrm{RL}}$. Here we make further assumptions on the confusion matrix: the reward is misreported to each level with specific probability, e.g.,

$$\mathbf{C}_{|\mathcal{R}|\times|\mathcal{R}|}^{\mathrm{RL}} = \begin{bmatrix} 1 - \sum_{i\neq 1} e_i, & e_2, & \cdots & e_{|\mathcal{R}|} \\ e_1, & 1 - \sum_{i\neq 2} e_i, & \cdots & e_{|\mathcal{R}|} \\ \vdots & \cdots & \ddots & \vdots \\ e_1, & e_2, & \cdots, & 1 - \sum_{i\neq|\mathcal{R}|} e_i \end{bmatrix} \tag{23}$$

Following the notations in A.1, we define the peer reward in multi-outcome settings as $r(s,a) = \tilde{r}(s,a) - r'$, where $r'$ is randomly sampled following a specific sample policy $\pi_{\mathrm{sample}}$ over all state-action pairs. Let $\widetilde{R}_{\mathrm{peer}}$, $R$, $\widetilde{R}$, and $R'$ denote the random variables corresponding to $\tilde{r}_{\mathrm{peer}}$, $r$, $\tilde{r}$, $r'$, $c_{ij}$ represents the entry of $\mathbf{C}_{|\mathcal{R}|\times|\mathcal{R}|}^{\mathrm{RL}}$. Then we have

$$\mathbb{E}_\pi\left[\widetilde{R}\right] = \sum_{i=1}^{|\mathcal{R}|} \mathbb{P}\left(R = R_i|\pi\right) \sum_{j=1}^{|\mathcal{R}|} c_{ij} R_j$$

$$= \sum_{i=1}^{|\mathcal{R}|} \mathbb{P}\left(R = R_i|\pi\right) \left[ \left(1 - \sum_{j\neq i} e_i\right) R_i + \sum_{j\neq i} e_j R_j \right]$$

$$= \sum_{i=1}^{|\mathcal{R}|} \mathbb{P}\left(R = R_i|\pi\right) \left[ \left(1 - \sum_{j=1}^{|\mathcal{R}|} e_i\right) R_i + \sum_{j=1}^{|\mathcal{R}|} e_j R_j \right]$$

$$= \left(1 - \sum_{j=1}^{|\mathcal{R}|} e_j\right) \mathbb{E}_\pi\left[R\right] + \sum_{j=1}^{|\mathcal{R}|} e_j R_j,$$

and

$$\mathbb{E}_{\pi_{\text{sample}}}\left[\widetilde{R}'\right] = \sum_{i=1}^{|\mathcal{R}|} R_i \cdot \mathbb{P}\left(\widetilde{R} = R_i|\pi_{\text{sample}}\right)$$

$$= \sum_{j=1}^{|\mathcal{R}|} R_j \sum_{i=1}^{|\mathcal{R}|} \mathbb{P}\left(R = R_i|\pi_{\text{sample}}\right) c_{ij}$$

$$= \sum_{j=1}^{|\mathcal{R}|} R_j \left[\sum_{i\neq j} \mathbb{P}\left(R = R_i|\pi_{\text{sample}}\right) e_j + \mathbb{P}\left(R = R_j|\pi_{\text{sample}}\right)\left(1 - \sum_{i\neq j} e_i\right)\right]$$

$$= \sum_{j=1}^{|\mathcal{R}|} R_j \left[\sum_{i=1}^{|\mathcal{R}|} \mathbb{P}\left(R = R_i|\pi_{\text{sample}}\right) e_j + \mathbb{P}\left(R = R_j|\pi_{\text{sample}}\right)\left(1 - \sum_{i=1}^{|\mathcal{R}|} e_i\right)\right]$$

$$= \left(1 - \sum_{i=1}^{|\mathcal{R}|} e_i\right) \mathbb{E}_{\pi_{\text{sample}}}[R] + \sum_{j=1}^{|\mathcal{R}|} e_j R_j.$$

Then, the peer reward is formulated as

$$\mathbb{E}\left[\widetilde{R}_{\text{peer}}\right] = \mathbb{E}_\pi\left[\widetilde{R}\right] - \mathbb{E}\left[\widetilde{R}'\right]$$

$$= \left(1 - \sum_{j=1}^{|\mathcal{R}|} e_j\right) \mathbb{E}_\pi[R] - \left(1 - \sum_{i=1}^{|\mathcal{R}|} e_i\right) \mathbb{E}_{\pi_{\text{sample}}}[R]$$

$$= \left(1 - \sum_{j=1}^{|\mathcal{R}|} e_j\right) \mathbb{E}_\pi[R] + \text{const.}$$

### A.4 Extension in Modern DRL algorithms

In this section, we give the following deep reinforcement learning algorithms combined with our peer reward in Algorithm A3 and A4. In Algorithm A3, we give the peer reward aided robust policy gradient algorithm, where the gradient in Equation 24 corresponds to the loss function $\ell((s,a), q) = q \log \pi_\theta(a|s)$, which is classification calibrated [17]. So the expectation of the gradient in 24 is an unbiased esitmation of the policy gradient in corresponding clean MDP. In (A4), we present a robust DQN algorithm with peer sampling, in which the origin loss is $\ell((s,a), \tilde{y})$, also classification calibrated. Thus the robustness can be proved via [17].

---

**Algorithm A3** Policy Gradient [51] with Peer Reward

---

**Require:** $\widetilde{\mathcal{M}} = (\mathcal{S}, \mathcal{A}, \widetilde{\mathcal{R}}, \mathcal{P}, \gamma)$, learning rate $\alpha \in (0, 1)$, initial state distribution $\beta_0$, weight parameter $\xi$.
1: Initialize $\pi_\theta: \mathcal{S} \times \mathcal{A} \to \mathbb{R}$ arbitrarily
2: **for** $episode = 1$ **to** $M$ **do**
3:     Collect trajectory $\tau_\theta = \{(s_i, a_i, \tilde{r}_i)\}_{i=0}^T$, where $s_0 \sim \beta_0$, $a_t \sim \pi_\theta(\cdot|s_t)$, $s_{t+1} \sim \mathcal{P}(\cdot|s_t, a_t)$.
4:     Compute $q_t = \sum_{i=t}^T \gamma^{t-i} \tilde{r}_i$ for all $t \in \{0, 1, \ldots, T\}$
5:     For each index $i \in \{0, 1, \ldots, T\}$, we independently sample another two different indices $j, k$,
6:     and update policy parameter $\theta$ following

$$\theta \leftarrow \theta + \alpha\left[q_i \nabla_\theta \log \pi_\theta(a_i|s_i) - \xi \cdot q_k \nabla_\theta \log \pi_\theta(a_j|s_j)\right] \tag{24}$$

7: **end for**
**Ensure:** $\pi_\theta$

---

**Algorithm A4** Deep $Q$-Network [49] with Peer Reward

---

**Require:** $\widetilde{\mathcal{M}} = (\mathcal{S}, \mathcal{A}, \widetilde{\mathcal{R}}, \mathcal{P}, \gamma)$, learning rate $\alpha \in (0, 1)$, initial state distribution $\beta_0$, weight parameter $\xi$.
 1: Initialize replay memory $\mathcal{D}$ to capacity $N$
 2: Initialize action-value function $Q$ with random weights
 3: **for** episode $= 1$ **to** $M$ **do**
 4:    **for** $t = 1$ **to** $T$ **do**
 5:       With probability $\epsilon$ select a random action $a_t$, otherwise select $a_t = \max_a Q^*(s, a)$
 6:       Execute action $a_t$ and observe reward $\tilde{r}_t$ and observation $s_{t+1}$
 7:       Store transition $(s_t, a_t, \tilde{r}_t, s_{t+1})$ in $\mathcal{D}$
 8:       Sample three random minibatches of transitions $(s_i, a_i, \tilde{r}_i, s_{i+1}), (s_j, a_j, \tilde{r}_j, s_{j+1}), (s_k, a_k, \tilde{r}_k, s_{k+1})$
          from $\mathcal{D}$.
 9:       Set $\tilde{y}_i = \begin{cases} \tilde{r}_i & \text{for terminal} s_i \\ \tilde{r}_i + \gamma \max_{a'} Q(s_{i+1}, a') & \text{for non-terminal } s_{i+1} \end{cases}$

10:       Set $\tilde{y}_{\text{peer}} = \begin{cases} \tilde{r}_k & \text{for terminal} s_i \\ \tilde{r}_k + \gamma \max_{a'} Q(s_{j+1}, a') & \text{for non-terminal } s_{j+1} \end{cases}$

11:       Perform a gradient descent step on $(\tilde{y}_i - Q(s_i, a_i))^2 - \xi \cdot (\tilde{y}_{\text{peer}} - Q(s_j, a_j))^2$
12:    **end for**
13: **end for**
**Ensure:** $Q$

---

## A.5 Further Discussions on the Effectiveness of PeerRL

We can also analyze why peer rewards are beneficial from the error upper bound. When $\xi = 1$, define the sample mean of rewards as follows.

$$\bar{\tilde{r}} := \frac{1}{T} \sum_{t=1}^{T} \tilde{r}(s_t, a_t), \quad \bar{\tilde{r}}_{\text{peer}} := \frac{1}{T} \sum_{t=1}^{T} \tilde{r}_{\text{peer}}(s_t, a_t) = \frac{1}{T} \sum_{t=1}^{T} [\tilde{r}(s_t, a_t) + (1 - r'_t)] - 1.$$

By Hoeffding's inequality, noting there are $T$ independent random variables in estimating $\bar{\tilde{r}}$ and $2T$ independent random variables in estimating $\bar{\tilde{r}}_{\text{peer}}$, we know w.p. at least $1 - \delta$,

$$|\bar{\tilde{r}} - \mathbb{E}[\tilde{r}]| \leq R_{\max} \sqrt{\frac{\ln 2/\delta}{2T}},$$

and

$$|\bar{\tilde{r}}_{\text{peer}} - \mathbb{E}[\bar{\tilde{r}}_{\text{peer}}]| \leq R_{\max} \sqrt{\frac{\ln 2/\delta}{T}}.$$

We can denote the relationship between reward estimates and the corresponding error rate estimates $\bar{e}_-, \bar{e}_+$ as:

$$\bar{\tilde{r}} = (1 - \bar{e}_- - \bar{e}_+)\bar{r} + \bar{e}_- r_+ + \bar{e}_+ r_-.$$

We have

$$\begin{aligned} |\bar{\tilde{r}} - \mathbb{E}[\tilde{r}]| =& |(1 - \bar{e}_- - \bar{e}_+)\bar{r} - (1 - e_- - e_+)\mathbb{E}[r] + (\bar{e}_- - e_-)r_+ + (\bar{e}_+ - e_+)r_-| \\ =& |(1 - \bar{e}_- - \bar{e}_+)\bar{r} - (1 - \bar{e}_- - \bar{e}_+)\mathbb{E}[r] + (e_- - \bar{e}_- + e_+ - \bar{e}_+)\mathbb{E}[r] + (\bar{e}_- - e_-)r_+ + (\bar{e}_+ - e_+)r_-| \\ \geq& (1 - \bar{e}_- - \bar{e}_+)|\bar{r} - \mathbb{E}[r]| - |\bar{e}_- - e_-|(r_+ + \mathbb{E}[r]) - |\bar{e}_+ - e_+|(r_- + \mathbb{E}[r]). \end{aligned}$$

Thus

$$|\bar{r} - \mathbb{E}[r]| \leq \frac{R_{\max}\sqrt{\frac{\ln 2/\delta}{2T}} + |\bar{e}_- - e_-|(r_+ + \mathbb{E}[r]) + |\bar{e}_+ - e_+|(r_- + \mathbb{E}[r])}{1 - \bar{e}_- - \bar{e}_+}.$$

Assume $\delta_e = |\bar{e}_- - e_-| = |\bar{e}_+ - e_+|$. We have

$$|\bar{r} - \mathbb{E}[r]| \leq \frac{R_{\max}\sqrt{\frac{\ln 2/\delta}{2T}} + \delta_e(r_+ + r_- + 2\mathbb{E}[r])}{1 - \bar{e}_- - \bar{e}_+} \tag{25}$$

Similarly, for peer rewards, note

$$\bar{\tilde{r}}_{\text{peer}} = (1 - \bar{e}_- - \bar{e}_+)(\bar{r} - (1 - \bar{p}_{\text{peer}})r_- - \bar{p}_{\text{peer}}r_+).$$

We have

$$
\begin{aligned}
|\bar{\tilde{r}}_{\text{peer}} - \mathbb{E}[\tilde{r}_{\text{peer}}]| =& |(1 - \bar{e}_- - \bar{e}_+)(\bar{r} - (1 - \bar{p}_{\text{peer}})r_- - \bar{p}_{\text{peer}}r_+) - (1 - \bar{e}_- - \bar{e}_+)(\mathbb{E}[r] - (1 - p_{\text{peer}})r_- - p_{\text{peer}}r_+) \\
& + (e_- - \bar{e}_- + e_+ - \bar{e}_+)(\mathbb{E}[r] - (1 - p_{\text{peer}})r_- - p_{\text{peer}}r_+)| \\
\geq & (1 - \bar{e}_- - \bar{e}_+)|\bar{r} - \mathbb{E}[r]| - (1 - \bar{e}_- - \bar{e}_+)|\bar{p}_{\text{peer}} - p_{\text{peer}}||r_- - r_+| - \\
& |\bar{e}_- - e_-| \cdot |\mathbb{E}[r] - (1 - p_{\text{peer}})r_- - p_{\text{peer}}r_+| - |\bar{e}_+ - e_+| \cdot |\mathbb{E}[r] - (1 - p_{\text{peer}})r_- - p_{\text{peer}}r_+|.
\end{aligned}
$$

Thus

$$
|\bar{r} - \mathbb{E}[r]| \leq \frac{R_{\max}\sqrt{\frac{\ln 2/\delta}{T}} + (|\bar{e}_+ - e_+| + |\bar{e}_- - e_-|) \cdot |\mathbb{E}[r] - (1 - p_{\text{peer}})r_- - p_{\text{peer}}r_+|}{1 - \bar{e}_- - \bar{e}_+} + |\bar{p}_{\text{peer}} - p_{\text{peer}}||r_- - r_+|.
$$

Assume $\delta_e = |\bar{e}_- - e_-| = |\bar{e}_+ - e_+| = |\bar{p}_{\text{peer}} - p_{\text{peer}}|$. We have

$$
|\bar{r} - \mathbb{E}[r]| \leq \frac{R_{\max}\sqrt{\frac{\ln 2/\delta}{T}} + 2\delta_e \cdot |\mathbb{E}[r] - (1 - p_{\text{peer}})r_- - p_{\text{peer}}r_+|}{1 - \bar{e}_- - \bar{e}_+} + \delta_e|r_- - r_+|. \tag{26}
$$

Comparing Eqn. (25) and Eqn. (26), for the high-noise case, we can infer peer rewards likely have lower sample complexity, i.e. is more sample efficient. For example, when $p_{\text{peer}} = 0.5$, $e_- = e_+ = 0.3$, $R_{\max} = 1$, $r_+ = 1$, $r_- = 0$, $\bar{e}_- - e_- = \bar{e}_+ - e_+ = \delta_e$, $\mathbb{E}[r] = 0.5$, we have

$$
|\bar{r} - \mathbb{E}[r]| \leq \frac{\sqrt{\frac{\ln 2/\delta}{2T}} + 2\delta_e}{0.4} \quad \text{(Plain Reward)},
$$

$$
|\bar{r} - \mathbb{E}[r]| \leq \frac{\sqrt{\frac{\ln 2/\delta}{T}} + 1.4\delta_e}{0.4} \quad \text{(Peer Reward)}.
$$

In this case, we know peer rewards have a lower error upper bound for estimating $r$ when $T$ is large.

## B Tie-Breaking: Toy Examples

To illustrate *tie-breaking* phenomenon when using peer reward, we consider a two-state Markov process (no actions) with varied noise models. An example code segment with stochastic rewards and discrete noise model ($e_- = e_+ = 0.45$) is provided below:

```python
def get_rewards(state, num_samples, noise_rate=0.45):
    if state == 0: r = np.random.choice([0, 1], p=[0.4, 0.6], size=num_samples)  # E[r] = 0.6
    else: r = np.random.choice([0, 1], p=[0.6, 0.4], size=num_samples)  # E[r] = 0.4
    mask = np.random.choice(2, p=(1 - noise_rate, noise_rate), size=num_samples)  # Add noise
    r = (1 - mask) * r + mask * (1 - r)
    return r

num_samples, xi = 1000, 0.1
is_correct_noisy, is_correct_peer = []. []
for _ in tqdm.trange(10000):
    # Baseline
    r_vec = np.stack([get_rewards(0, num_samples), get_rewards(1, num_samples)], axis=1)
    r_hat = np.mean(r_vec, axis=0)
    is_correct_noisy.append(r_hat[0] > r_hat[1])
    # PeerRL
    neg_samples = np.concatenate([get_rewards(0, num_samples), get_rewards(1, num_samples)])
    np.random.shuffle(neg_samples)  # Randomly permutes the elements
    neg_samples0, neg_samples1 = np.split(neg_samples, 2)
    r_vec = np.stack([get_rewards(0, num_samples) - xi * neg_samples0,
                     get_rewards(1, num_samples) - xi * neg_samples1], axis=1)
    r_hat = np.mean(r_vec, axis=0)
```

```
22        is_correct_peer.append(r_hat[0] > r_hat[1])
23
24    print('\nBaseline Success: %.3f\n' % np.mean(is_correct_noisy))
25    print('\nPeer RL Success: %.3f\n' % np.mean(is_correct_peer))
```

In Table A1, we conducted more experiments with different noise models and reported the absolute accuracy differences between PeerRL and baseline (noisy reward) in the following three cases: (1) "Correct" - successfully inferring the better state $s_1$ with larger expected reward, (2) "Tie" - cannot infer which state is better as the means of collected rewards in two states are equal, (3) "Incorrect" - wrongly inferring state $s_2$ is better ("Incorrect"). As we can see, PeerRL exploits the "discreteness" of the reward thus breaking ties to obtain more examples with good-quality supervision. This *tie breaking* phenomenon also happens for stochastic reward and bounded/discretized continuous reward.

Table A1: Tie breaking toy examples under varied noise models.

| Bounded continuous noise | Correct | Tie | Incorrect |
|---|---|---|---|
| $s_1$ : r = np.clip(np.random.normal(0.6,1.0, num_samples),0,1) 
 $s_2$ : r = np.clip(np.random.normal(0.4,1.0, num_samples),0,1) | **+3.4%** | **-5.3%** | **+1.9%** |
| $s_1$ : r = np.clip(np.random.laplace(0.6,1.0, num_samples),0,1) 
 $s_2$ : r = np.clip(np.random.laplace(0.4,1.0, num_samples),0,1) | **+2.0%** | **-4.8%** | **+2.8%** |
| Discretized continuous noise | Correct | Tie | Incorrect |
| $s_1$ = r = np.random.normal(0.6, 1.0, num_samples) 
 $s_2$ = r = np.random.normal(0.4, 1.0, num_samples) 
 bins = np.arange(0, 1.01, 0.01), inds = np.digitize(r, bins) 
 r = bins[inds - 1] | **+6.2%** | **-12.6%** | **+6.4%** |
| Stochastic reward with discrete noise | Correct | Tie | Incorrect |
| $s_1$ : r = np.random.choice([0, 1], p=[0.6, 0.4], size=num_samples) 
 $s_2$ : r = np.random.choice([0, 1], p=[0.4, 0.6], size=num_samples) 
 e = 0.4, mask = np.random.choice(2, p=(1 - e, e), size=num_samples) 
 r = (1 - mask) * r + mask * (1 - r) | **+11.7%** | **-23.1%** | **+11.4%** |
| $s_1$ : r = np.random.poisson(0.6, 1.0, num_samples) 
 $s_2$ : r = np.random.poisson(0.4, 1.0, num_samples) | **+10.2%** | **-20.8%** | **+10.6%** |
| Deterministic reward with discrete noise | Correct | Tie | Incorrect |
| $s_1$ : r = np.random.choice([0, 1], p=[0.6, 0.4], size=num_samples) 
 $s_2$ : r = np.random.choice([0, 1], p=[0.4, 0.6], size=num_samples) 
 e = 0.4, mask = np.random.choice(2, p=(1 - e, e), size=num_samples) 
 r = (1 - mask) * r + mask * (1 - r) | **+10.5%** | **-21.2%** | **+10.7%** |
| Continuous noise | Correct | Tie | Incorrect |
| $s_1$ : r = np.clip(np.random.normal(0.6,1.0, num_samples),0,1) 
 $s_2$ : r = np.random.normal(0.4,1.0, num_samples) | **+0.0%** | **-0.0%** | **+0.0%** |
| $s_1$ : r = np.random.laplace(0.6,1.0, num_samples) 
 $s_2$ : r = np.clip(np.random.laplace(0.4,1.0, num_samples),0,1) | **+0.0%** | **-0.0%** | **+0.0%** |

## C   Analysis of PeerBC

We prove that the policy learned by PeerBC converges to the expert policy when observing a sufficient amount of weak demonstrations in Theorem A3.

**Theorem A3.** *With probability at least $1 - \delta$, the error rate is upper-bounded by*

$$R^*_{D_E} \leq \frac{1 + \xi}{1 - e_- - e_+} \sqrt{\frac{2 \log 2/\delta}{N}}, \tag{27}$$

*where $N$ is the number of state-action pairs demonstrated by the expert.*

*Proof.* Recall $\widetilde{\mathcal{D}}_E$ denotes the joint distribution of imperfect expert' state-action pair $(s, \tilde{a})$. Assume there is a perfect expert and the corresponding state-action pairs $(s, a) \sim \mathcal{D}_E$. The indicator classification loss $\mathbb{1}(\pi(s), a)$ is specified here for a clean presentation, where $\mathbb{1}(\pi(s), a) = 1$ when

$\pi(s) \neq a$, otherwise $\mathbb{1}(\pi(s), a) = 0$. Let $\widetilde{D}_E := \{(s_i, \tilde{a}_i)\}_{i=1}^N$ be the set of imperfect demonstrations, and $D_E := \{(s_i, \tilde{a}_i)\}_{i=1}^N$ be the set of weak demonstrations. Define:

$$R_{\mathcal{D}_E}(\pi) := \mathbb{E}_{(s,a) \sim \mathcal{D}_E}\left[\mathbb{1}(\pi(s), a)\right], \ R_{\widetilde{\mathcal{D}}_E}(\pi) := \mathbb{E}_{(s,\tilde{a}) \sim \mathcal{D}_E}\left[\mathbb{1}(\pi(s), \tilde{a})\right]$$

$$\hat{R}_{D_E}(\pi) := \frac{1}{N}\sum_{i \in [N]} \mathbb{1}(\pi(s_i), a_i), \ \hat{R}_{\widetilde{D}_E}(\pi) := \frac{1}{N}\sum_{i \in [N]} \mathbb{1}(\pi(s_i), \tilde{a}_i).$$

Note we focus on the analyses of loss in this proof. The negative of loss can be seen as a reward. Denote by $\pi_{\widetilde{D}_E}$ and $\pi_{\widetilde{\mathcal{D}}_E}$ be the optimal policy obtained with minimizing the indicator loss with dataset $\widetilde{D}_E$ and distribution $\widetilde{\mathcal{D}}_E$. We shorten $\pi_{\widetilde{D}_E}$ as $\tilde{\pi}^*$, which is the best policy we can learn from imperfect demonstration with our algorithm. Let $\pi^*$ be the policy for the perfect expert. We would like to see the performance gap of policy learning between imperfect demonstrations and perfect demonstrations, i.e. $R_{\mathcal{D}_E}(\tilde{\pi}^*) - R_{\mathcal{D}_E}(\pi^*)$. Using Hoeffding's inequality with probability at least $1 - \delta$, we have

$$|\hat{R}_{\widetilde{D}_E}(\pi) - R_{\widetilde{\mathcal{D}}_E}(\pi)| \leq (1 + \xi)\sqrt{\frac{\log 2/\delta}{2N}}.$$

Note we also have

$$R_{\widetilde{\mathcal{D}}_E}(\tilde{\pi}^*) - R_{\widetilde{\mathcal{D}}_E}(\pi_{\widetilde{\mathcal{D}}_E})$$

$$\leq \hat{R}_{\widetilde{D}_E}(\tilde{\pi}^*) - \hat{R}_{\widetilde{D}_E}(\pi_{\widetilde{\mathcal{D}}_E}) + \left(R_{\widetilde{\mathcal{D}}_E}(\tilde{\pi}^*) - \hat{R}_{\widetilde{D}_E}(\tilde{\pi}^*)\right)$$

$$+ \left(\hat{R}_{\widetilde{D}_E}(\pi_{\widetilde{\mathcal{D}}_E}) - R_{\widetilde{\mathcal{D}}_E}(\pi_{\widetilde{\mathcal{D}}_E})\right)$$

$$\leq 0 + 2\max_{\pi}\left|\hat{R}_{\widetilde{D}_E}(\pi) - R_{\widetilde{\mathcal{D}}_E}(\pi)\right|$$

$$\leq (1 + \xi)\sqrt{\frac{2\log 2/\delta}{N}}.$$

Before proceeding, we need to define a constant to show the affect of label noise. When the dimension of action space is 2, the problem is essentially a binary classification with noisy labels [17], where the noise rate (a.k.a confusion matrix) is defined as $e_+ = \mathbb{P}(\tilde{\pi}_E(s) = A_-|\pi^*(s) = A_+)$ and $e_- = \mathbb{P}(\tilde{\pi}_E(s) = A_+|\pi^*(s) = A_-)$. Recall the action space is defined as $\mathcal{A} = \{A_+, A_-\}$. The noise constant is denoted by $e = e_{-1} + e_{+1}$. Accordingly, when the dimension of action space is $|\mathcal{R}| > 2$, we can also get similar results under uniform noise where

$$e_u := \mathbb{P}(\tilde{\pi}_E(s) = u|\pi^*(s) = u'), u' \neq u. \tag{28}$$

The noise constant $e$ is denoted by $e = \sum_{u=1}^{|\mathcal{R}|} e_u$. The feature-independent assumption holds thus the properties of peer loss functions [17] can be used, i.e.

$$R_{\mathcal{D}_E}(\tilde{\pi}^*) - R_{\mathcal{D}_E}(\pi^*)$$

$$= \frac{1}{1 - e}\left(R_{\widetilde{\mathcal{D}}_E}(\tilde{\pi}^*) - R_{\widetilde{\mathcal{D}}_E}(\pi_{\widetilde{\mathcal{D}}_E})\right)$$

$$\leq \frac{1 + \xi}{1 - e}\sqrt{\frac{2\log 2/\delta}{N}}$$

From definition and deterministic assumption for $\pi^*$, we have $R_{\mathcal{D}_E}(\pi^*) = 0$. Thus the error rate in the $k$-th iteration is

$$R_{\mathcal{D}_E}(\tilde{\pi}^*) \leq R_{\mathcal{D}_E}(\pi^*) + \frac{1 + \xi}{1 - e}\sqrt{\frac{2\log 2/\delta}{N}}$$

$$= \frac{1 + \xi}{1 - e}\sqrt{\frac{2\log 2/\delta}{N}}. \tag{29}$$

Note $R_{\mathcal{D}_E}(\tilde{\pi}^*) = R_{\widetilde{D}_E}$ by definition. $\qquad\square$

# D   Supplementary Experiments

## D.1   Experimental Setup

We set up our experiments within the popular OpenAI `stable-baselines`[2] and `keras-rl`[3] framework. Specifically, three popular RL algorithms including Deep-$Q$-Network (DQN) [49, 56], Dueling-DQN (DDQN) [50] and Proximal Policy Optimization Algorithms (PPO) are evaluated in a varied of OpenAI Gym environments including classic control games (`CartPole`, `Acrobot`) and vision-based Atari-2600 games (`Breakout`, `Boxing`, `Enduro`, `Freeway`, `Pong`).

## D.2   Implementation Details

**RL with noisy reward**   Following [7], we consider the binary reward $\{-1, 1\}$ for Cartpole where the symmetric noise is synthesized with different error rates $e = e_- = e_+$. We adopted a five-layer fully connected network and the Adam optimizer. The model is trained for 10,000 steps with the learning rate of $1e^{-3}$ and the Boltzmann exploration strategy. The update rate of target model and the memory size are $1e^{-2}$ and 50,000. The performance is reported under 10 independent trials with different random seeds.

**BC with weak expert**   We train the imperfect expert on the framework `stable-baselines` with default network architecture for Atari and hyper-parameters from `rl-baselines-zoo`[4]. The expert model is trained for $1,400,000$ steps for Pong and $2,000,000$ steps for Boxing, Enduro and Freeway. For each of those environment, We use the trained model to generate 100 trajectories, and behavior cloning is performed on these trajectories. We adopt cross entropy loss for behavior cloning and add a small constant ($1 \times 10^{-8}$) for each logit after the softmax operation for peer term to avoid this term become too large. In BC experiments, the batchsize is 128, learning rate is $1 \times 10^{-4}$ and the $\epsilon$ value for Adam optimizer is $1 \times 10^{-8}$.

**Policy co-training**   For the experiments on Gym (CartPole and Acrobot), we mask the first coordinate in the state vector for one view and the second for the other, same as [43]. Both policies are trained with PPO[58] + PeerBC. In each iteration, we sample 128 steps from each of the 8 parallel environments. These samples are fed to PPO training with a batchsize of 256, a learning rate of $2.5 \times 10^{-4}$ and a clip range of 0.1. Both learning rate and clip range decay to 0 throughout time. We represent the policy by a fully connected network with 2 hidden layers, each has 128 units.

For the experiments on Atari (Pong and Breakout), the input is raw game images. We adopt the preprocess introduced in [49] and mask the pixels in odd columns for one view and even columns for the other. The policy we use adopts a default CNN as in `stable-baselines`. Batchsize, learning rate, clip range and other hyper-parameters are the same as Gym experiments. Note that we only add PeerBC after 1000 episodes.

## D.3   Supplementary Results for Figure 2 and Table 1

Table A2: Numerical performance of DDQN on CartPole with true reward ($r$), noisy reward ($\tilde{r}$), surrogate reward $\hat{r}$ [7], and peer reward $\tilde{r}_{\mathrm{peer}}(\xi = 0.2)$. $\mathcal{R}_{avg}$ denotes average reward per episode after convergence, (last five episodes) the higher ($\uparrow$) the better; $N_{epi}$ denotes total episodes involved in 10,000 steps, the lower ($\downarrow$) the better.

| | | $e = 0.1$ | | $e = 0.2$ | | $e = 0.3$ | | $e = 0.4$ | |
| --- | --- | --- | --- | --- | --- | --- | --- | --- | --- |
| | | $\mathcal{R}_{avg} \uparrow$ | $N_{epi} \downarrow$ | $\mathcal{R}_{avg} \uparrow$ | $N_{epi} \downarrow$ | $\mathcal{R}_{avg} \uparrow$ | $N_{epi} \downarrow$ | $\mathcal{R}_{avg} \uparrow$ | $N_{epi} \downarrow$ |
| DQN | $r$ | $183.6 \pm 7.6$ | $101.3 \pm 4.8$ | $184.0 \pm 7.3$ | $101.5 \pm 4.6$ | $184.0 \pm 7.3$ | $101.5 \pm 4.6$ | $184.0 \pm 7.3$ | $101.5 \pm 4.6$ |
| | $\tilde{r}$ | $\mathbf{189.3 \pm 12.7}$ | $98.2 \pm 6.5$ | $189.7 \pm 7.9$ | $110.5 \pm 7.1$ | $183.2 \pm 9.8$ | $130.5 \pm 7.7$ | $169.7 \pm 18.6$ | $150.2 \pm 11.4$ |
| | $\hat{r}$ | $188.3 \pm 8.2$ | $101.1 \pm 6.2$ | $\mathbf{192.7 \pm 9.2}$ | $97.9 \pm 6.4$ | $185.4 \pm 15.9$ | $116.9 \pm 11.0$ | $\mathbf{184.8 \pm 16.4}$ | $123.1 \pm 8.6$ |
| | $\tilde{r}_{\mathrm{peer}}$ | $177.2 \pm 19.1$ | $\mathbf{91.2 \pm 5.9}$ | $170.0 \pm 24.8$ | $\mathbf{94.6 \pm 8.5}$ | $\mathbf{190.5 \pm 14.3}$ | $\mathbf{99.4 \pm 5.2}$ | $183.1 \pm 13.3$ | $\mathbf{118.1 \pm 10.7}$ |
| DDQN | $r$ | $195.6 \pm 3.1$ | $101.2 \pm 3.2$ | $195.6 \pm 3.1$ | $101.2 \pm 3.2$ | $195.6 \pm 3.1$ | $101.2 \pm 3.2$ | $195.2 \pm 3.0$ | $101.2 \pm 3.3$ |
| | $\tilde{r}$ | $185.2 \pm 15.6$ | $114.6 \pm 6.0$ | $168.8 \pm 13.6$ | $123.9 \pm 9.6$ | $177.1 \pm 11.2$ | $133.2 \pm 9.1$ | $185.5 \pm 10.9$ | $163.1 \pm 11.0$ |
| | $\hat{r}$ | $183.9 \pm 10.4$ | $110.6 \pm 6.7$ | $165.1 \pm 18.2$ | $113.9 \pm 9.6$ | $\mathbf{192.2 \pm 10.9}$ | $115.5 \pm 4.3$ | $179.2 \pm 6.6$ | $125.8 \pm 9.6$ |
| | $\tilde{r}_{\mathrm{peer}}$ | $\mathbf{198.5 \pm 2.3}$ | $\mathbf{86.2 \pm 5.0}$ | $\mathbf{195.5 \pm 9.1}$ | $\mathbf{85.3 \pm 5.4}$ | $174.1 \pm 32.5$ | $\mathbf{88.8 \pm 6.3}$ | $\mathbf{191.8 \pm 8.5}$ | $\mathbf{106.9 \pm 9.2}$ |

[2] https://github.com/hill-a/stable-baselines
[3] https://github.com/keras-rl/keras-rl
[4] https://github.com/araffin/rl-baselines-zoo/blob/master/hyperparams/ppo2.yml#L1

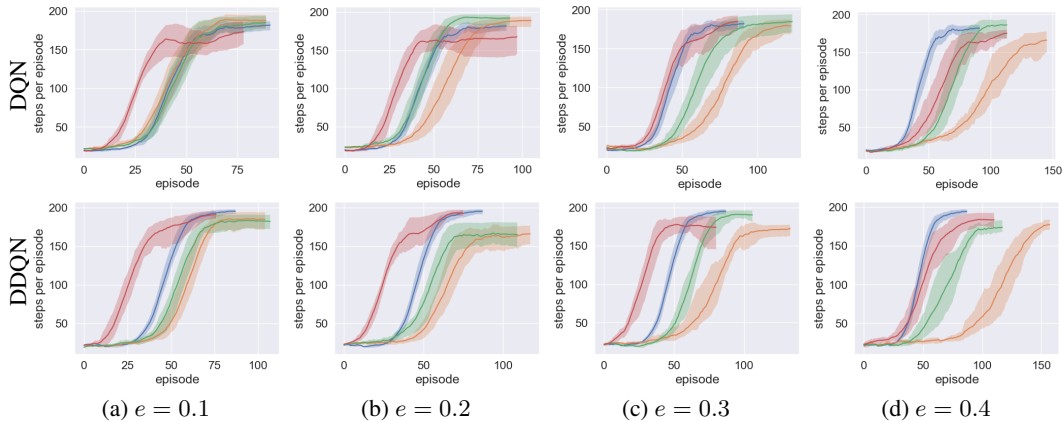

Figure A1: Learning curves on CartPole game with true reward ($r$) ■, noisy reward ($\tilde{r}$) ■, surrogate reward [7] ($\hat{r}$) ■, and peer reward ($\tilde{r}_{\text{peer}}$, $\xi = 0.2$) ■. Each experiment is repeated 10 times with different random seeds.

## D.4 Sensitivity Analysis of Peer Penalty $\xi$

In this section, we analyze the sensitivity of $\xi$ in RL and BC tasks. Note that we did not tune this hyperparameter extensively in all the experiments presented above since we found our method works robustly in a wide range of $\xi$.

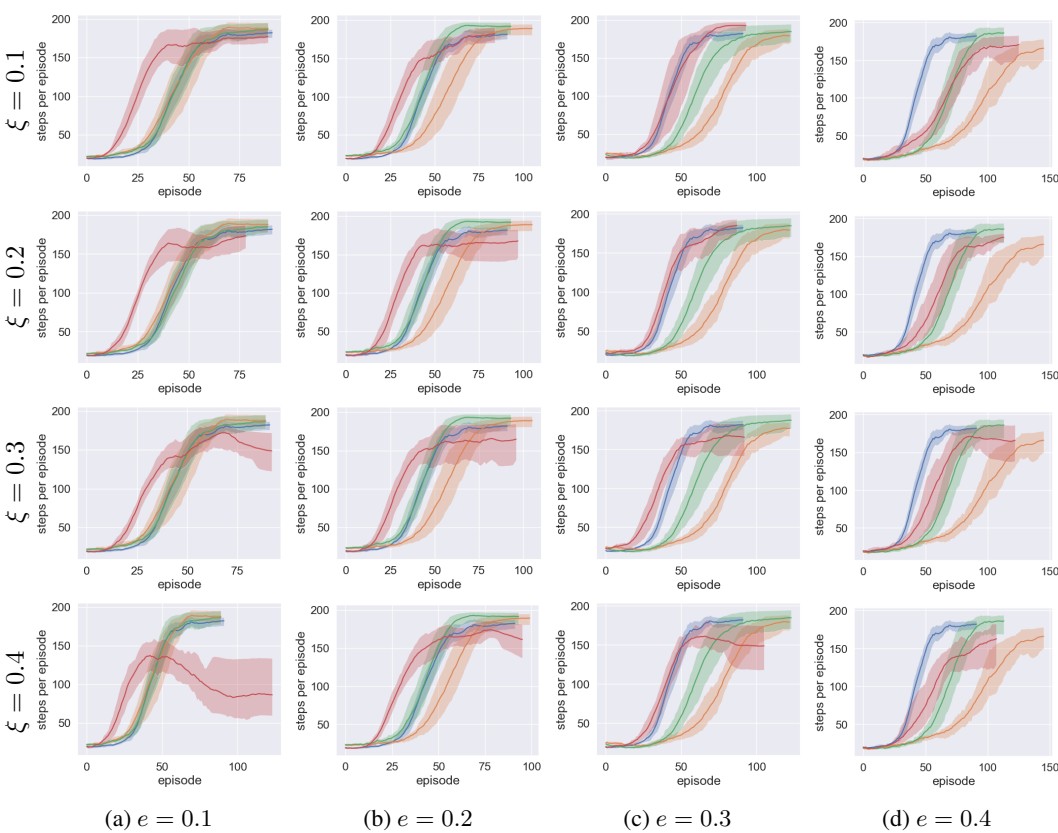

Figure A2: Learning curves of DQN on CartPole game with peer reward ($\tilde{r}_{\text{peer}}$) ■ under different choices of $\xi$ (from 0.1 to 0.4).

**RL with noisy reward** We repeat the experiment in Figure A1 for DQN but with a varying $\xi$ from 0.1 to 0.4. As shown in Figure A2, our method works reasonably and leads to faster convergence

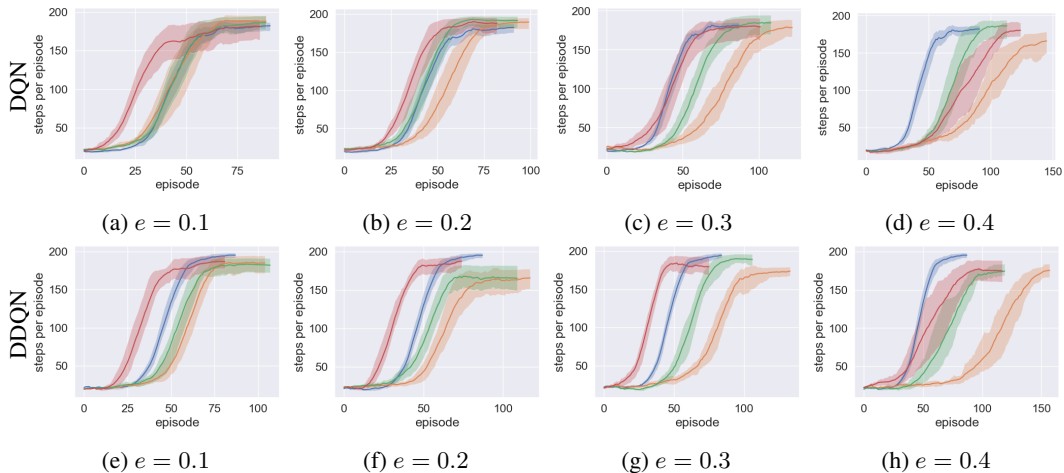

Figure A3: Learning curves of DQN on CartPole game with peer rewards ($\tilde{r}_{\text{peer}}$) ■. Here, a linear decay $\xi$ is applied during training procedure (initial $\xi = 0.4$). Compared to static $\xi = 0.4$, the linear decay peer penalty stabilizes the convergence of RL algorithms.

compared to baselines. However, we found that the late stage of training, a small $\xi$ is necessary since the agent already gains useful knowledge and make reasonable actions, therefore, an over-large penalty might avoid the agent achieving simple agreements with the supervision signals, especially in a low-noise regime (see $\xi = 0.4, e = 0.1$). This observation inspires us that a decay schedule of $\xi$ might be helpful in stabilizing the training of PeerRL algorithms. To verify this hypothesis, we repeat the above experiments but with a linear decay $\xi$ that decreases from 0.4 to 0.1. In Figure A3, we found the linear decay schedule is able to stabilize the convergence of PeerRL algorithms compared to static $\xi = 0.4$. The theoretical principles and insights of dynamic peer penalty merit further study.

**BC from weak demonstrations** We conduct experiments on Pong with 12 different $\xi$ values, varying from 0.1 to 1.2. From Figure A4, we can see PeerBC outperforms pure behavior cloning and SQIL[37] when $\xi$ is within $[0.1, 0.7]$, revealing our proposed PeerBC is a superior behavior cloning approach able to better elicit information from imperfect demonstrations.

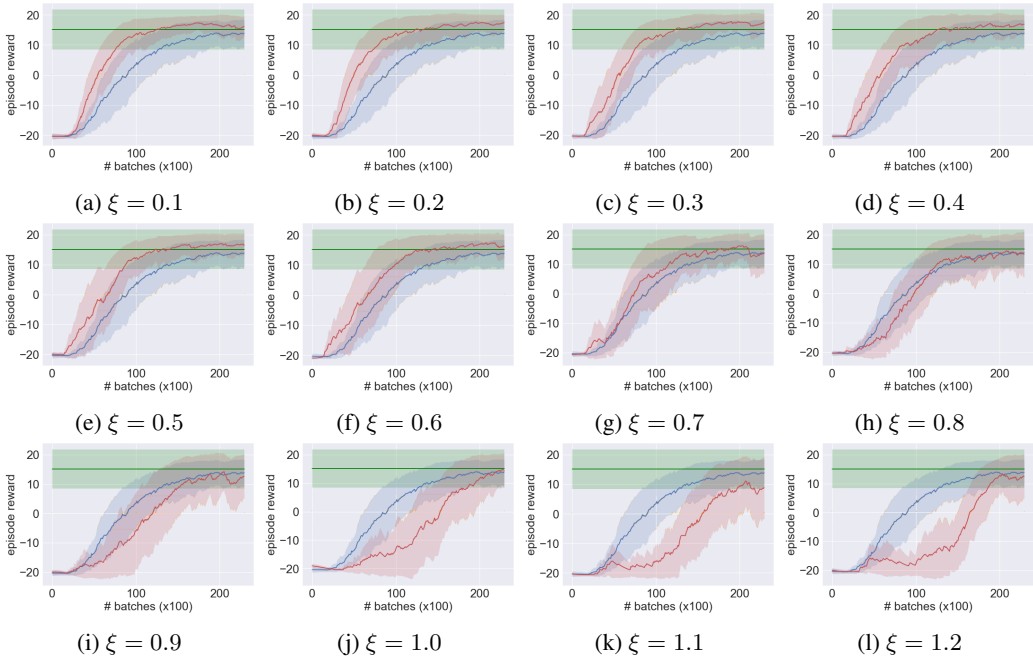

(a) ξ = 0.1      (b) ξ = 0.2      (c) ξ = 0.3      (d) ξ = 0.4

(e) ξ = 0.5      (f) ξ = 0.6      (g) ξ = 0.7      (h) ξ = 0.8

(i) ξ = 0.9      (j) ξ = 1.0      (k) ξ = 1.1      (l) ξ = 1.2

Figure A4: Sensitivity analysis of $\xi$ for PeerBC on Pong with behavior cloning ■, PeerBC ■ ($\xi$ varies from $0.2$ to $0.5$ and $1.0$) and expert ■. Each experiment is repeated under 3 different random seeds.

## D.5 Stochastic Policy for Behavioral Cloning

In this section, we analyze the stochasticity of the imperfect expert model and fully-converged PPO agent (assumed to be the clean expert), and show that our PeerBC can handle both cases when the clean expert is stochastic and when it's rather deterministic.

Figure A5: The policy entropy of the PPO agent during training. The imperfect expert model is trained for $0.2 \times 10^7$ timesteps as the red line indicates.

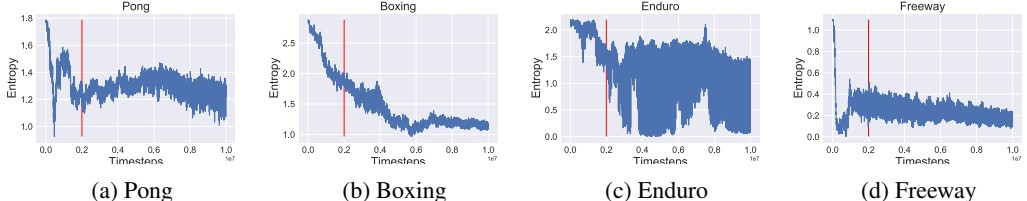

(a) Pong      (b) Boxing      (c) Enduro      (d) Freeway

Table A3: The policy entropy of the PPO agent during training.

| Timesteps ($\times 10^7$) | Pong | Boxing | Enduro | Freeway |
|---|---|---|---|---|
| 0.2 (Imperfect Expert) | 1.201 | 1.949 | 1.637 | 0.318 |
| 1.0 (Fully converged PPO) | 1.250 | 1.168 | 1.126 | 0.171 |

Table A4: The mean value of the highest action probability over 1000 steps.

| Trained timesteps ($\times 10^7$) | Pong | Boxing | Enduro | Freeway |
|---|---|---|---|---|
| 1.0 (Fully converged PPO) | 0.492 | 0.579 | 0.664 | 0.903 |

We plot the entropy of the PPO agent during training on four environments from the BC task in Figure A5, and we give the entropy value of the imperfect expert model and the optimal policy in Table A3. We observe that except for Freeway, the entropy of expert policies is always larger than 1. We calculate the mean value of the highest action probability over 1000 steps for the full-converged PPO agents in Table A4, which again verifies that the true expert policy we aim to recover might not be fully deterministic. These results demonstrate the flexibility of our proposed approach in dealing with both stochastic and deterministic clean expert policies in practice, although a deterministic clean expert policy is assumed in our theoretical analysis.

Also, from Figure A5 and Table A3, we notice that the entropy of imperfect expert models are higher than the fully converged PPO agents, implying that the expert models might contain an amount of noise. That's because there might be states on which the expert has not seen enough and the selected actions contain much noise. This is consistent with our claim, that the benefits of PeerBC might come from two aspects, both noise reduction of the imperfect expert and inducing a more deterministic policy.