# OpenReview forum: "Policy Learning Using Weak Supervision"
_NeurIPS.cc/2021/Conference — NeurIPS 2021 Poster_

### Official Review · Reviewer_58fG · 2021-06-30

**Rating:** 7
**Confidence:** 4

**Summary:**

This paper proposes a method for RL and imitation learning in the setting where the reward function or expert actions are corrupted with noise. The main idea is to regularize the learned policy to have *high* loss on unaligned state-action-reward examples. Experiments show that the proposed method outperforms baselines that either learn directly using the noisy rewards/actions or learn an auxiliary function that predicts the mean reward/action.


**Limitations And Societal Impact:**

The concluding paragraph does a great job discussing limitations and areas for future work. I did not see any discussion of societal impacts, either positive or negative.


**Main Review:**

**Originality**:
The proposed method is (to the best of my knowledge) novel. The paper does good job describing the differences with the most related prior work [Which learns to predict a non-noisy version of the rewards/actions, and then uses them for learning]. My understanding is that the key benefit of the the proposed approach is that it doesn't require knowledge about the noise distribution and doesn't require a separate estimator of the mean reward/action.

**Quality**:
* **Theory**: The main technical claims about sample complexity seem correct (I quickly read through the proofs in the Appendix), building directly on top of prior work. My main concern about the technical claims is that they don't seem to motivate the proposed method. That is, it's unclear if the Theorems show that the proposed method is more sample efficient than either (1) directly learning using noisy rewards/actions or (2) learning using the empirical mean estimates for the rewards/actions. I would highly encourage the authors to discuss this. Said in other words, while it's clear why the proposed method should have the same solution as learning directly on the noisy examples, it's unclear why injecting more noise (but adding a random term to the loss function) should result in faster learning.
* **Experiments**: The paper did a great job comparing both the RL setting and the imitation learning setting, comparing against the most similar prior work. The experiments in the RL setting are limited to two very simple tasks. I'd be curious to see how the method scales to more complex tasks. One suggestion: I would highly recommend including the baseline results on the Pendulum task; omitting baseline results on the tasks where the baseline performs as well as the proposed method is concerning. The imitation learning results on Atari are also nice. I found the co-training experiments quite hard to follow. The discussion of the benefits of PeerRL (L292) seems intriguing, but I'm not sure if I believe the three offered explanations. For example, the argument about human rewards being imperfect seems to contradict the claim that the proposed method optimizes whatever reward function is given. Similarly, the claim that the proposed method decreases variance seems suspicious because (a) it seems like it actually *increases* the variance of the gradient estimator and (b) this same logic would suggest that the baseline from [7] should outperform the proposed method.

**Clarity**:
The paper is generally well written and reads quite well. The related work section is well organized and discusses relevant prior work. The paper does a great job of introducing and explaining notation before using it. See list of minor writing comments at the end of the review. One high-level suggestion is to spend more time explaining the problem settings. Since the problem setting considered in this paper is different from most other RL papers, it's be good to explain why it is useful and interesting. If possible, I'd highly recommend adding an experiment that motivates this setting.

**Significance**:
I am on the fence about the significance of the paper. On the one hand, the proposed method is quite simple and performs well in a variety of experiments. On the other hand, it's unclear if the analysis actually explains *why* the proposed method is performing better (see previous comment), and it's unclear whether the problem being solved is practically relevant.


**Minor comments**:
* L16 "breakthrough" -> "breakthroughs"
* L16 "these techniques" -- Which techniques?
* L19 "expensive to obtain in practice" -- Add a citation.
* L20 "thus" -- Grammar error.
* L21 "are often imperfect due to limited resources or environment noise" -- I'm unsure if this claim is actually correct. Noise in the environment does not affect the distribution $p(a | s)$, which is all we need for behavior cloning.
* L22 - L29 -- I found this paragraph confusing because it was unclear what "weak supervision" was referring to. I'd highly recommend providing *real-world* examples of this problem setting.
* L26 "imperfect solution" -> "solution"
* L28 "Then as a response, we..." -> "We ..."
* L30 - L31 -- I found this sentence confusing.
* L38 "weakness of the supervision" -- Unclear what this means.
* Generally try to avoid adjectives and adverbs (e.g., "intensive analysis" and "jointly demonstrate" and "mainly three-folds").
* At the end of the introduction, I was still unsure what problem was being solved.
* L57 "in-efficient" -> "inefficient"
* L58 "an RL algorithm" -- Add a citation.
* L62 "reasoning" -> "reasoning about"
* L81 "unify" -- Unify with what?
* L104 "both worlds" -- Unclear what "worlds" refers to here.
* L107 - L111 -- I found these few sentences hard to follow.
* L114 - L127 -- These definitions are great, and very helpful!
* L128 -- From reading this paragraph, it was hard to tell whether $C$ was known or not known.
* L132 "taken policy" -- Unclear what this means.
* L136 - L137 -- I found the definition of Eva() pretty confusing. I might recommend explaining the behavior cloning case first without introducing Eva(), and later describing how these results can be extended to the RL setting.
* L137 $J(\pi)$ -- I found it confusing that the same notation, $J(\pi)$, was defined differently in three different places: L92, L98, L137.
* L146 "correlated agreement" -- No need for the quotes.
* L148 "instances" -- Unclear what an "instance" is.
* L151 - L156 -- This intuition is pretty clever!
* L157 - L164 -- This example is somewhat hard to follow. Consider explaining it with less notation, perhaps with pictures.
* L165 "consolidate our ..." -- Unclear what this means.
* L174 "with agreement check" -- Unclear what this means.
* Lemma 1 -- This result is fairly elementary and could be moved to the Appendix.
* L196 - L199 -- I found these sentences a bit confusing.
* L207 "we found that" -- Unclear if this is a theoretical or empirical result.
* Figures -- Please make the figures colorblind-friendly.
* L276 - L277 -- This sentence is pretty wordy and can be greatly shortened.
* L278 "other baselines" -- I would highly recommend adding a separate section that describes the baselines.
* L279 $e=0.4$ -- It could be worth noting that $e \le 0.5$. I initially thought $e \in [0, 1]$ and assumed that 0.4 was a rather small value.
* L281 "which again ... " -- This can be cut.
* L282 "more challenging" -- Inverted pendulum is not really more challenging than CartPole.
* L286 "experiment" -- Grammar error?
* L286 "uniform noise" -- Why is this a reasonable problem setting? Is this similar to some real-world tasks?
* L288 "biased noise" -- Why is the noise biased?
* L300 - L301 -- I didn't understand this sentence.
* L307 "three seeds" -- Is the same expert policy used for each seed? [That's fine, I just wanted to clarify.]
* L315 "GAIL and SQIL" -- These methods are designed for the setting where the agent can interact with the environment. Is this the setting in which they were used?
* L334 "which again ..." -- This sentence can be cut.


-----------------------------
**Review decision**: Overall, the proposed method seems simple and empirically seems to work well. The paper is generally well written. I have two main concerns with the paper. First, it is unclear *why* the proposed method should outperform the baselines; the analysis doesn't seem to answer this question. Second, it is unclear if the paper is actually solving a useful problem. While this problem has been studied in prior work, the problem setting differs from the standard RL setting. I would encourage the authors to address these two concerns in the rebuttal.

**Update after discussion**: Thank you for the discussion and additional experiments studying the "discreteness". I think I finally understand where the "magic" is coming from. At this point, my concerns about (1) why the method should work and (2) why this is a useful problem setting have been well addressed. I have increased my review to score = 7. In the final version of the paper, I would highly encourage the authors to include many of the points from the discussion

**Time Spent Reviewing:**

10 - 20 (including discussion)

---

> ### Author Response · Authors · 2021-08-10
> **Response to Reviewer 58fG (Part 1/2)**
>
> We thank the reviewer for thoughtful reviews and comments. We also appreciate that the reviewer raises insightful questions, provides useful suggestions in improving the presentation, and lists exhaustive writing comments. We address the concerns as follows.
>
> **Q1**: It's unclear if the Theorems show that the proposed method is more sample efficient than either (1) directly learning using noisy rewards/actions or (2) learning using the empirical mean estimates for the rewards/actions.
> **A1**: First of all, we provide some intuitions in the Remark (Line 193 - 200) on why the proposed method is more beneficial compared to the noisy reward. This is mainly because the magnitude of noise terms of noisy reward $\tilde{r}$
>  becomes much larger than proposed in the high-noise regime which will dilute the informativeness of $\mathbb E[r]$. On the contrary, $\mathbb E[\tilde r_{peer}]$ contains a moderate constant noise thus maintaining more useful training signals of the true reward in practice (see experimental results, the models with noisy rewards rarely converge).
>
> To analyze this more rigorously as suggested, we have the following derivations to measure how the statistical estimates diverge from the expected reward $\mathbb E[r]$ (from the sample complexity perspective). Assume $\xi = 1$ and
> $$
> \bar {\tilde r}:=\frac{1}{T} \sum_{t=1}^T \tilde r(s_t,a_t), \quad
> \bar {\tilde r}\_{\text{peer}}:=\frac{1}{T} \sum_{t=1}^T {\tilde r}\_{\text{peer}}(s_t,a_t) =
> \frac{1}{T} \sum_{t=1}^T \left[{\tilde r}(s_t,a_t) + (1- r'_t) \right] - 1.
> $$
>
> From Hoeffding's inequality, noting there are $T$ independent random variables in estimating $\bar {\tilde r}$ and $2T$ independent random variables in estimating $\bar {\tilde r}_{\text{peer}}$ (but normalized by $T$), we know w.p. at least $1-\delta$,
>
> \begin{align}
> |\bar {\tilde r} - \mathbb E [{\tilde r}]| \le R_{\text{max}}\sqrt{\frac{\ln 2/\delta}{2T}}, \quad |\bar {\tilde r}\_{\text{peer}} - \mathbb E [{\tilde r}\_{\text{peer}}]| \le R_{\text{max}}\sqrt{\frac{\ln 2/\delta}{T}}.
> \end{align}
>
> We can denote the relationship between reward estimates and the corresponding error rate estimates $\bar e_-, \bar e_+$ as:
> \begin{align}
> \bar {\tilde r} = (1-\bar e_- - \bar e_+) \bar r + \bar e_- r_+ + \bar e_+ r_-.
> \end{align}
>
> We have
>
> \begin{align}
>     |\bar {\tilde r} - \mathbb E [{\tilde r}]| =& |(1-\bar e_- - \bar e_+) \bar r - (1- e_- -  e_+) \mathbb E[r] + (\bar e_- - e_-) r_+ + (\bar e_+ - e_+) r_-| \\\\
>     =& |(1-\bar e_- - \bar e_+) \bar r - (1- \bar e_- - \bar e_+) \mathbb E[r] + (e_- -\bar e_- + e_+ -\bar e_+ )\mathbb E[r] + (\bar e_- - e_-) r_+ + (\bar e_+ - e_+) r_-| \\\\
>     \ge& (1-\bar e_- - \bar e_+)| \bar r -  \mathbb E[r] | - |\bar e_- - e_- |(r_+ + \mathbb E[r]) - |\bar e_+ - e_+ |(r_-+\mathbb E[r]).
> \end{align}
> Thus
> \begin{align}
> | \bar r -  \mathbb E[r] | \le \frac{ R_{\text{max}}\sqrt{\frac{\ln 2/\delta}{2T}} +  |\bar e_- - e_- |(r_+ + \mathbb E[r]) + |\bar e_+ - e_+ |(r_-+\mathbb E[r])}{1-\bar e_- - \bar e_+}.
> \end{align}
>
> Assume $\delta_e = |\bar e_- - e_- | = |\bar e_+ - e_+ |$. We have
> \begin{equation}
>     | \bar r -  \mathbb E[r] | \le \frac{ R_{\text{max}}\sqrt{\frac{\ln 2/\delta}{2T}} +  \delta_e(r_+ + r_- +  2\mathbb E[r]) }{1-\bar e_- - \bar e_+}.
> \end{equation}
>
> Similarly, for peer rewards, note $ \bar{\tilde{r}}\_{\mathrm{peer}} =(1-\bar e_- - \bar e_+) (\bar r - (1 - \bar p_\text{peer}) r_- - \bar p\_\text{peer} r_+).$
> We have
>
> \begin{align}
>     |\bar{\tilde{r}}\_{\mathrm{peer}} - \mathbb E [ {\tilde r}\_{\text{peer}}]| =& |(1-\bar e_- - \bar e_+) (\bar r - (1 - \bar p\_\text{peer}) r_- - \bar p_\text{peer} r_+)
>     -(1-\bar e_- - \bar e_+) (\mathbb E [r] - (1 -  p_\text{peer}) r_- -  p_\text{peer} r_+) \\\\
>     & + (e_- -\bar e_- + e_+ -\bar e_+ )(\mathbb E [r] - (1 -  p_\text{peer}) r_- -  p_\text{peer} r_+)
>     |\\\\
>     \ge& (1-\bar e_- - \bar e_+)| \bar r -  \mathbb E[r] | - (1-\bar e_- - \bar e_+)|\bar p_{\text{peer}}-p_{\text{peer}}||r_--r_+| - \\\\
>     &|\bar e_- - e_- | \cdot |\mathbb E [r] - (1 -  p_\text{peer}) r_- -  p_\text{peer} r_+| - |\bar e_+ - e_+ |\cdot |\mathbb E [r] - (1 -  p_\text{peer}) r_- -  p_\text{peer} r_+|.
> \end{align}
> Thus
> \begin{align}
> | \bar r -  \mathbb E[r] | \le \frac{ R_{\text{max}}\sqrt{\frac{\ln 2/\delta}{T}} +
>  (|\bar e_+ - e_+ | + |\bar e_- - e_- |)\cdot |\mathbb E [r] - (1 -  p_\text{peer}) r_- -  p_\text{peer} r_+|
> }{1-\bar e_- - \bar e_+} + |\bar p_{\text{peer}}-p_{\text{peer}}||r_--r_+|.
> \end{align}
> Assume $\delta_e = |\bar e_- - e_- | = |\bar e_+ - e_+ | = | \bar p_\text{peer}- p_\text{peer}|$. We have
> \begin{equation}
> | \bar r -  \mathbb E[r] | \le \frac{ R\_{\text{max}}\sqrt{\frac{\ln 2/\delta}{T}} +
>  2 \delta_e\cdot |\mathbb E [r] - (1 -  p\_\text{peer}) r_- -  p\_\text{peer} r_+|
> }{1-\bar e_- - \bar e_+} + \delta_e|r_--r_+|.
> \end{equation}
>
> For the high-noise case, we can infer peer rewards likely have lower sample complexity, i.e. is more sample efficient. For example, when $p_{\text{peer}}=0.5$, $e_-=e_+=0.3$, $R_{\text{max}}=1$, $r_+=1$, $r_-=0$, $\bar e_- - e_- = \bar e_+ - e_+ = \delta_e$, $\mathbb E[r]=0.5$, we have
> \begin{align}
>     | \bar r -  \mathbb E[r] | &\le \frac{ \sqrt{\frac{\ln 2/\delta}{2T}} +  2\delta_e }{0.4} ~~~ \text{(Plain Reward)}, \\\\
> | \bar r -  \mathbb E[r] | &\le \frac{ \sqrt{\frac{\ln 2/\delta}{T}} + 1.4 \delta_e}{0.4}  ~~~ \text{(Peer Reward)} .
> \end{align}
> In this case, we know peer rewards have a tighter error bound for estimating $r$ **when $T$ is sufficiently large.**
>
> **Q2**:  I would highly recommend including the baseline results on the Pendulum task; omitting baseline results on the tasks where the baseline performs as well as the proposed method is concerning. \
> **A2**: Thanks for the suggestion. We are omitting the baseline results since the curve will overlap largely with the peer reward (Line 289-291). As recommended by the reviewer, we will include the surrogate reward results as well in Figure 3. To further alleviate the concern, we conducted an extra experiment on Pong (one larger environment on Atari):
>
> | $e_{-} = e_{+} = 0.2$ | $\mathcal{R}_{avg}$ | $T_{converge}$ |
> |:---------------------:|:-------------------:|:--------------:|
> |  surrogate reward [7] |    $19.3 \pm 0.4$   |   $1.48 * 1e^7$  |
> |          ours         |    $19.1 \pm 0.3$   |   $1.13 * 1e^7$  |
>
> Specifically, we set the noise rate as $e_{-} = e_{+} = 0.2$ and record the average score ($\mathcal R_{avg}$) and time for convergence (number of iterations when the R_avg starts becoming larger than 18.5). As shown in Table, our proposed method can achieve similar performance as [7] while yielding faster convergence (similar to CartPole and Pendulum). Note that our method offloads the burdens of prior knowledge for the confusion matrix.

---

> > ### Author Response · Authors · 2021-08-10
> > **Response to Reviewer 58fG (Part 2/2)**
> >
> > **Q3**: I found the co-training experiments quite hard to follow. \
> > **A3**: We apologize for not stating the experiments for co-training clearly. In general, we aim to solve a more challenging real task where weak supervision occurs during the learning procedure. Different from “RL with perturbed reward”, and “BC with imperfect expert”, we do not use synthetic experimental setup and follow the same setting as [1]. Specifically, we train two agents from scratch and each agent learns from both the environment (RL using the reward provided by the environment) and the weak demonstrations sampled from the other agent. As shown in Table 3 and Figure 5, our proposed PeerCT framework leads to faster convergence and better final results compared with CoPiEr [R1] (A, B in Table 3 denotes two different agents or views). We will make this section clearer in the revision.
> >
> > [R1] Co-training for Policy Learning. Song et al., UAI 2019.
> >
> >
> > **Q4**: Hypothesis about why does peer reward performs better than true reward sometimes? \
> > **Q4.1**: The argument about human rewards being imperfect seems to contradict the claim that the proposed method optimizes whatever reward function is given. \
> > **A4.1**: Yeah, we agree that the “imperfect human rewards” assumption does not match with our assumption in “RL with perturbed rewards” where the rewards are randomly flipped based on a confusion matrix. Actually, this hypothesis is inspired based on the observations in BC and Co-Training experiments. To test the generalizability of our method, we consider two challenging tasks BC with imperfect experts and Co-Training, where the weak supervision is not synthesized based on the noise model. Instead, the weak supervision comes from a not well-trained PPO agent (BC) or another learning agent (Co-Training). From the experiments, we know that our method achieves better performance than the original policy (“true but imperfect” expert), which is **pretty similar** to RL results! Therefore, we make this “imperfect human rewards” hypothesis like imperfect policies (*no noise is injected*) in BC and Co-Training experiments.
> >
> > **Q4.2**: The claim that the proposed method decreases variance seems suspicious because (a) it seems like it actually increases the variance of the gradient estimator and (b) this same logic would suggest that the baseline from [7] should outperform the proposed method. \
> > **A4.2**: We apologize for the incorrect statement here. (for point a) The reviewer is correct, peer reward **does not reduce the variance** compared to the true reward. We still believe that proper reward reshaping might be a potential factor as some previous works found that proper reward scaling plays an important role for RL [R2, R3]. (for point b) Nice reasoning! In [7], the author actually reports similar phenomenon like us [see Figure 1(b) and 1(c)] -- “In some cases, our method even achieves higher cumulative reward - this is surprising to us at first, but we conjecture that the inserted noise together with our noise-removal unbiased estimator adds another layer of exploration, which proves to be beneficial in some settings.” Compared to [7], it seems this phenomenon for peer reward is more obvious for the CartPole environment.
> >
> > We thank the reviewer for in-depth analysis and reasoning for this interesting phenomenon, which indeed merits future study.
> >
> > [R2] Policy invariance under reward transformations: Theory and application to reward shaping. Andrew Y. Ng, Daishi Harada, Stuart Russell. ICML 1999. \
> > [R3] Learning to Utilize Shaping Rewards: A New Approach of Reward Shaping. Hu et al., 2020. NeurIPS 2020.
> >
> > **Q5**: More motivations/explanations on the problem setup - why the problem is useful and interesting? \
> > **A5**: Thanks for the suggestion! We fully agree that more motivations for the *weakly supervised policy learning* setting will definitely help readers understand our problem setup. In standard supervised learning, "learning from noise" or "noisy label problem" has attracted great attention in the research community as acquiring precise annotations is very expensive. To reduce costs and improve annotation efficiency, people reply on some auto-labeling tools and crowdsourcing platforms, which introduce unknown noise in the labels. However, for decision-making problems, there are only a few past works exploring these topics separately in their specific domains [R1, R2].
> >
> > In this work, we would like to bring people's attention to the weak supervision problem in policy learning. Most existing decision-making algorithms require agents to receive high-quality supervision signals, e.g., reward or expert demonstrations, which are either infeasible or prohibitively expensive to obtain in practice. For instance, (1) the reward may be collected through faulty sensors thus not credible [R1, R2], (2) the demonstrations by an expert in behavioral cloning (BC) are often imperfect due to limited resources and environmental noise [R3, R4]. To promote the study in this area, we first formulate a meta-framework to study RL/BC with weak supervision signals and call it *weakly supervised policy learning*. We then introduce a theoretically principled solution to handle this challenging problem  (see Introduction). Another strong evidence that shows the practicability of this problem setup is our experiments on "BC with imperfect demonstrations" and "Co-Training" which mimics the real applications. In these two settings, we did not add synthetic noise but solving problems that can happen in the real world.
> >
> > We will add more motivations in the revision.

---

> > > ### Comment · Reviewer_58fG · 2021-08-12
> > > **Quick clarification**
> > >
> > > Thank you for the detailed clarifications. I had one lingering question:
> > >
> > > **Q1**: I was confused by the answer here. I was imagining that we'd compare the variance of $\tilde{r}$ (L176) with $\tilde{r}_\text{peer} = \tilde{r} - \xi \tilde{r}'$ (L177). Since $\tilde{r}'$ is sampled independently, we know the variance is
> > > $Var[\tilde{r}_\text{peer}] = Var[\tilde{r}] + \xi^2 Var[\tilde{r}'] = (1 + \xi^2) Var[\tilde{r}] > Var[\tilde{r}]$. This would suggest that the rewards used in the proposed method ($\tilde{r}_\text{peer}$ have higher variance than the baseline). Perhaps there's a bug in my reasoning?

---

> > > > ### Author Response · Authors · 2021-08-12
> > > > **Thanks for the further question!**
> > > >
> > > > Yes, you are right that the variance of peer rewards is higher than the variance of original rewards. Note a typo fixed in our response Response to Reviewer 58fG (Part 1/2). The error upper bound of $\bar{\tilde r}\_{peer}$ should be $O(\frac{1}{T})$ rather than $O(\frac{1}{4T})$. So by comparing two error bounds, we have
> > > > \begin{align}
> > > > |\bar {\tilde r} - \mathbb E [{\tilde r}]| \le R\_{\text{max}}\sqrt{\frac{\ln 2/\delta}{2T}}, ~and ~~ |\bar {\tilde r}\_{\text{peer}} - \mathbb E [\bar {\tilde r}\_{\text{peer}}]| \le R\_{\text{max}}\sqrt{\frac{\ln 2/\delta}{T}}.
> > > > \end{align}
> > > > From this aspect, peer rewards have a larger variance, which supports your analyses. But it is fine since neither the noisy rewards nor the noisy peer rewards are our focus. We need to pay attention to the true reward $r$. That's why we try to analyze the error bound $|\bar r-\mathbb E[r]|$ in both cases.
> > > > Based on the analyses in our previous response, we have:
> > > > \begin{align}
> > > >     | \bar r -  \mathbb E[r] | \le \frac{ \sqrt{\frac{\ln 2/\delta}{2T}} +  2\delta_e }{0.4} ~~~ \text{(true reward estimation without peer)},
> > > > \end{align}
> > > > and
> > > > \begin{align}
> > > > | \bar r -  \mathbb E[r] | \le \frac{ \sqrt{\frac{\ln 2/\delta}{T}} +
> > > >  1.4 \delta_e
> > > > }{0.4}  ~~~ \text{(true reward estimation with peer)} .
> > > > \end{align}
> > > > In this case, we know peer rewards have a tighter error bound for estimating $r$ when $T$ is sufficiently large. We appreciate your question, which helps us further explain why our method is potentially better.

---

> > > > > ### Comment · Reviewer_58fG · 2021-08-16
> > > > > **Clarification**
> > > > >
> > > > > I'm a bit confused by the response, in particular the following:
> > > > >
> > > > > >  But it is fine since neither the noisy rewards nor the noisy peer rewards are our focus. We need to pay attention to the true reward $r$
> > > > >
> > > > > My understanding is that (1) there exists a true deterministic reward function, (2) we only observe noisy samples of this reward function, and (3) the proposed method adds noise to these noisy samples (in the form of sampling rewards from different states and actions). Is this correct?
> > > > >
> > > > > If the used reward function $\bar{r}$ is an unbiased estimator of the true reward function, then the expected squared deviation $E[(\bar{r} - E[r])^2]$ is simply the variance. The reasoning in my previous comment suggests that the this variance is higher for the proposed method than for the baseline.

---

> > > > > > ### Author Response · Authors · 2021-08-18
> > > > > > **Response to "Reviewer's Clarification"**
> > > > > >
> > > > > > Your understanding (1-3) is correct and we agreed that the variance of peer reward is larger than the baseline (noisy reward) in previous responses.
> > > > > > \begin{align}
> > > > > > Var[\tilde r_{\text{peer}}] = Var[\tilde r] + \xi^2 Var[\tilde r] = (1 + \xi^2) Var[\tilde r] > Var[\tilde r]
> > > > > > \end{align}
> > > > > > However, we would like to highlight that noisy reward has a lower variance **does not mean** that it is superior to our proposal - peer reward. This is because (1) **neither noisy reward nor peer reward is an unbiased estimation of the true reward**, (2) the scale of $\mathbb{E}[\tilde r]$ is different from $\mathbb E[\tilde r_{\text{peer}}]$ due to the peer penalty term. Specifically,
> > > > > > \begin{align}
> > > > > > Var[\tilde r]  = \mathbb{E}[(\tilde r - \mathbb{E}[\tilde r])^2], \quad Var[\tilde r\_{\text{peer}}]  = \mathbb{E}[(\tilde r\_{\text{peer}} - \mathbb E[\tilde r\_{\text{peer}}])^2]
> > > > > > \end{align}
> > > > > > Note that $\mathbb E[\tilde r]$ and $\mathbb E[\tilde r_{\text{peer}}]$ are an affine transformation of $\mathbb E[r]$, as shown in Line 195-196 (Remark 1):
> > > > > > \begin{align}
> > > > > >     \mathbb E[\tilde r] &= \eta \cdot \left(\mathbb E[r] + \frac{e_+}{1 - e_- - e_+} r_- + \frac{e_-}{1 - e_- - e_+} r_+\right),\\\\
> > > > > >     \mathbb E [\tilde{r}\_{\text{peer}}] &= \eta \cdot (\mathbb E [r] - (1 - p\_{\text{peer}}) r_- - p\_{\text{peer}} r_+) \ne \mathbb E[\tilde r] ,
> > > > > > \end{align}
> > > > > > It is less insightful to compare $Var[\tilde r_{\text{peer}}]$ and $Var[\tilde r]$ directly and we cannot conclude which approach is better only from the evidence $Var[\tilde r_{\text{peer}}] > Var[\tilde r]$.
> > > > > >
> > > > > > To understand why peer reward performs better than baseline, we refer the reviewer to Remark 1 (Line 193 - 200). Compared to the baseline, our proposal can offset the noise introduced by penalizing over-agreements (subtracting randomly sampled noisy pairs) using the CA mechanism. This leads to a lower constant offset  in the high-noise regime. Specifically, when $e_{-} + e_{+} \rightarrow 1$, we know that $\mathbb E[\tilde r]$ is completely dominated by the second noise term ($\frac{e_+}{1 - e_- - e_+} r_- + \frac{e_-}{1 - e_- - e_+} r_+ \rightarrow \infty$). In other words, the difference in first term under different actions (optional or non-optimal) will become less significant due to the large noise offset. On the contrary, for our peer reward, this problem will not appear as $(1 - p\_{\text{peer}}) r_- + p\_{\text{peer}} r_+$ has the same scale as $r$ so the noise will not cover the useful reward signals.
> > > > > >
> > > > > > We hope this clarifies our response clearly. Looking forward to any follow-up discussions!

---

> > > > > > > ### Comment · Reviewer_58fG · 2021-08-22
> > > > > > > **Discussion**
> > > > > > >
> > > > > > > Apologies for the delayed response.
> > > > > > >
> > > > > > > The author's clarification about the rewards having different expectations makes sense; I agree that comparing variances in that setting doesn't make sense.
> > > > > > >
> > > > > > > Perhaps I can explain my reservation in a different way: from 30,000 ft, the paper seems to argue that in settings with noisy rewards, adding even more noise results in better learning. This is a very surprising claim. To draw an analogy, it seems similar to arguing that estimating $\mu$ from samples $x \sim N(\mu, \sigma=1)$ is *worse* than estimating $\mu$ from samples $x+\epsilon$, where $x \sim N(\mu, \sigma=1), \epsilon \sim N(0, \sigma=0.1)$. I'm trying to understand why the claim in this analogy is bogus, but the claim in the paper isn't.

---

> > > > > > > > ### Author Response · Authors · 2021-08-22
> > > > > > > > **Further Clarification**
> > > > > > > >
> > > > > > > > Thanks for your follow-up discussions! We understand that your concerns actually come from this fact that adding more noise to noisy rewards results in better learning, which seems surprising and counterintuitive. For your better understanding, we organize the responses in the following two aspects: (1) clarification on your analogy example (why it cannot reflect the case we are dealing with), (2) more intuitions on why proposed CA objective works.
> > > > > > > >
> > > > > > > > First of all, we emphasize that the noise model considered in the paper is actually a form of **biased-noise**. This is **very different** from the noise in your analogy example which considered **zero-mean** Gaussian noise. Under the zero-mean noise model, the focus is to reduce the variance thus adding more noise will make the estimation more difficult. However, in the more challenging biased noise setting [1, 2, 3] the focus is more on the side of bias correction instead of variance reduction. Specifically, we have
> > > > > > > > \begin{align}
> > > > > > > >     \mathbb E[\tilde r] &= (1 - e_- - e_+) \cdot \left(\mathbb E[r] + \frac{e_+}{1 - e_- - e_+} r_- + \frac{e_-}{1 - e_- - e_+} r_+\right),\\\\
> > > > > > > >     \mathbb E [\tilde{r}\_{\text{peer}}] &= (1- e_- - e_+) \cdot (\mathbb E [r] - (1 - p\_{\text{peer}}) r_- - p\_{\text{peer}} r_+).
> > > > > > > > \end{align}
> > > > > > > > Assume this binary noise setting: $e_{-} = e_{+} = 0.45, (r_{-}, r_{+}) = (0, 1)$, then we know that
> > > > > > > > $\mathbb E[\tilde r] = 0.1 (\mathbb E[r] + 4.5)$ and $\mathbb E[\tilde r_\text{peer}] = \mathbb E[r_\text{peer}] = 0.1 (\mathbb E[r] - p_\text{peer})$, where $p_\text{peer} \in (0, 1)$. Under this setting, it is easier to estimate $10 \cdot \mathbb E[r]$ using $\tilde r_\text{peer}$ instead of $\tilde r$ since the noise rate is unknown and peer solution is less biased. Note that we also conduct experiments with variance reduction technique (VRT) in the supplementary material (Sec C.5) and we found that simply reducing variance will not help convergence except the bias is properly corrected. Similar observations are also reported in [2]. **In summary, we know that although peer reward (similar to surrogate reward in previous literature [2]) increases the variance (no free-lunch), it will lead to a better estimation of true reward due to lower bias.**
> > > > > > > >
> > > > > > > > Second, we understand it might be not intuitive why the CA objective can implicitly correct the biased-noise in the weak supervision perfectly. **Note that the correctness is already verified by previous literature [3] in the supervised learning setting**.
> > > > > > > > Intuitively, both the noisy reward  and peer penalty term encodes the noise knowledge implicitly. The carefully constructed form of peer reward allows offsetting the noise in these two terms, leading to a nice property for peer reward - invariant against the noise and is an affine transformation of true reward in expectation.
> > > > > > > >
> > > > > > > > The intuition of the CA objective comes from information verification methods in the crowdsourcing literature. Assuming we have collected reports on multiple tasks from several reporters via crowdsourcing, correlated agreement (CA) is a mechanism that scores the reports from any two reporters $A$ and $B$ on the same task $i$ without accessing the ground truth. Let $S$ be a simple score function based on whether two reports agree (like $S(r_1, r_2) = 1$ if $r_1 = r_2$, and 0 otherwise). CA will first score the reports from the two reporters on the task $i$ with the score function , i.e. $S(r_A^i, r_B^i)$, where superscript denotes the task and subscript denotes the reporter. Then CA samples two peer tasks $j, k$ and scores the reports from $A, B$ on them respectively, i.e. $S(r_A^j, r_B^k)$. The final score is $S(r_A^i, r_B^i) - S(r_A^j, r_B^k)$. This mechanism punishes blind agreement in which case the two reporters yield the same reports across all the tasks, which is non-informative but will receive a high score under only the first term $S(r_A^i, r_B^i)$. It has been proved that under mild assumptions, with the CA scoring function, the truth-telling strategy obtains the highest score. Back to our PeerPL, we treat our policy as one reporter, the weak supervision as another one, and our $Eva$ function as the score function to get our CA objective. The truth-telling strategy corresponds to the optimal policy, which the CA objective will intuitively induce.
> > > > > > > >
> > > > > > > > Looking forward to further discussions and clarifications if there is still something unclear.
> > > > > > > >
> > > > > > > > [1] Learning with Noisy Labels. Natarajan et al., 2013. \
> > > > > > > > [2] Reinforcement Learning with Perturbed Reward. Wang et al., 2019. \
> > > > > > > > [3] Peer Loss Functions: Learning from Noisy Labels without Knowing Noise Rates. Liu et al., 2020.

---

> > > > > > > > > ### Comment · Reviewer_58fG · 2021-08-23
> > > > > > > > > **Discussion**
> > > > > > > > >
> > > > > > > > > Thank you for the additional clarification. The issue of bias is one that I had indeed overlooked. After reading through the response and the proof of Lemma 1 in the appendix, it now makes sense to me how the peer RL trades bias for variance.
> > > > > > > > >
> > > > > > > > > At this point, I have two questions:
> > > > > > > > >
> > > > > > > > > **Question 1**: Peer RL seems to remove the bias from the noisy rewards, but it is still a scaled version of the expected reward, right? Recovering the true expected reward would require knowledge of the noise parameters $e_+$ and $e_-$, right?
> > > > > > > > >
> > > > > > > > > **Question 2**: In the finite-horizon setting, RL is invariant to affine transformations of the reward function. Thus, the bias that peer RL avoids wouldn't actually change the optimal policy, right?
> > > > > > > > >
> > > > > > > > > **Simple Experiment**: Finally, to get some more intuition peer RL, I implemented a very rudimentary version on a 2-state Markov process (no actions). The aim was to see whether we could infer which state was better. This experiment (code below) seems to suggest that the baseline (which ignores the bias and just takes the average of the noisy rewards) is more accurate at inferring which state is better compared with peer RL (81% accuracy vs 73% accuracy). Please let me know if it seems there is a bug in this simple example, or if there's a good reason why it isn't representative of more complex problems.
> > > > > > > > >
> > > > > > > > > ```
> > > > > > > > > import numpy as np
> > > > > > > > > import tqdm
> > > > > > > > >
> > > > > > > > > def get_rewards(state, num_samples):
> > > > > > > > >   if state == 0:
> > > > > > > > >     r = np.random.choice([0, 1], p=[0.4, 0.6], size=num_samples)  # E[r] = 0.6
> > > > > > > > >   elif state == 1:
> > > > > > > > >     r = np.random.choice([0, 1], p=[0.6, 0.4], size=num_samples)  # E[r] = 0.4
> > > > > > > > >   else:
> > > > > > > > >     raise NotImplementedError
> > > > > > > > >   # Add noise
> > > > > > > > >   mask = np.random.choice(2, p=(1 - 0.45, 0.45), size=num_samples)
> > > > > > > > >   r = (1 - mask) * r + mask * (1 - r)
> > > > > > > > >   return r
> > > > > > > > >
> > > > > > > > > # Baseline
> > > > > > > > > num_samples = 1000
> > > > > > > > > is_correct_vec = []
> > > > > > > > > for _ in tqdm.trange(10000):
> > > > > > > > >   r_vec = np.stack([get_rewards(0, num_samples), get_rewards(1, num_samples)], axis=1)
> > > > > > > > >   r_hat = np.mean(r_vec, axis=0)
> > > > > > > > >   is_correct = (r_hat[0] > r_hat[1])
> > > > > > > > >   is_correct_vec.append(is_correct)
> > > > > > > > > print('\nBaseline Success: %.3f\n' % np.mean(is_correct_vec))
> > > > > > > > >
> > > > > > > > > # Peer RL
> > > > > > > > > is_correct_vec = []
> > > > > > > > > for _ in tqdm.trange(10000):
> > > > > > > > >   neg_samples = np.concatenate([get_rewards(0, num_samples), get_rewards(1, num_samples)])
> > > > > > > > >   np.random.shuffle(neg_samples)  # Randomly permutes the elements
> > > > > > > > >   neg_samples0, neg_samples1 = np.split(neg_samples, 2)
> > > > > > > > >   r_vec = np.stack([get_rewards(0, num_samples) - neg_samples0, get_rewards(1, num_samples) - neg_samples1], axis=1)
> > > > > > > > >   r_hat = np.mean(r_vec, axis=0)
> > > > > > > > >   is_correct = (r_hat[0] > r_hat[1])
> > > > > > > > >   is_correct_vec.append(is_correct)
> > > > > > > > > print('\nPeer RL Success: %.3f\n' % np.mean(is_correct_vec))
> > > > > > > > > ```

---

> > > > > > > > > > ### Author Response · Authors · 2021-08-23
> > > > > > > > > > **Follow-up Discussion**
> > > > > > > > > >
> > > > > > > > > > Thanks for your follow-up discussions!
> > > > > > > > > >
> > > > > > > > > > **Question 1**: Peer RL seems to remove the bias from the noisy rewards, but it is still a scaled version of the expected reward, right? Recovering the true expected reward would require knowledge of the noise parameters $e_{+}$ and $e_{-}$, right? \
> > > > > > > > > > **Answer 1**: Yes, if we want to “fully recover” the true expected reward (not the scaled version), the noise parameters $e_{+}$ and $e_{-}$ have to be known. However, in many cases, we do not need to obtain the exact estimation of true reward (RL) or true labels (supervised learning) for optimization. For instance, the RL algorithms can converge to the optimal policy with positively scaled reward. For supervised learning, we can optimize with positively scaled loss functions - peer loss can recover Bayes optimal classifier [3]. In practice, the noise rates are often inaccessible or infeasible to estimate. Compared to previous works [1,2], our solution offloads the burden of estimating noise rates.
> > > > > > > > > >
> > > > > > > > > > **Question 2**: In the finite-horizon setting, RL is invariant to affine transformations of the reward function. Thus, the bias that peer RL avoids wouldn't actually change the optimal policy, right? \
> > > > > > > > > > **Answer 2**: Yes, under the current assumptions (tabular Q-Learning), the baseline will also converge to the optimal policy with sufficient observations in theory. However, in practice, the large noise offset (which is dependent on the nose rate) for baseline will cover the useful signals thus leading to lower convergence speed as shown in Figure 2 (it converges to the optimal point but with more iterations). In more complex Pendulum or Atari tasks, the algorithm with noisy rewards (baseline) can barely converge to the optimal policy. Similarly, for the supervised learning setting, [1] shows that learning with noisy labels will yield significant performance drop during testing. We will also demonstrate the benefits of PeerRL in the toy example you provided.
> > > > > > > > > >
> > > > > > > > > > [1] Learning with Noisy Labels. Natarajan et al., 2013. \
> > > > > > > > > > [2] Reinforcement Learning with Perturbed Reward. Wang et al., 2019. \
> > > > > > > > > > [3] Peer Loss Functions: Learning from Noisy Labels without Knowing Noise Rates. Liu et al., 2020.
> > > > > > > > > >
> > > > > > > > > > **Simple Experiment**: Thanks a lot for providing this intuitive example with minimal implementation! We found it is actually a great example to show the benefits of PeerRL. If possible, we would like to incorporate it in the revision and provide more analysis.
> > > > > > > > > > - **Why PeerRL perform worse than baseline (73% vs 81% actually)**: This is due to improper choice of penalty coefficient $\xi = 1$. As also shown in the paper (Table 2 and Sec C.4 in Appendix: Figure A2, A3, A4), $\xi = 1$ is too large thus leading to a bad performance (too much noise, high variance). If we set $\xi=0.1$ (or $\xi=0.2$) in this toy example,
> > > > > > > > > > ```
> > > > > > > > > > r_vec = np.stack([get_rewards(0, num_samples) - 0.1 * neg_samples0, get_rewards(1, num_samples) - 0.1 * neg_samples1], axis=1)
> > > > > > > > > > ```
> > > > > > > > > > To reduce the variance of results, we increase number of trials from 10000 to 100000. Then the accuracy of inferring better state using PeerRL becomes 81.3%, which is slightly better than baseline's 80.9%.
> > > > > > > > > >
> > > > > > > > > > - We also want to emphasize that in this example, the task is very simple - using the mean to infer which state is better. Especially when `num_samples` is large, the difference between two approaches is minor. More specifically, if we reduce `num_samples` from 1000 to 100, then we have the following results: baseline: 58.3%, PeerRL: 61.2% (+2.9%). If we reduce `num_samples` to 10, we have the following results: baseline: 44.6%, PeerRL: 52.3% (+7.7%). On the contrary, if we increase `num_samples` to 10000, both baseline and PeerRL will have 99.7% accuracy (since there are sufficient observations and the noise is distinguishable).
> > > > > > > > > >
> > > > > > > > > > Any further discussions are welcome and appreciated!

---

> > > > > > > > > > > ### Comment · Reviewer_58fG · 2021-08-23
> > > > > > > > > > > **Discussion**
> > > > > > > > > > >
> > > > > > > > > > > Thank you for answering the questions, as well as for providing feedback on the simple experiment. The importance of $\xi$ was indeed something I had overlooked.
> > > > > > > > > > >
> > > > > > > > > > > Spending more time thinking about this paper, my current hypothesis is that the good performance of PeerRL (relative to the baseline) exploits the "discreteness" of the rewards (an assumption clearly stated in the paper). Namely, that the rewards for different states can be "tied" (i.e., receive equal value) and that such ties are counted as failures. For example, consider the setting where `num_samples = 1`. There are then 4 possible values for the rewards of the two states ((0, 0), (0, 1), (1, 0), and (1, 1)). One outcome ((1, 0)) corresponds to the correct guess: that state0 has higher value than state1. Likewise, one outcome ((0, 1)) corresponds to the incorrect guess, and two outcomes ((0, 0), (1, 1)) correspond to a "tie". It is unclear whether guessing "tie" is either good or bad. Nonetheless, we can compute the probabilities of these outcomes for both the baseline (which looks at $\tilde{r}$) and for PeerRL (which looks at $\tilde{r}_\text{peer}$). I am using $\xi = 1$ to simplify the algebra; empirically the conclusions for $\xi = 0.1$ seem similar.
> > > > > > > > > > >
> > > > > > > > > > > | | Baseline      | Peer RL |
> > > > > > > > > > > | --------- | ----------- | ----------- |
> > > > > > > > > > > | Correct | 0.26      | 0.32       |
> > > > > > > > > > > | Tie | 0.5   | 0.375        |
> > > > > > > > > > > | Incorrect | 0.24   | 0.305        |
> > > > > > > > > > >
> > > > > > > > > > > (Let me know if it is unclear where these numbers are coming from, and I can provide the algebra.)
> > > > > > > > > > >
> > > > > > > > > > > There are two ways to read this table. If we count "Ties" as failures, then PeerRL is definitively better than the baseline (32% vs 26%); if we count "Ties" as successes, than the baseline is definitively better (76% vs 69.5%). My conclusion from this analysis is that any results with CA are highly contingent on how "ties" are evaluate. **Do the authors agree with this analysis? Is there a reason to suspect that breaking ties in one way or another is especially important for the RL setting?**
> > > > > > > > > > >
> > > > > > > > > > > To try to understand this "discreteness" further, I replaced the categorical reward distribution with a Normal reward distribution ($N(\mu = 0.4, \sigma = 1)$ vs $N(\mu = 0.6, \sigma = 1)$). For various values of $\xi$, I consistently found that the baseline performed at least as well as PeerRL. **Do the authors believe that a version of PeerRL should work with continuous rewards?**

---

> > > > > > > > > > > > ### Author Response · Authors · 2021-08-24
> > > > > > > > > > > > **Follow-up Discussion**
> > > > > > > > > > > >
> > > > > > > > > > > > Thanks for your thoughtful analysis! We totally agree with the hypothesis that PeerRL exploits the "discreteness" to break ties in reward evaluation. We provided more experimental results for the toy example and they seem to support this hypothesis as well.
> > > > > > > > > > > >
> > > > > > > > > > > > **Should Tie be evaluated as success or failure?** \
> > > > > > > > > > > > Generally speaking, the "tie" states are usually unstable, uncertain and less informative for optimization. For instance, in supervised learning, tie state means that the classifier’s predictions in two classes have similar confidence levels (similar logits) which is not a final state if sufficient data are provided. For RL, tie state means that the rewards for different states are the same which is less informative as they neither serve as positive nor negative examples. More specifically, similar to some RL exploration works [1], PeerRL aims to minimize the uncertainty thus encouraging exploration in the early stage. On the other hand, it is known that positive examples are extremely important in RL learning. From the toy example, we know that PeerRL might access more positive examples, which is beneficial for optimization. For some RL algorithms (e.g., DQN), these useful experiences might even be upsampled and stored (replay buffer) thus yield faster convergence.
> > > > > > > > > > > >
> > > > > > > > > > > > **Tie breaking in CA?** \
> > > > > > > > > > > > "Tie breaking" is an intriguing property of the CA mechanism, which has been observed and analyzed by some very recent works.
> > > > > > > > > > > >
> > > > > > > > > > > > In the simpler supervised learning setting, [2] shows that both loss correction [3]  and CA (peer loss) will extreme the decisions to the correct direction and CA extremes even more when it is confident (see analysis of Corollary 2). Therefore, when there is sufficient information (number of examples), peer loss tends to perform better than loss correction. “When there is insufficient information, peer loss extremes the prediction to the wrong label with a substantial probability” thus “the power of peer loss does seem to drop”. These claims verify that CA tends to downplay the tie states and extreme the predictions.
> > > > > > > > > > > >
> > > > > > > > > > > > For RL, we already obtain similar insights from the toy example you provided. However, more rigorous analysis merits further study since the dynamics become more complicated. In this two-state Markov process example, we ignore the sequential property (the future states will be affected by current observations) and mimic the agent policy learning procedure with a simple nonparametric solution.
> > > > > > > > > > > >
> > > > > > > > > > > > **Extension to continuous reward?** \
> > > > > > > > > > > > First of all, in our experiments, we use the discretization to handle the continuous reward (Pendulum) following [4]. As shown in the following results, if we use clipping to bound the continuous noise to [0, 1] (introducing some discreteness factor), PeerRL outperforms the baseline consistently.
> > > > > > > > > > > >
> > > > > > > > > > > > (all experiments are conducted with `num_samples = 2`, `num_trials=100,000` and $\xi=0.1$) \
> > > > > > > > > > > > **Continuous noise + clipping** \
> > > > > > > > > > > > *Case1* \
> > > > > > > > > > > > state1: `r = np.clip(np.random.normal(0.6, 1.0, num_samples), 0, 1)` \
> > > > > > > > > > > > state2: `r = np.clip(np.random.normal(0.4, 1.0, num_samples), 0, 1)`
> > > > > > > > > > > >
> > > > > > > > > > > > |          | Correct | Tie   | Incorrect |
> > > > > > > > > > > > |----------|---------|-------|-----------|
> > > > > > > > > > > > | Baseline | 0.546   | 0.056 | 0.398     |
> > > > > > > > > > > > | PeerRL   | 0.580   | 0.003 | 0.417     |
> > > > > > > > > > > >
> > > > > > > > > > > > *Case2* \
> > > > > > > > > > > > state1: `r = np.clip(np.random.normal(0.6, 5.0, num_samples), 0, 1)` \
> > > > > > > > > > > > state2: `r = np.clip(np.random.normal(0.4, 5.0, num_samples), 0, 1)`
> > > > > > > > > > > >
> > > > > > > > > > > > |          | Correct | Tie   | Incorrect |
> > > > > > > > > > > > |----------|---------|-------|-----------|
> > > > > > > > > > > > | Baseline | 0.419   | 0.214 | 0.366     |
> > > > > > > > > > > > | PeerRL   | 0.502   | 0.046 | 0.451     |
> > > > > > > > > > > >
> > > > > > > > > > > > **Discretized continuous noise**
> > > > > > > > > > > > ```
> > > > > > > > > > > > bins = np.arange(0, 1.01, 0.01)  # 100 intervals
> > > > > > > > > > > > inds = np.digitize(r, bins)
> > > > > > > > > > > > r = bins[inds - 1]
> > > > > > > > > > > > ```
> > > > > > > > > > > > *Case1* \
> > > > > > > > > > > > state 1: `r = np.random.normal(0.6, 1.0, num_samples)` \
> > > > > > > > > > > > state 2: `r = np.random.normal(0.4, 1.0, num_samples)`
> > > > > > > > > > > >
> > > > > > > > > > > > |          | Correct | Tie   | Incorrect |
> > > > > > > > > > > > |----------|---------|-------|-----------|
> > > > > > > > > > > > | Baseline | 0.429   | 0.150 | 0.421     |
> > > > > > > > > > > > | PeerRL   | 0.491   | 0.024 | 0.485     |
> > > > > > > > > > > >
> > > > > > > > > > > > *Case2* \
> > > > > > > > > > > > state 1: `r = np.random.laplace(0.6, 1.0, num_samples)` \
> > > > > > > > > > > > state 2: `r = np.random.laplace(0.4, 1.0, num_samples)`
> > > > > > > > > > > >
> > > > > > > > > > > > |          | Correct | Tie   | Incorrect |
> > > > > > > > > > > > |----------|---------|-------|-----------|
> > > > > > > > > > > > | Baseline | 0.429   | 0.150 | 0.421     |
> > > > > > > > > > > > | PeerRL   | 0.491   | 0.024 | 0.485     |
> > > > > > > > > > > >
> > > > > > > > > > > > **Poisson noise** \
> > > > > > > > > > > > state1: `r = np.random.poisson(0.6, 1.0, num_samples)` \
> > > > > > > > > > > > state2: `r = np.random.poisson(0.4, 1.0, num_samples)`
> > > > > > > > > > > >
> > > > > > > > > > > > |          | Correct | Tie   | Incorrect |
> > > > > > > > > > > > |----------|---------|-------|-----------|
> > > > > > > > > > > > | Baseline | 0.430   | 0.166 | 0.405     |
> > > > > > > > > > > > | PeerRL   | 0.502   | 0.030 | 0.468     |
> > > > > > > > > > > >
> > > > > > > > > > > > We believe that these two cases (assumptions) are practical in some real applications. However, we admit that these two cases still exploit the discreteness for tie breaking.
> > > > > > > > > > > >
> > > > > > > > > > > > **Unbounded continuous noise** \
> > > > > > > > > > > > *Case1* \
> > > > > > > > > > > > state1: `r = np.random.normal(0.6, 1.0, num_samples)` \
> > > > > > > > > > > > state2: `r = np.random.normal(0.4, 1.0, num_samples)`
> > > > > > > > > > > >
> > > > > > > > > > > > |          | Correct | Tie   | Incorrect |
> > > > > > > > > > > > |----------|---------|-------|-----------|
> > > > > > > > > > > > | Baseline | 0.580   | 0.000 | 0.420     |
> > > > > > > > > > > > | PeerRL   | 0.579   | 0.000 | 0.421     |
> > > > > > > > > > > >
> > > > > > > > > > > > *Case 2* \
> > > > > > > > > > > > state1: `r = np.random.laplace(0.6, 1.0, num_samples)` \
> > > > > > > > > > > > state2: `r = np.random.laplace(0.4, 1.0, num_samples)`
> > > > > > > > > > > >
> > > > > > > > > > > > |          | Correct | Tie   | Incorrect |
> > > > > > > > > > > > |----------|---------|-------|-----------|
> > > > > > > > > > > > | Baseline | 0.560   | 0.000 | 0.440     |
> > > > > > > > > > > > | PeerRL   | 0.563   | 0.000 | 0.437     |
> > > > > > > > > > > >
> > > > > > > > > > > >
> > > > > > > > > > > > If we consider the unbounded continuous noise, then the performance of baseline and PeerRL is equivalent. This also makes sense since **it is impossible to distinguish the noise properly without any assumption of the noise model**. Please see Q4/A4 for reviewer Hvaz in which we provide more discussions on the challenges to handle continuous noise. **To our best knowledge, there is no existing work dealing with the general biased continuous noise**.
> > > > > > > > > > > >
> > > > > > > > > > > > [1] Curiosity-driven Exploration by Self-supervised Prediction. Pathak et al., 2017. \
> > > > > > > > > > > > [2] Understanding Instance-Level Label Noise: Disparate Impacts and Treatments. Liu et al., 2021. \
> > > > > > > > > > > > [3] Learning from Noisy Labels. Natarajan et al., 2013. \
> > > > > > > > > > > > [4] Reinforcement Learning with Perturbed Reward. Wang et al., 2019.
> > > > > > > > > > > >
> > > > > > > > > > > > Any further discussions are welcome and appreciated!

---

> > > > > > > > > > > > > ### Comment · Reviewer_58fG · 2021-08-25
> > > > > > > > > > > > > **Author response**
> > > > > > > > > > > > >
> > > > > > > > > > > > > Thank you for the discussion and additional experiments studying the "discreteness". I think I finally understand where the "magic" is coming from. At this point, my concerns about (1) why the method should work and (2) why this is a useful problem setting have been well addressed. I have increased by review to score = 7.
> > > > > > > > > > > > >
> > > > > > > > > > > > > In the final version of the paper, I would highly encourage the authors to include many of the points from this discussion, especially:
> > > > > > > > > > > > > * Intuition for why CA should be possible. On the surface, the claim that adding more noise should result in better estimates seems nonsensical. The claim that the method is decreasing bias at the cost of increasing variance likewise doesn't make sense in the RL setting, as RL methods are invariant to bias. Readers who spent a few minutes skimming the current paper will probably (incorrectly) conclude that it is simply wrong. I would recommend revising the abstract and introduction to raise this potential concern (e.g., "At first glance, it may appear that adding noise would make one's estimates strictly worse. Indeed, this is the case for many types of noise and evaluation criterion. However, in \\$(our setting), we exploit the inductive bias that \\$(inductive bias)..."
> > > > > > > > > > > > > * Emphasizing the importance of $\xi$
> > > > > > > > > > > > > * Clarifying how CA exploits discreteness/boundedness
> > > > > > > > > > > > > * Additional motivation for why the noisy setting is useful (perhaps emphasize human-in-the-loop settings).
> > > > > > > > > > > > > * Include all results from the pendulum task

---

> > > > > > > > > > > > > > ### Author Response · Authors · 2021-08-25
> > > > > > > > > > > > > > **Many thanks for your thoughtful suggestions**
> > > > > > > > > > > > > >
> > > > > > > > > > > > > > Thanks a lot for spending precious time (10-20 hours) in reviewing, participating in discussions, raising insightful questions and even implementing a toy example for in-depth analysis! We are really grateful for your great help in improving the paper presentation and glad that your concerns are well addressed. As suggested, we would like to improve the presentation in the following aspects:
> > > > > > > > > > > > > >
> > > > > > > > > > > > > > - Clarifying clearer why CA can help the evaluation process in our setting as suggested - abstract and introduction.
> > > > > > > > > > > > > > - Highlighting the importance of discreteness/boundedness assumption for CA and explaining how it exploits this property. We would like to demonstrate it using the toy examples with corresponding analysis presented in our discussions.
> > > > > > > > > > > > > > - Emphasizing the importance of $\xi$ using a Remark paragraph (moving some contents from Appendix).
> > > > > > > > > > > > > > - Better motivating the weak supervision setting for policy learning with the “human-in-the-loop” and “robotics with sensor noise/adversary”settings.
> > > > > > > > > > > > > > - Including the surrogate reward baseline for pendulum task and making a table as well (similar to Atari experiments for BC) to measure the significance.
> > > > > > > > > > > > > >
> > > > > > > > > > > > > > Your suggestions/comments are highly appreciated and we would like to thank you in the acknowledgement.

---

### Official Review · Reviewer_xvWU · 2021-07-01

**Rating:** 6
**Confidence:** 3

**Summary:**

This paper proposed a framework to improve the efficacy of sensorimotor control learning. The main idea consists of augmenting the training loss with a specifically designed term that avoids the policy to "overfit" to noise present in the reward (for RL) or in the demonstrations (for behavioral cloning).  This extra term aims to minimize the probability that when randomly associating rewards and state-action pairs, the policy performs poorly. This idea, borrowed by the theory of correlated agreement, should improve the policy performance when the supervision (in terms of demonstration or reward) is corrupted by noise. Experiments show that the proposed framework achieves comparable or better results than prior works.

**Limitations And Societal Impact:**

The limitations and possible societal impacts have been properly discussed in the checklist. It would be nice to move some sentences to the main paper, just to have a complete picture in the discussion.

**Main Review:**

The proposed framework appears to build upon existing work on correlation agreement ([34] in paper), but generalizing such methods to the problem of decision making. Overall, I think that the strong connection with prior work was not adequately discussed. Indeed, it is unclear what are the fundamental differences with existing methods (if it is just an application is not generally a problem but should be clearly stated).

In my opinion, the work presents an interesting approach to a very important and general problem (learning policies with noisy supervision). However, the method and the experiments do not completely back up all the claims. More specifically:

- The proof that the proposed method (asymptotically) converges to the optimal policy despite noisy rewards seems a bit hand-wavy and not completely rigorous. The remark on Lemma 1 (line 193) shows the noisy supervision only differs up to a constant with respect to the proposed peer supervision method. Clearly, this constant can have a strong effect on the final performance, but how much should the noise be? This question also raises the question of whether under the assumption of a large enough sample size and constant noise even the policy without any peer reward would converge to the optimum. Proof A1 in the appendix seems to support this hypothesis since only an affine transformation of the reward (independent of the noise level) is required to guarantee convergence (line 580).
- The experimental results are overall extensive, but comparison with previous work is only partial and sometimes even omitted. Specifically, it would have been interesting to see how the approach would perform in comparison to [7], which exactly aims to solve the same problem (learning from noisy rewards). While this comparison is present for the simplest of the demonstrators (the pendulum) it was omitted for the more challenging ones, e.g. the atari games. Strong baselines on imitation learning, e.g. dagger or dart, are also omitted from the experiments. The evaluation of atari games is by itself a bit strange since no noise in the reward was added (line 326). The paper claims that there is anyway noise given by the real-world application but, to the best of my knowledge, no noise is present in the simulation and the reward observed exactly match the real rewards (I might be wrong on this last point, and clarification in the text could solve the issue).

Overall, I am convinced that the paper is about an important and timely problem, i.e. learning a sensorimotor policy in the presence of noise in the training signal. This problem is relevant for a very large community, including machine learning, robotics, and computer vision. In addition, I really appreciate that an effort was done to cover under the same umbrella a lot of different learning frameworks (RL, Imitation Learning, Co-training). However, I believe that the framework is still not up to its claims and both clarifications in the theoretical part and more extensive experiments could significantly strengthen the contribution.

On a less important side, I think that the current presentation, despite being enough to enable reproducibility, lacks clarity and is sometimes difficult to follow. The main problem is the fact that many terms are used to indicate similar things (e.g. Figure 1, why is the loss function Y indicated as a peer agent? This could cause confusion with the policy co-training part). Similarly, it would be nice to always explicitly state where in the appendix the proofs of the lemma/theorems can be found (this is currently not the case of theorem 2). Finally, it would be nice to put the comparison with the strong imitation learning baselines (GAIL and SQIL) in the main paper.

----------------

Issue solved after rebuttal.

**Time Spent Reviewing:**

5

---

> ### Author Response · Authors · 2021-08-10
> **Response to Reviewer xvWU**
>
> We thank the reviewer for thoughtful reviews and comments. We are glad the reviewer believes “this paper is about an important and timely problem”, “relevant to a very large community”, “extensive experiments on a lot of different learning frameworks”. We address the concerns as follows.
>
> **Q1**: Connection with prior work ([34] in paper) was not adequately discussed. \
> **A1**: Our work is **indeed inspired** by existing work on correlation agreement (CA, [34] in paper) and generalizes similar ideas to the sequential decision-making problems. More specifically, we first provide a unified formulation of the **weakly supervised policy learning problems** and then provide instantiations on RL and BC (PeerPL) to perform policy evaluation using CA. Different from the noisy labels in the supervised learning setting considered in [34], the weak supervision for our problems can be either **perturbed rewards** or **imperfect expert demonstrations** - which leads to several algorithm variants. We conduct extensive experiments to evaluate the proposed framework on three challenging tasks (RL with perturbed reward, BC with imperfect experts, and Co-Training for policy learning). Finally, we **rigorously study the convergence rates or sample complexity** for RL/BC (Theorem 1, 2). We apologize for not stating the relation with [34] properly in related work. We will clarify the connection with [34] clearly in the revision.
>
> **Q2**: Whether under the assumption of a large enough sample size and constant noise, even the policy without any peer reward would converge to the optimum?  Lemma 1 (line 193) shows the noisy supervision only differs up to a constant with respect to the proposed peer supervision method, how will the constant affect the final performance? \
> **A2**: Yes, the policy with noisy rewards can also converge to the optimum in this ideal setting (sufficient samples for phased Q-Learning). As pointed out, this is because the affine transformation of reward (independent of the noise level) will not affect the convergence to the optimal policy (Line 580) if the noise is distinguishable ($e_{-} + e_{+} < 1$). However, as shown in the **Remark (Line 193 - 200)**, the magnitude of noise terms of noisy reward $\tilde{r}$ becomes much larger than proposed $\tilde{r}_{peer}$ in the high-noise regime which will dilute the informativeness of $\mathbb{E}[r]$.
>
> On the contrary, $\mathbb{E}[\tilde{r}_{peer}]$ contains a moderate constant noise thus maintaining more useful training signals of the true reward in practice (see experimental results, the models with noisy rewards rarely converge). We highly encourage the reviewer to check the response **Q1/A1 for reviewer 58fG** in which we provide a more rigorous analysis of the error bound for the statistical estimation for noisy rewards and peer rewards. It answers why peer rewards are beneficial from the “sample complexity” perspective.
>
>
> **Q3**: Comparison with previous work is not complete (only partial) \
> **Q3.1**: It would be interesting to see how the approach would perform in comparison to [7] in Atari. \
> **A3.1**: Thanks for the suggestion. We only conduct comparisons on CartPole and Pendulum with [7] for ease of experiments. To further alleviate the reviewer’s concern, we conducted an extra experiment on Pong (one environment on Atari):
>
> | $e_{-} = e_{+} = 0.2$ | $\mathcal{R}_{avg}$ | $T_{converge}$ |
> |:---------------------:|:-------------------:|:--------------:|
> |  surrogate reward [7] |    $19.3 \pm 0.4$   |   $1.48 * 1e^7$  |
> |          ours (PeerRL)         |    $19.1 \pm 0.3$   |   $1.13 * 1e^7$  |
>
> Specifically, we set the noise rate as $e_{-} = e_{+} = 0.2$ and recorded the average score ($\mathcal R_{avg}$) and iterations taken for convergence (number of iterations when the $\mathcal{R}_{avg}$ starts becoming larger than 18.5). As shown in Table, our proposed method can achieve similar performance as [7] while yielding faster convergence (similar to CartPole and Pendulum). Note that our method offloads the burdens of prior knowledge for the confusion matrix. We will work on adding more noise rates and environments in the final version if time permits.
>
> **Q3.2**: Strong baselines on imitation learning, e.g. dagger or dart, are also omitted from the experiments. \
> **A3.2**: Thanks for the suggestion. We agree that these baselines can be used as a reference in the evaluation. We omitted the numbers from previous papers since they cannot cover all environments we evaluated (only some environments like Pong and Breakout). We will work on adding more reference baselines on imitation learning in the revision. We kindly note that these baselines cannot be compared with PeerBC directly (our algorithm is not guaranteed to perform better than all state-of-the-art imitation learning algorithms) so it should not be treated as an unfixable weakness.
>
> **Q4**: The evaluation of atari games is by itself a bit strange since no noise in the reward was added (line 326). The paper claims that there is anyway noise given by the real-world application but, to the best of my knowledge, no noise is present in the simulation and the reward observed exactly matches the real rewards. \
> **A4**: First of all, we would like to clarify that in the behavioral cloning (BC) setting, the weak supervision is the demonstrations (or actions) sampled from experts rather than the rewards in the RL case.
>
> Instead of stating it “strange” or treating it as a weakness, we would like to say it is actually an interesting observation or benefits of the proposed method we want to share with the community. Different from “RL with perturbed reward” which adds synthetic noise following the assumptions (or noise model) in theory, we aim to solve a more challenging and practical problem for BC. Specifically, we use a not fully-converged PPO agent as the expert (treating it as a noisy version of the optimal policy approximately) which generates the demonstrations. From the experiments on Atari, we observe that our method can even recover a better policy than PPO experts just by training on weak demonstrations! In this case, the error rates are also unknown which again varies the effectiveness of our approach in recovering true policy even without the prior knowledge of noise. It also demonstrates the generalizability of the proposed approach in solving practical problems!
>
> For the co-training experiments, we move one step further. We adopt the same setting as previous work [R1] (no synthetic noise or weak expert) and just training from scratch. In this case, the weak supervision comes from the demonstrations from the other agent.
>
> [R1] Co-training for Policy Learning. Song et al., UAI 2019.
>
>
> **Q5**: Why is the loss function Y indicated as a peer agent? This could cause confusion with the policy co-training part. \
> **A5**: Thanks for pointing it out. There is a little abusion with the terminology “peer agent” which causes some confusion. We want to clarify that in the co-training part, the peer agent just refers to the other agent since there are two agents in this setting. We will clarify it in the revision.
>
> **Q6**: It would be nice to put the comparison with the strong imitation learning baselines (GAIL and SQIL) in the main paper. \
> **A6**: Thanks for the suggestion! We will put these results in the main paper in the revision.

---

> > ### Comment · Reviewer_xvWU · 2021-08-20
> > **Thanks for the clarifications!**
> >
> > Thank you very much for taking the time to answer my questions! I now better see the connection with previous work.
> > However, I am not completely satisfied by the answer to the theoretical part, which in the end seems to be mainly supported by practical observations (Answer A2). This adds to the comments raised by R4, which seem to raise similar concerns about some key parts of the proof.
> >
> > Regarding the new experiments, I think the paper could benefit from better experimental results (this comment was not fully answered in the rebuttal), but I see this issue as less concerning than the previous one. For these reasons, I am willing to keep my evaluation.

---

> > > ### Author Response · Authors · 2021-08-20
> > > **Thanks for the follow-up comments!**
> > >
> > > Thanks a lot for the comments after rebuttal. To your major concerns in the theoretical part, we want to highlight that it is also supported by the theoretical analysis (Please refer to Remark1 and Q1/A1 for reviewer 58fG). It provides from another perspective that why peer reward is superior to baseline. The reviewer 58fG pointed that the variance of peer reward is larger than the baseline which is true, but it does not mean our solution is worse since **neither peer reward nor baseline approaches are unbiased estimation of true reward** (misunderstood by Reviewer 58fG). Compared to baseline, our approach is *less biased* in the RL setting and has the benefits of noise-invariance $\mathbb E[\tilde{r}_\text{peer}] = \mathbb E[r_\text{peer}]$. For the behavioral cloning setting, please note that the noisy reward has no way to retrieve the true actions thus no convergence guarantee for noisy reward. On the contrary, our PeerBC solution has the guarantee to converge to the optimal policy (as supported in paper [34]).

---

> > > ### Author Response · Authors · 2021-08-27
> > > **Further discussions on the theoretical analysis**
> > >
> > > We thank the reviewer again for raising thoughtful questions! After more discussions with R4 (Reviewer 58fG), we would like to provide more clarifications on your concerns:
> > >
> > > First of all, we would like to mention that the theoretical analysis for RL (Lemma 1 and Theorem 1) is rigorous and correct: (1) the peer reward is indeed noise invariant - $\mathbb E[\tilde{r}_\text{peer}] = \mathbb E[r_\text{peer}]$; (2) under given assumptions, Q-Learning with peer reward will converge *w.p.1.* to the optimal policy. While these observations **does not** reveal why proposed peer reward is superior over baseline (noisy reward), we believe these are interesting facts and worthwhile to be shown.
> > >
> > > To understand why peer reward is more advantageous, we provided the discussions in Remark 1 and aim to explain from the perspective of bias. We have shown that PeerRL can get a better estimation of scaled true reward (see Remark1 and first response to Reviewer 58fG), which give us some insights on why peer reward is superior in practice. **While in theory the RL algorithms is invariant to positively linearly transformed reward (Lemma A1) with infinite observations, in practice the reward reshaping will affect the performance significantly as shown in many previous works [1, 2]. Note that the magic of reward reshaping has not been revealed yet (with rigorous analysis) to our best knowledge.** Generally speaking, the convergence or sample complexity analysis for general RL algorithms (beyond tabular Q-Learning) is very challenging and not resolved yet (even for standard DQN). Therefore, in this paper, we did not aim to fully understand the learning dynamics for RL but provide insightful analysis on simplified setting.
> > >
> > > Apart from RL, the rigorous analysis on BC setting is also our technical contribution -- **in BC setting, the noisy reward cannot recover the optimal policy since under i.i.d. assumption, the algorithms are not invariant to the bias. On the contrary, our proposed PeerBC can converge to the optimal policy with sufficient observations.** This (Theorem 2) thanks to the noise invariance (Lemma 1) property of CA.
> > >
> > > Finally, for the experiments, we believe that our extensive experiments on RL, BC and Co-Training on different environments (control games, Atari) are informative and significant to show our unified CA solution can mitigate the weak supervision problem in policy learning. We will add the Atari experiments for RL (e.g., PeerRL converges faster than surrogate reward on Pong and the noisy reward can barely converge to the nearly optimal point). Moreover, we would like to include the toy examples (Markov process with two states) suggested by Reviewer 58fG to better explain why CA is more advantageous. As shown in the last several discussions (folded by OpenReview) with Reviewer 58fG, our PeerRL consistently outperforms the noisy reward under discrete or bounded noise by breaking the "tie" states.
> > >
> > > We wonder whether these clarifications are clearer and whether the discussions with Reviewer 4 (58fG) is helpful. We are pleased to take any further questions and comments.
> > >
> > > Looking forward to follow-up discussions!
> > >
> > > [1] Policy invariance under reward transformations: Theory and application to reward shaping. Andrew Y. Ng, Daishi Harada, Stuart Russell. ICML 1999. \
> > > [2] Learning to Utilize Shaping Rewards: A New Approach of Reward Shaping. Hu et al., 2020. NeurIPS 2020.

---

> > > > ### Comment · Reviewer_xvWU · 2021-08-27
> > > > **Thanks for the clarification!**
> > > >
> > > > I think my concerns on the theoretical analysis can now be flagged as solved, particularly after reading the last answer and the response to Rev. 58fG. I will therefore change my score. However, I have a question about the experiment. Why excluding baselines like DART/Dagger? The previous answer was that they require interaction with the environment, but the proposed methods (PeerRL and PeerBC) also require some form of interaction. Could the authors comment on this point? Thanks in advance!

---

> > > > > ### Author Response · Authors · 2021-08-28
> > > > > **Further clarifications on the imitation learning results**
> > > > >
> > > > > Thanks for your prompt response! We are glad that further clarifications and the discussions with Rev. 58fG address your concerns on theoretical analysis well. In what follows, we would like to discuss more about your suggestion on including more imitation learning baselines (e.g., Dagger/DART).
> > > > >
> > > > > First of all, we thank the reviewer very much for this great suggestion. We agree that these baselines are useful references in the comparison (even if no noise is added). As discussed in the previous response Q3.2/A3.2, we are working on adding those baselines in the imitation learning experiments. To enable more IL baselines, we have built our approach based on existing implementations (frameworks: https://github.com/DLR-RM/stable-baselines3 and https://github.com/HumanCompatibleAI/imitation). We conducted additional experiments experiment on the `CartPole-v1` environment. Similarly, we trained an imperfect expert (not fully converged PPO) to provide the weak demonstrations. Specifically, the PPO is trained for 50k iterations and we unroll the trajectories to obtain demonstrations in 16 episodes (in total 3521 state-action pairs). Then we compared BC, PeerBC with other IL baselines including DAgger [1], AIRL [2], GAIL [3]. The results are shown in the following table:
> > > > >
> > > > > | CartPole-v1 |       Score      |
> > > > > |-------------|:----------------:|
> > > > > | BC          | 220.1 $\pm$ 62.3 |
> > > > > | PeerBC      | 259.0 $\pm$ 44.9 |
> > > > > | DAgger [1]      | 279.5 $\pm$ 58.0 |
> > > > > | Peer-DAgger | 469.1 $\pm$ 74.1 |
> > > > > | AIRL [2]       |  95.2 $\pm$ 21.0 |
> > > > > | GAIL [3]       | 218.8 $\pm$ 47.0 |
> > > > >
> > > > > **Since DAgger deals with the distribution shift issue, not the weak supervision, we believe the results of PeerBC are not directly comparable to DAgger (as said in the previous response - also the reason we exclude the results of DAgger before).** To decouple the potential factors, we proposed another algorithm Peer-DAgger which adds the CA objective in the BC agent, but trains in the DAgger manner (querying the imperfect expert to augment the training sets). As shown in the Table, we found that Peer-DAgger outperforms DAgger significantly, which again verifies the effectiveness of CA mechanism. The learning curves are shown in this anonymous link: https://ibb.co/BsFVrP2, where the pink & orange curves denote the DAgger (two runs) and gray & blue curves denote the Peer-DAgger (two runs). We can observe that Peer-DAgger converges faster and better consistently.
> > > > >
> > > > > We will report these results and provide corresponding analysis in the revision. We hope this additional experiment addresses your concern. Any further questions and comments are appreciated!
> > > > >
> > > > > [1] A Reduction of Imitation Learning and Structured Prediction to No-Regret Online Learning. Ross et al., 2011. \
> > > > > [2] Generative Adversarial Imitation Learning. Ho et al., 2017. \
> > > > > [3] Learning Robust Rewards with Adversarial Inverse Reinforcement Learning. Fu et al., 2018.

---

> > > > > > ### Comment · Reviewer_xvWU · 2021-08-30
> > > > > > **Follow-up experiment**
> > > > > >
> > > > > > Thanks for this new set of experiments, which clearly decouples the problem of distribution shift with the one of weak supervision. What I still don't like is the high sensitivity to hyper-parameters, which however can be solved in future work. Therefore, I am willing to support acceptance.

---

> > > > > > > ### Author Response · Authors · 2021-08-30
> > > > > > > **Thanks for the discussion and valuable comments!**
> > > > > > >
> > > > > > > Thanks a lot for actively participating in the discussions, raising insightful questions! We are glad that your concerns are well addressed. We will make the presentation clearer and include the additional DAgger experiment in the revision.
> > > > > > >
> > > > > > > As for the choice of peer penalty coefficient $\xi$, we agree that it might introduce additional overheads, and more rigorous studies on the effect of parameter $\xi$ merit further study in future work. However, we did not tune this hyperparameter extensively in practice since we found our method works robustly in a wide range of $\xi$. In the paper, we conducted the sensitivity analysis of $\xi$ in Table 2 and Appendix C.4 (Figure A2, A4). We found that as long as the $\xi$ is not overly large (e.g., $\xi > 1$), our solution can work very robustly ($\xi=0.1$ to $\xi=0.4$). In the experiments, we just pick $\xi=0.1$ or 0.2 which can already lead to competitive performance (e.g., all experiments for RL/BC and Policy-learning, the DAgger experiment, and the two-state Markov process example). Moreover, we also proposed a decay schedule of $\xi$ which further stabilizes the training of the PeerRL algorithm in Appendix C.4 since we observe that a large penalty term might not be needed in the late stage of RL training. We will highlight the choice of the $\xi$ in a Remark paragraph in the revision. Thanks for the suggestion!

---

### Official Review · Reviewer_aqkG · 2021-07-16

**Rating:** 7
**Confidence:** 4

**Summary:**

The authors propose a general and unified framework to study policy learning without strong or high-quality supervision signals. This unified framework comprises, but not limited to, the main approaches for policy learning, including those using rewards signals in reinforcement learning (RL) and demonstrations through behavioral cloning (BC). The authors also propose PeerPL, a solution for efficient policy learning using only weak supervision signals, which evaluates an agent’s learning policy with noisy signals, treated as policy of an imperfect “peer agent”. The weak supervision signal comes in the form of correlation with the imperfect peer agent’s report, while the solution encourages generalization through penalizing identical reports.

**Limitations And Societal Impact:**

There is no discussion section for potential negative societal impacts present in the main paper. Though a short description of why this is excluded is included after the reference section, with the attached submission checklist.

**Main Review:**

Originality: The originality comes from that the proposed method differs to other noisy or weak supervision methods in which prior knowledge of noisy rates, i.e. the confusion matrices of noisy signals vs true signals, are not needed. Also, the evaluation function is designed so that identical behavior with the weak agent is auto penalized, by utilizing randomly sampled signal pairs. Related works are adequately cited in both Section 1 and Section 2.2.
Quality: The intuition of how the penalization in correlation assessment is interpreted as overfitting in supervision learning sounds convincing. The theoretical proofs and experimental results look promising. The extension to co-training in partially observable environment using this framework with an RL and BC hybrid algorithm (PeerCT) is clear and well-organized, matching the overall flow of paper.
Clarity: The paper is clearly written, with toy examples which adequately illustrate the core idea of the proposed evaluation function. The experimental setup is also clear, with detailed descriptions on how “imperfect” expert demonstrations were obtained. The authors also provided some valid reasons on performance improvement from PeerPL such as implicitly encouraging higher exploration (presuming through penalizing evaluation function).
Significance: The proposal seems to provide an elegant formulation to address weak supervision signal systematically. The results of single-view co-training show that the proposed solution is feasible and perform significantly better in partially observable conditions compared to other co-training algorithms. The results could serve as good reference for subsequent works either in the field of co-training or weakly supervised policy learning.


**Time Spent Reviewing:**

4

---

> ### Author Response · Authors · 2021-08-10
> **Response to Reviewer aqkG**
>
> We thank the reviewer for thoughtful reviews and comments. We are glad that the reviewer recognized our contributions -- “clear setup”, “well-organized”, “convincing intuitions”, and “promising theoretical analysis and experimental results”. We will move the discussions of potential negative societal impacts from the checklist to the main paper in the final version.

---

### Official Review · Reviewer_Hvaz · 2021-07-17

**Rating:** 7
**Confidence:** 3

**Summary:**

In this paper, the authors propose a general framework for learning sequential decision-making policies with noisy feedback. Two categories of noisy feedback are considered, noisy reward in reinforcement learning and noisy demonstrations in behavioral cloning. The concept of “correlated agreement” (CA) is proposed as a regularization mechanism for a policy to learn from the noisy feedback without overfitting to it. An additional policy co-training setting is also considered. Empirical evaluations control games and Atari show that CA is effective in enabling policy learning from noisy feedback.

**Limitations And Societal Impact:**

Adequately addressed.

**Main Review:**

Post-rebuttal comments: thanks to the authors for providing a detailed response. All my questions are answered satisfactorily. My score will stay the same.

Originality: The application of CA to policy learning from noisy feedback is novel. The discussion of related works is adequate.

Quality: This paper is technically sound. I appreciate the theoretical analyses for PeerRL and PeerBC, which show convergence to optimal policy with CA. I have the following questions.

1. In the paragraph on BC with weak demonstration (starting at line 120), from which state distribution are the weak demonstrations sampled? I understand the noisy actions are the result of flipping from the true expert action. Are the states generated by following the true expert actions or the noisy ones?

2. I don’t quite understand the point of the toy example given in section 3.1 (starting at line 157). It is true the evaluation score is lower with CA, but does that mean that there is a better policy than the fully memorized one?

3. I would like to understand what, if any, obstacle exists to extend the noisy model from discrete to continuous. If the reward/action flipping follows a continuous distribution, is any theoretical result impacted?

Clarity: This paper is well-written and easy to read. Good job!

Significance: One limitation the authors acknowledged is that the empirical evaluations are not real-world. I think the significance would improve quite a bit with the inclusion of an example of a real robotic system, for instance. Nevertheless, I think the novelty of CA, the theoretical contributions and the strong performance on the synthetic environments provide enough significance.

Minor comment: This recent ICLR 2021 paper should be included in the related works discussion.
Behavioral Cloning from Noisy Demonstrations, Fumihiro Sasaki, Ryota Yamashina, ICLR 2021


**Time Spent Reviewing:**

3

---

> ### Author Response · Authors · 2021-08-10
> **Response to Reviewer Hvaz**
>
> Thanks for your thoughtful reviews and comments. We are glad that our work is recognized as “novel” and “technically sound”. We also appreciate the insight questions raised by the reviewer on improving the presentation clarity.
>
> **Q1**: For BC with weak supervision, are the states generated following the true expert actions or the noisy ones?
> **A1**: In theory, there is no constraint on the **state distribution** from which the weak demonstrations are sampled. That is to say, the states can be generated either following the true expert action or the noisy ones (since the source of states is not important as long as the actions are collected by the true expert). More specifically, in BC, we are treating each *state-action* pair $(s, \tilde{a})$ or weak supervision i.i.d. where  $\tilde{a}$ is the noisy observations of true expert actions $a$. Assume a true expert policy $\pi(s)$, then we have the following relation $\pi(s) = a, s \in \mathcal{S}$. Let $\tilde{a}$ denote the noisy copies of true actions $a$, our theorems show that as long as sufficient weak demonstrations are collected $\tilde{D} = \\{(s_i, \tilde{a}_i)\\}, i= 1, \dots, N$, our proposed method can recover the true expert policy $\pi(s)$. This shows that as long as the state $s$ is valid ($s \in \mathcal{S}$) for the true expert, our theoretical analysis holds.
> In experiments, to solve a more challenging problem, we directly train from the observations collected by not fully converged PPO models instead of using the true expert states and flipping the noisy actions. In this case, the state distribution follows the noisy expert distribution. However, we note that the state distribution won’t affect our theoretical analysis for BC.
>
> **Q2**: Clarity for the toy example (line 157). It is true the evaluation score is lower with CA, but does that mean that there is a better policy than the fully memorized one? \
> **A2**: Great question! In this toy example, we just want to illustrate that blind agreement on weak supervision will be penalized by CA. There might be a better policy than fully memorizing the training data in terms of the generalization ability considering the fact that the supervision is not perfect. In this simple example with only five action observations, we cannot identify a better policy due to missing assumptions (no states or noisy rate provided, insufficient data examples). However, as long as we obtain sufficient weak demonstrations, our method will converge to the optimal policy which is a better policy than fully memorizing the noisy labels in terms of the generalization ability. Thanks for pointing it out. We will make the intuition for this toy example clearer in the revision.
>
> **Q3**: What is the obstacle when extending the noisy model from discrete to continuous? \
> **A3**: There are multiple challenges in extending the noisy model to the continuous setting. Even in the standard supervised learning with noisy label setting, discrete noisy models are quite common and widely adopted in the literature [R1, R2]. This is because (1) the discrete setting is cleaner/easier for theoretical analysis (2) the confusion matrix can approximate most learnable noise (both discrete and continuous) in real applications. If we want to extend to a continuous setting for supervised learning, more assumptions on the continuous noise are required - e.g, Gaussian/Laplacian noise with a non-zero mean (zero-mean noise is much easier to solve) or noise modeled by a mixture of Gaussian. It is in general very challenging to analyze the neural networks learning dynamics with continuous and biased noise injected. Also, different assumptions may require completely different tools to analyze the optimality and convergence rates. Considering continuous noise also increases the difficulty in recovering the true reward due to more stochastic observations.
>
> Even some solutions are found in the supervised learning setting with stronger assumptions, it is non-trivial to adapt them to policy learning algorithms. Since the noise can be amplified and compounded in sequential decision-making problems, stronger assumptions might be needed to ensure the continuous noise will not affect the convergence (while it increases the variance). Furthermore, generalizing to continuous states is another challenging problem that has not been solved clearly even for standard RL settings. We believe extending the discrete noise model to a general continuous setting (not zero-mean setting) is an interesting research direction, which merits further study.
>
> [R1] “Learning from corrupted binary labels via class-probability estimation” Menon et al. ICML 2015. \
> [R2] “Learning from noisy labels” Natarajan et al. NIPS 2013
>
> **Q4**: Discussing one recent work in the related work. \
> **A4**: Thanks for providing this recent work! We will cite Sasaki et al., 2021 and add discussion in the revision.

---

### Author Response · Authors · 2021-08-10
**General response**

We would like to thank the reviewers again for their thoughtful reviews and valuable comments. We appreciate that all reviewers recognize our contributions in some aspects and raise insightful questions. In what follows, we summarize the major concerns from the reviewers and our responses.

**Clarity on why the proposed method should outperform the baselines** [Reviewers xvWU, 58fG] \
As shown in the Remark (Line 193 - 200), the magnitude of noise terms of noisy reward $\tilde{r}$ becomes much larger than proposed $\tilde{r}\_{peer}$ in the high-noise regime which will dilute the informativeness of $\mathbb{E}[r]$. On the contrary, $\mathbb{E}[\tilde{r}\_{peer}]$ contains a moderate constant noise thus maintaining more useful training signals of the true reward in practice. In **Q1/A1 for reviewer 58fG**, we provide additional analysis of the error bound for the statistical estimation for noisy rewards and peer rewards.


**Why the problem is useful and interesting** [Reviewer 58fG] \
Due to limited resources and untrustworthy sensors, the weak supervision problem is widely existent in policy learning tasks - for example, we cannot always learn from the best expert (or access the perfect expert all the time), there is some noise in recording the observations, actions or reward signals. We will add more motivations in the revision for a better story. The relevance of the problem is also recognized by other reviewers [Hvaz, aqkG, xvWU].

---

### Decision · Program_Chairs · 2021-09-27

**Decision:**

Accept (Poster)

**Comment:**

This paper was an exemplary case of the value of discussion between the reviewers and the authors. After multiple clarifications, additional experiments and even a code snippet examples, the reviewers agreed that the paper provides a valuable contribution to a problem of weak supervision in RL. I encourage the authors to address reviewers' extensive comments and update the paper for the final revision.